# Diffusion and Flow-based Copulas: Forgetting and Remembering Dependencies

**David Huk**[*]
Department of Statistics
University of Warwick
United Kingdom
David.Huk@warwick.ac.uk

**Theodoros Damoulas**
Department of Statistics, Department of Computer Science
University of Warwick
United Kingdom
T.Damoulas@warwick.ac.uk

## Abstract

Copulas are a fundamental tool for modelling multivariate dependencies in data, forming the method of choice in diverse fields and applications. However, the adoption of existing models for multimodal and high-dimensional dependencies is hindered by restrictive assumptions and poor scaling. In this work, we present methods for modelling copulas based on the principles of diffusions and flows. We design two processes that progressively *forget* inter-variable dependencies while leaving dimension-wise distributions unaffected, provably defining valid copulas at all times. We show how to obtain copula models by learning to *remember the forgotten dependencies* from each process, theoretically recovering the true copula at optimality. The first instantiation of our framework focuses on direct density estimation, while the second specialises in expedient sampling. Empirically, we demonstrate the superior performance of our proposed methods over state-of-the-art copula approaches in modelling complex and high-dimensional dependencies from scientific datasets and images. Our work enhances the representational power of copula models, empowering applications and paving the way for their adoption on larger scales and more challenging domains.

## 1 Introduction

Given a collection of $d$ continuous random variables, a simple model for their joint probability density function is the product of the corresponding $d$ univariate densities (Peterson, 1987). Of course, this omits inter-variable dependence, which is given by a $d$-dimensional density supported on the unit hypercube with uniform marginal distributions[1] called a *copula* (Sklar, 1959). Indeed, the copula uniquely and exactly represents the inter-variable dependence, unlike correlation or mutual information (Geenens, 2023), fully disentangling the marginal behaviour from the joint.

This disentanglement enables a modular approach for multivariate modelling: first, model the univariate variables independently, and second, model their dependence with a copula. This is predominant in applications where the marginal behaviour is known or requires specific properties, as copulas respect these by construction. Copulas produce well-specified probabilistically calibrated marginals to represent extreme events accurately; properties that are crucial for state-of-the-art models across fields such as weather forecasting (Cong & Brady, 2012; Huk et al., 2023), hydrology (Salvadori & De Michele, 2007; Van de Velde et al., 2023), risk management (Kole et al., 2007; Dewick & Liu, 2022), retail (Salinas et al., 2019), and causal inference (Evans & Didelez, 2024). Another popular application is to induce dependence to correct pre-existing independent models, as used in generative modelling (Tagasovska et al., 2019; Liu et al., 2024), multi-agent imitation learning (Wang et al., 2021), uncertainty quantification (Chilinski & Silva, 2020; Tagasovska et al., 2023), Bayesian non-parametrics (Huk et al., 2024), and classification calibration (Panda & Garain, 2025). Copulas are also invariant to monotonic transformations of variables, benefiting multi-objective Bayesian optimisation (Park et al., 2024). Finally, they isolate inter-variable dependencies in an

---

[*]Work partly done during an internship within the AI team at Unilink Software Ltd.

[1]In this work, marginal distributions refer to univariate dimension-wise marginals, unlike (multivariate) time-marginals common in the literature on diffusions.

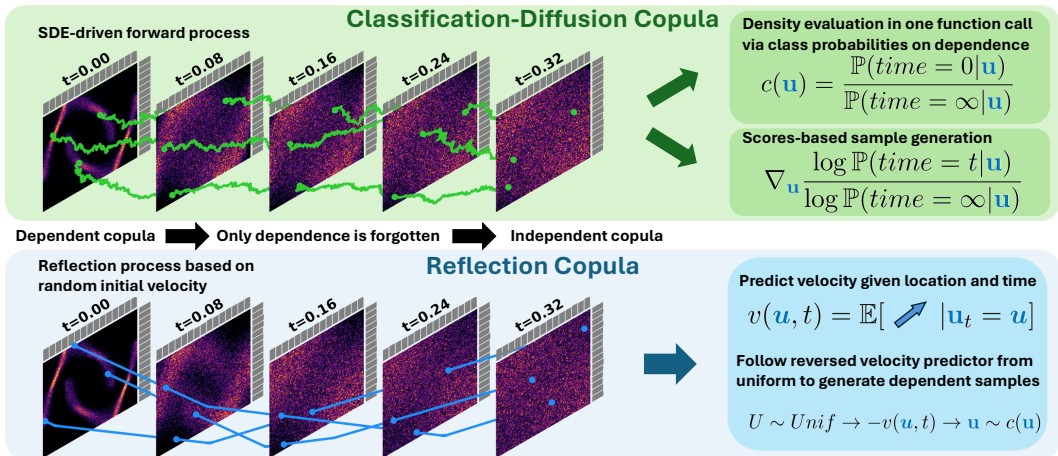

Figure 1: **Overview of proposed copula models.** We design forward processes to forget inter-variable dependencies but preserve dimension-wise marginals. Our *classification-diffusion copula* and *reflection copula* learn by remembering the forgotten dependencies of these processes.

interpretable way, facilitating the study of neuron spikes in the brain (Berkes et al., 2008; Verzelli & Sacerdote, 2019), increasing the flexibility of Bayesian networks (Elidan, 2010), enabling synthetic data generation (Patki et al., 2016; Sun et al., 2019) with privacy-preserving applications (Griesbauer et al., 2025), and empowering conformal prediction (Sun & Yu, 2023; Park & Cho, 2025).

Yet, current copula models impose restrictive assumptions limiting their effectiveness on high-dimensional and non-trivial data. For instance, Gaussian copulas (see Ch. 3 in Hofert et al. (2018)) only capture diagonal symmetric dependencies, while vine copulas (Nagler & Czado, 2016) omit parts of the dependence and explore an exponential model space with the data's dimension. Existing deep copula models suffer from mode collapse issues (Hofert et al., 2021; Janke et al., 2021) and struggle when sampling from multi-modal and high-dimensional data (Huk et al., 2025).

To address these challenges, we design copulas representing dependence purely through the formalism of diffusions and flows (Song & Ermon, 2019; Lipman et al., 2023). The first step of our analysis is to design appropriate forward processes, motivating our first research question:

> **Q1:** *How to design a stochastic process that only forgets dependence?*

We answer it by introducing two processes that maintain their marginal distributions but progressively *forget* the inter-dimension dependence. The first is grounded in stochastic differential equations with time-invariant marginal distributions. The second consists of endowing samples with random velocities and reflecting them in a hypercube through time. We show that, when applied to copula data, these processes provably preserve uniform marginal distributions, i.e., they define valid copulas at all times with vanishing dependence. This insight motivates our second question:

> **Q2:** *Can we remember the forgotten dependencies to obtain copula densities and samples?*

Motivated by the main uses of copulas, for each process, we propose a model with a separate goal: in Sec. 3, we present an effective density estimator we term the *classification-diffusion copula*, and in Sec. 4, we introduce the *reflection copula* as a generative dependence model suited for expedient sampling (See Figure 1). Theoretically, we show both models recover the ground truth copula at optimality, enabling density estimation and sampling. We summarise our contributions as follows:

1. We introduce two processes to purely forget the dependence of data with time (Proposition 2, Proposition 7), defining a spectrum from a dependent to an independent copula.

2. We propose a *classification-diffusion copula* which provably learns the dependence in data (Theorem 5), outputs copula densities in a single model evaluation (Proposition 3), and leverages diffusion algorithms for sampling (Proposition 4).

3. We introduce the *reflection copula* to learn the probability path of forgotten dependencies and generate samples by following this path in reverse (Proposition 8).

4. We achieve state-of-the-art results in dependence modelling, surpassing existing classical and deep copulas on a variety of complex real-world data. Our copulas are the first instance able to scale to such high-dimensional and multimodal dependencies (see Fig. 5).

## 2 BACKGROUND ON COPULA MODELS

**Notation and setup.** We denote observed data $\mathbf{x} = (x^1, \ldots, x^d) \in \mathcal{X} \subseteq \mathbb{R}^d$ coming from a cumulative distribution function (CDF) $P(.)$ with probability density function (PDF) $\boldsymbol{p}(.)$, and refer to $\mathcal{X}$ as the *data scale*. We assume that dimension-wise marginal CDFs $\{P^i(.)\}_{i=1}^d$ and PDFs $\{p^i(.)\}_{i=1}^d$ are given. Next, we denote copula observations as $\mathbf{u} = (u^1, \ldots, u^d) \in [0,1]^d$ with $u^i = P^i(x^i)$, calling this the *copula scale*. Finally, for $\Phi$ the univariate standard Gaussian CDF, we also work with variables $\mathbf{z} = (z^1, \ldots, z^d) \in \mathbb{R}^d$ for $z^i = \Phi^{-1}(u^i)$, which we say are on the *Gaussian scale*. We always denote time in subscripts, keeping superscripts for the dimension.

**The task.** We are given data $\mathbf{x}$ and know marginal distributions either through assumptions on the model or by pre-emptively estimating them. These marginal distributions are used to transform the data $\mathbf{x}$ to copula scale observations $\mathbf{u}$ encapsulating dependence. The copula modelling task then consists of providing a valid copula model for $\mathbf{u}$ to produce samples or likelihood evaluations.

An example of this is multivariate time series forecasting, where one strives to describe the future joint distribution over many variables, such as the state of multiple agents in imitation learning, see Apdx B. Such uses of copulas are a useful way of dealing with the exceedingly large dimension-alities of multivariate timeseries by decomposing them into marginal forecasts which are stitched together through a copula, decomposing the original problem into manageable subtasks. First, one estimates the individual distributions for each agent separately, possibly utilising expert knowledge to ensure calibration, and secondly, unites all univariate distributions with a copula to prescribe a given dependence structure.

**Copula-based modelling.** A copula is a joint probability distribution over univariate marginal distributions, owing its name to the Latin term for a link or bond. Indeed, the copula links together independent marginals, infusing them with dependence to form a valid joint distribution:

**Theorem 1** (Sklar (1959))**.** *Let $P$ be a $d$-dimensional distribution on $\mathcal{X} \subset \mathbb{R}^d$ with continuous marginal cumulative distribution functions $P^1, P^2, \ldots, P^d$. Then there exists a unique copula distribution $C : [0,1]^d \mapsto [0,1]$ such that for all $\mathbf{x} = (\mathbf{x}^1, \mathbf{x}^2, \ldots, \mathbf{x}^d) \in \mathbb{R}^d$:*

$$P(x^1, \ldots, x^d) = C(P^1(x^1), \ldots, P^d(x^d)).$$

*And if a probability density function $\boldsymbol{p} : \mathcal{X} \mapsto \mathbb{R}$ is available:*

$$\boldsymbol{p}(x^1, \ldots, x^d) = p^1(x^1) \cdot \ldots \cdot p^d(x^d) \cdot c(P^1(x^1), \ldots, P^d(x^d)), \quad (1)$$

*where $p^1(x^1), \ldots, p^d(x^d)$ are the marginal PDFs, and $c : [0,1]^d \mapsto (0, \infty)$ is the copula PDF.*

This theorem allows for a two-step estimation process of a joint density: First, model marginal distributions, and second, model the dependence between them with a copula. This grants great flexibility, as marginals can be tailored to the specific problem in mind, while the choice of the copula model can prescribe particular joint behaviours. Eq. (1) is used to obtain density evaluations, and sampling is done by applying the inverse marginal distributions $P^{i-1}(u^i)$ to copula samples $\mathbf{u}$. Correctly modelling the copula is therefore of utmost importance to faithfully represent data.

**Popular copulas used in machine learning** Most works adopt one of two models due to their ability to quickly produce densities and samples, even in high dimensions. Firstly, the Gaussian copula is generally the first copula to be implemented in applications, and is used in Elidan (2010); Patki et al. (2016); Salinas et al. (2019); Wang et al. (2021); Huk et al. (2023); Panda & Garain (2025). Second, the vine copula (see Aas et al. (2009); Nagler & Czado (2016)) is the *de facto* model for copula density evaluation and sampling, being used in the majority of cases not using a Gaussian, as in Tagasovska et al. (2019; 2023); Huk et al. (2024); Park et al. (2024); Park & Cho

(2025); Griesbauer et al. (2025). Other model classes, such as Archimedean copulas, have seen some applications (Liu et al., 2025) but remain limited to diagonal and symmetric dependencies and scale poorly to high dimensions. Their deep variants have overcome the large computational complexity of early works (Ling et al., 2020), and have been shown to effectively scale to dimensions as high as 20 (Ng et al., 2021; 2022). While the work of Ng et al. (2021) remains bound to diagonal relationships of an Archimedean copula, Ng et al. (2022) gains flexibility via the Archimax class. Fully general deep copulas have been explored in Hofert et al. (2021); Janke et al. (2021) as generative metric-matching networks and in Kamthe et al. (2021) as normalising flows.

**Current state of the art in copula models.** Recently, Huk et al. (2025) showed that any copula density is equivalent to a binary Bayes optimal classifier between dependent and independent data with the same marginal distributions. Their framework yields flexible models, achieving state-of-the-art performance in copula modelling for complex dependencies. Unfortunately, it only estimates densities, meaning sampling requires expensive Markov Chain Monte Carlo scaling as $\mathcal{O}(d^{4/3})$ with the data dimension $d$, making this method inappropriate for multimodal or high-dimensional settings.

In the next Section, we propose our first novel framework for copula modelling, by relying on diffusion processes specifically designed to forget dependence. In doing so, we depart from the ratio copula's dichotomy between dependence and independence to study a continuous range of decaying dependencies. This framework shift lets us leverage diffusion-based sampling, making our method scalable to high-dimensional and multimodal dependencies (Azangulov et al., 2024).

# 3 CLASSIFICATION-DIFFUSION COPULA

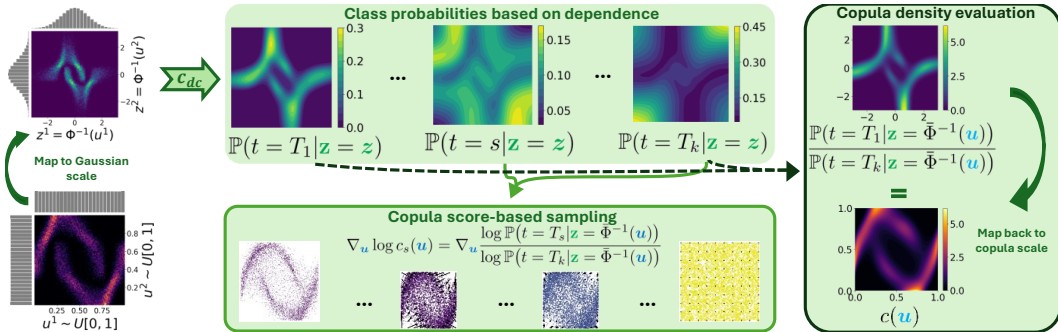

Figure 2: **Classifier-Diffusion Copula.** We map copula data $\mathbf{u}$ to Gaussian scale data $\mathbf{z}$, to which we apply an Ornstein-Uhlenbeck process up to a time $t$. We train a multinomial classifier $c_{dc}$ to identify the diffusion time $t$ based on $\mathbf{z}$'s dependence. This classifier recovers the copula density.

In this Section, we develop *classification-diffusion copulas* whose strength lies in performing accurate density evaluation in a single function call, while also generating samples through diffusion-based sampling methods. A diagram of the approach is shown in Fig. 2.

## 3.1 FORWARD PROCESS TO FORGET DEPENDENCE

Since copulas are density ratios (Huk et al., 2025), their densities are preserved under diffeomorphisms (Choi et al., 2021). Thus, noticing that $\mathbf{x}, \mathbf{u}$ and $\mathbf{z}$ stem from invertible mappings of each other, we model the copula of $\mathbf{x}$ through the copula of $\mathbf{z}$ given they share the same dependence. This lets us define a process on $\mathbb{R}^d$ instead of $[0,1]^d$. We augment our variables $\mathbf{z}$ with a time dimension to denote $\mathbf{z}_t$ as the variables following a density $\tilde{p}_t$ for $t \in [0,\infty)$, with the original data (on the Gaussian scale) at $t = 0$. In the following, we formalise and answer our first research question:

> **Q1 (a):** *How to define a process which preserves $\tilde{p}_t^i = \tilde{p}_s^i$ for any dimension $i$ at all times $t, s \geq 0$, while diffusing the copula $c_t$ of $\mathbf{z}_t$ to independence as $t \to \infty$?*

**Ornstein-Uhlenbeck on the Gaussian scale.** As $\mathbf{u}$ is marginally uniform, $\mathbf{z}$ is marginally standard Gaussian with density $\mathcal{N}(z_t; 0, 1)$ by the probability integral transform. This insight motivates the following process on $\mathbf{z}_t = (z_t^1, \ldots, z_t^d)$ for $t \geq 0$:

$$d\mathbf{z}_t = -\mathbf{z}_t \, dt + \sqrt{2} \, d\mathcal{B}_t, \tag{2}$$

where $\mathcal{B}_t$ denotes Brownian motion. We call Eq. (2) the forward process of our model as it moves the data $\mathbf{z}_0$ forward in time to $\mathbf{z}_t$. This is an Ornstein–Uhlenbeck (OU) process that is independent across dimensions, and has the property of having the standard $d-$dimensional Gaussian as its stationary distribution in the limit of time $t \mapsto \infty$ (Särkkä & Solin, 2019), fulfilling our first desiderata on marginals. For diffusion-based sampling, the OU process enjoys faster convergence to its limiting distribution (Brešar & Mijatović, 2024) and is easier to learn from (Reu et al., 2024) compared to other stochastic processes, which motivates our choice by Theorem 2 of Chen et al. (2023). It is also easy to simulate due to being analytically tractable. To fulfil our desiderata on the copula, we inspect what happens to $\mathbf{u}_t := \Phi^{-1}(\mathbf{z}_t)$ on the copula scale, yielding the following result. The proof is given in Apdx. A.1.

**Proposition 2** (Convergence to independence copula). *For copula samples $\mathbf{z}_0$ on the Gaussian scale, consider $\mathbf{u}_t^i = \Phi(\mathbf{z}_t^i)$ with $\mathbf{z}_t^i$ as defined through the process in Eq. (2). Then, this process maintains uniform marginal distributions at all times:*

$$\mathbf{u}_t^i \sim U(0, 1), \quad \forall t \geq 0, i \in \{1, \ldots, d\}.$$

*Moreover, denote by $c_t$ the copula of $\mathbf{z}_t$ under Eq. (2). Then, $c_t$ converges to the independence copula as $t \to \infty$ in the Kullback-Leibler divergence with rate $\mathcal{O}(e^{-2t})$.*

For complex distributions, a process with a limiting distribution closer to the data distribution in the Kullback-Leibler divergence is preferable (Rhodes et al., 2020; Chen et al., 2023). Hence, we modify Eq. (2), by adding a correlation matrix $\Sigma$:

$$d\mathbf{z}_t = -\mathbf{z}_t \, dt + \sqrt{2}\Sigma^{1/2} \, d\mathcal{B}_t, \text{ with } \Sigma_{(i,i)} = 1 \text{ for } i \in \{i, \ldots, d\}. \tag{3}$$

This process converges to $\mathcal{N}(\mathbf{0}, \Sigma)$, and we prove its convergence in Proposition 9 of the Appendix. In what follows, we will focus on Eq. (2), and defer the derivations for Eq. (3) to Apdx. A.5.

## 3.2 REMEMBERING DEPENDENCIES FOR COPULA DENSITIES AND SAMPLING

We present the *classification-diffusion copula* to learn the true copula density from process (2). As standard diffusion models require expensive numerical integration (Song et al., 2021) for likelihoods, we instead answer the following question to facilitate density-reliant copula applications:

> **Q2 (a):** *How to obtain direct copula densities without foregoing effective sampling?*

**Classifying diffused data based on remaining dependence.** We begin by discretising the forward process of Eq. (2) into times[2] $T_1, T_2, \ldots, T_k$ with $T_1 = 0, T_k = \infty$, each with respective densities denoted $\tilde{\boldsymbol{p}}_{T_1}, \ldots, \tilde{\boldsymbol{p}}_{T_k}$. We define the *classification-diffusion copula* as a function which, given a data point $\mathbf{z}$, outputs a vector of $k$ probabilities corresponding to the likelihood of $\mathbf{z}$ originating from each of the $k$ times $c_{dc} : \mathbb{R}^d \mapsto \{h \in [0, 1]^k | \sum_{t=1}^k h^t = 1\}$, with:

$$c_{dc}(\boldsymbol{z}) = \big(\mathbb{P}(t = T_1 | \mathbf{z} = \boldsymbol{z}), \ldots, \mathbb{P}(t = T_k | \mathbf{z} = \boldsymbol{z})\big). \tag{4}$$

The obtained convergence rates of the forward process in the previous Section motivate our choice of time steps $T_1, T_2, \ldots, T_k$ in practice to achieve proportional changes in the dependence in line with our convergence rate in Prop. 2, see Apdx for further details. B. This vector (4) of correct class probabilities can be linked to the copula density as follows:

**Proposition 3** ($c_{dc}$ copula density evaluation). *For the classification task as defined in Eq. (4), the true copula density $c(\boldsymbol{u})$ at a given value $\boldsymbol{u} \in [0, 1]^d$ is equal to the following probability ratio:*

$$c(\boldsymbol{u}) = \mathbb{P}(t = T_1 | \mathbf{z} = \bar{\Phi}^{-1}(\boldsymbol{u})) / \mathbb{P}(t = T_k | \mathbf{z} = \bar{\Phi}^{-1}(\boldsymbol{u})),$$

*where $\bar{\Phi}^{-1}(\boldsymbol{u})$ is the inverse standard Gaussian CDF applied dimensions-wise to $\boldsymbol{u}$.*

---

[2]We take $T_k$ as a large time when the forward process is expected to be at stationarity, see Apdx. B.3.

We include a proof in Apdx. A.2. Next, we show how to express the score of a copula at a given dependence level in terms of the class probabilities alone, enabling score-based sampling. The proof is in Apdx. A.3 and advances the technique of Yadin et al. (2024) to copula densities.

**Proposition 4** ($c_{dc}$ copula score-based sampling). *Consider the classification task as defined in Eq. (4), and let $c_s$ be the copula of $\mathbf{u}_s = \bar{\Phi}(\mathbf{z}_s)$ with $\bar{\Phi}$ the standard Gaussian CDF applied element-wise, and $\mathbf{z}_s$ defined by the process of Eq. (2) at time $s \in \{T_1, \ldots, T_k\}$. Then, for $\odot$ denoting element-wise multiplication, the true copula score $\nabla_{\mathbf{u}} \log c_s(\mathbf{u})$ is given by:*

$$\nabla_{\mathbf{u}} \log c_s(\mathbf{u}) = w(\mathbf{u}) \odot \Big[ \nabla_{\mathbf{z}} \log(\mathbb{P}(t = T_s | \mathbf{z} = \mathbf{z})) - \nabla_{\mathbf{z}} \log(\mathbb{P}(t = T_k | \mathbf{z} = \mathbf{z})) \Big]_{\mathbf{z} = \bar{\Phi}^{-1}(\mathbf{u})}$$

*where for $\phi$ as the Gaussian pdf, we have $w(\mathbf{u}) := \big( \frac{1}{\phi\big(\Phi^{-1}(u^1)\big)}, \ldots, \frac{1}{\phi\big(\Phi^{-1}(u^d)\big)} \big)$.*

Consequently, given an accurate model for the class probabilities, we can employ powerful diffusion model algorithms to obtain copula samples through Langevin dynamics (Song & Ermon, 2019), see Alg. 1 in Apdx. B. Importantly, under the popular assumption that the true distribution is supported on a manifold with dimension $< d$, diffusion-based sampling scales more efficiently to datasets of higher dimensions (Azangulov et al., 2024). This is in stark contrast to current deep copula methods (Huk et al., 2025) requiring Hamiltonian Monte Carlo in the ambient dimension $d$.

**Training the $c_{dc}$ model.** Having demonstrated how to leverage class probabilities to perform copula density estimation and copula sampling, our next contribution is to prove how to estimate such a model correctly. To obtain both accurate densities and high-fidelity samples, we adopt a mixture loss between cross-entropy for the dependence level and mean squared error for the score, which has been empirically successful in previous work (Yadin et al., 2024). We provide a novel result that theoretically shows its correctness with a proof in Apdx. A.4.

**Theorem 5** (Loss function for $c_{dc}$). *For the classification task defined in Eq. (4), and for a family of multinomial models $c_{dc}(\mathbf{z}; \theta) : \mathbb{R}^d \mapsto \{h \in [0,1]^k | \sum_{t=1}^k h^t = 1\}$ indexed by parameters $\theta$, assume there exists a set of parameters $\Theta^* = \{\theta : c_{dc}(\mathbf{z}; \theta) = \big( \mathbb{P}(t = T_1 | \mathbf{z} = \mathbf{z}), \ldots, \mathbb{P}(t = T_k | \mathbf{z} = \mathbf{z}) \big)\}$, for all $\mathbf{z} \in \mathbb{R}^d$. Then, for any weight $\alpha > 0$, $\{\theta : \theta = \arg\min_\theta \mathcal{L}_{c_{dc}}(\theta)\} = \Theta^*$ for the loss:*

$$\mathcal{L}_{c_{dc}}(\theta) = \alpha \cdot \sum_{s=1}^k \mathbb{E}_{\mathbf{z} \sim \tilde{\mathbf{p}}_{T_s}} \big[ -\log c_{dc}^{(s)}(\mathbf{z}; \theta) \big]$$

$$+ \sum_{s=1}^k \mathbb{E}_{\substack{\mathbf{z}_{T_1} \sim \tilde{\mathbf{p}}_{T_1} \\ \epsilon \sim \mathcal{N}(\mathbf{0}, \mathbf{I})}} \big[ ||\hat{\epsilon}_s(c_{dc}(e^{-T_s} \cdot \mathbf{z}_{T_1} + \sqrt{1 - e^{-2 \cdot T_s}} \cdot \epsilon; \theta)) - \epsilon||^2 \big]$$

*where $c_{dc}^{(s)}(\mathbf{z}; \theta)$ denotes the s-th component of the vector $c_{dc}(\mathbf{z}; \theta)$, and with*

$$\hat{\epsilon}_s(c_{dc}(\mathbf{z}; \theta)) := \sqrt{1 - e^{-2 \cdot T_s}} \cdot (\nabla_{\mathbf{z}} \log c_{dc}^{(k)}(\mathbf{z}; \theta) - \nabla_{\mathbf{z}} \log c_{dc}^{(s)}(\mathbf{z}; \theta) + \mathbf{z}).$$

We choose $\alpha$ such that both loss terms are of the same magnitude, following Yadin et al. (2024), with an ablation study on $\alpha$ in Apdx. B.4. As this result only holds at optimality, we validate sample properties in practice in Apdx. B.5 via statistical tests, rank diagnostics and calibration metrics, with visualisations in Apdx. B.6. We give a full overview for training the $c_{dc}$ in Alg. 2 of Apdx. B.

## 4 REFLECTION COPULA

As many copula applications only require sampling, we design a generative copula model through processes that forget dependence, inspired by flow methods. We provide a diagram in Fig. 3.

### 4.1 FORGETTING DEPENDENCE ON THE COPULA HYPERCUBE

In distinction from Sec. 3, here we directly design our method on the $[0,1]^d$ copula scale. To obtain a stochastic path from a dependent to an independent copula, we answer:

**Q1 (b):** *How to forget dependence directly on $[0,1]^d$ and maintain uniform marginals?*

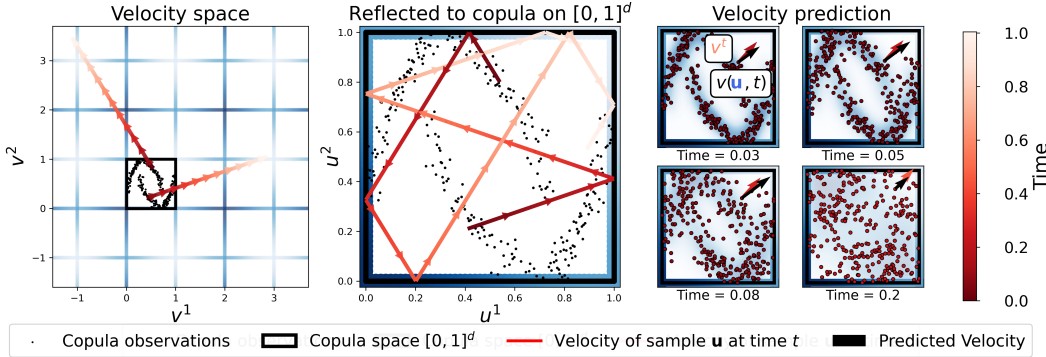

Figure 3: **Reflection copula design.** (Left panel) Copula data in black are given red velocities to move with time. (Middle panel) Trajectories are reflected from $\mathbb{R}^d$ to $[0,1]^d$ following mirrored blue outlines. (Third panel) Reflected trajectories diffuse the copula with time. The reflection copula then learns a velocity predictor $v(\mathbf{u}, t) := \mathbb{E}[\mathbf{v}_t | \mathbf{u} = \mathbf{u}]$ for the average velocity, needed for sampling.

We start with copula observations $\mathbf{u} \in [0,1]^d$, and again augment our space with a time dimension $t \in [0,\infty)$. We let $\mathbf{u}_0 := \mathbf{u}$ and endow samples with velocities $\mathbf{v}_0 \sim \mathcal{N}(\mathbf{0}, \mathbf{I}_d)$. We define a process for the evolution of sample-velocity pairs $(\mathbf{u}_t, \mathbf{v}_t)$ through time via dimension-wise reflections.

**Definition 6** (Reflection process). *For initial samples $\mathbf{u}_0 \in [0,1]^d$ and velocities $\mathbf{v}_0 \sim \mathcal{N}(\mathbf{0}, \mathbf{I}_d)$, define the univariate hypercube reflection operator $\mathcal{R} : \mathbb{R} \times \mathbb{R} \mapsto [0,1] \times \mathbb{R}$ as*

$$\mathcal{R}(x, y) = \begin{cases} (x - \lfloor x \rfloor, y), & \text{if } \lfloor x \rfloor \text{ is even,} \\ (1 - x + \lfloor x \rfloor, -y), & \text{if } \lfloor x \rfloor \text{ is odd} \end{cases} . \tag{5}$$

*The reflection process for sample-velocity pairs $(\mathbf{u}_t, \mathbf{v}_t)$ at time $t \geq 0$ is defined dimension-wise as*

$$(\mathbf{u}_t^i, \mathbf{v}_t^i) = \mathcal{R}(\mathbf{u}_0^i + t \cdot \mathbf{v}_0^i, \mathbf{v}_0^i). \tag{6}$$

The process starts from a copula distribution, selects a direction according to a Gaussian, and bounces around the $[0,1]^d$ hypercube through time. The only randomness in this process comes from the velocity, so the trajectory is deterministic once $\mathbf{v}_0$ is sampled, see the diagram in Fig. 3. We prove that this process defines a valid copula at all times as it converges to independence.

**Proposition 7** (Reflection converges to independence). *For any initial point $\mathbf{u}_0 \in [0,1]^d$ with $\mathbf{v}_0 \sim \mathcal{N}(\mathbf{0}, \mathbf{I_d})$ as in Definition 6, the reflected point $\mathbf{u}_t$ converges in distribution to the independent copula on the hypercube as $t \to \infty$. Further, if $\mathbf{u}_0$ follows a copula distribution, then $\mathbf{u}_t$ also follows some copula distribution $c_t$ with uniform marginals for any time $t > 0$.*

We give a proof in Apdx. A.6. Note that the copula $c_t$ is generally different from the true copula $c$ and will have less dependence with time. As such, this process creates a stochastic interpolation between a dependent copula and an independent one with time, gradually reducing dependence.

### 4.2 REMEMBERING DEPENDENCE BY PREDICTING VELOCITIES

To obtain a generative copula model from this process, we answer the following:

> **Q2 (b):** *How to generate samples by learning from the reflection process in Definition 6?*

Intuitively, as velocities are the only randomness in the system, if we learn the expected velocity, we should learn the system's average behaviour. We leverage a connection between our velocities and the probability path governing the change in copula distributions through time, as is shown in the following result. We include a proof of its applicability to the reflection process in Apdx. A.7.

**Proposition 8** (Holderrieth et al. (2024)). *Consider the reflection process introduced in Eq. (6) applied to copula samples $\mathbf{u}_0 \sim c(\mathbf{u})$ with velocities $\mathbf{v}_0 \sim \mathcal{N}(\mathbf{0}, \mathbf{I}_d)$. Let $v^*(\mathbf{u}, t) = \mathbb{E}[\mathbf{v}_t | \mathbf{u}_t = \mathbf{u}]$ be the expected velocity at time $t \geq 0$ and location $\mathbf{u} \in [0,1]^d$, and denote by $c_T$ the copula of*

*samples* $\mathbf{u}_T$ *at time* $T$. *Then, starting from a value* $\mathbf{u}_T \sim c_T(\mathbf{u}_T)$ *and following the probability path given by the ordinary differential equation*

$$\frac{d}{dt}\boldsymbol{u}_t = v^*(\boldsymbol{u}, t) \tag{7}$$

*backwards from* $T \to 0$, *the destination point comes from the initial copula distribution* $\mathbf{u}_0 \sim c(\mathbf{u})$.

In other words, the expected velocity of this process is enough to generate samples. However, we first require a sample $\mathbf{u}_T$ at time $T$ from our process. As we show in Proposition 7, $\mathbf{u}_T \sim U[0,1]^d$ for $T \to \infty$. Therefore, we initialise Eq. (7) with uniform samples for a suitably large time $T$ (see Apdx. B.3) and numerically solve it to generate copula samples. We illustrate this procedure in Fig. 4 and Alg. 3.

**Optimal velocity predictor.** As $v^*(\boldsymbol{u}, t)$ is not available in closed form, our *reflection copula* model learns to approximate the average velocity with a velocity predictor $v_\theta(\boldsymbol{u}, t) : [0,1]^d \times [0, \infty) \mapsto \mathbb{R}^d$. Analogous to flow models, for a sufficiently rich model class, minimising the mean square error of predictions against sample velocities of our systems recovers the expected velocity:

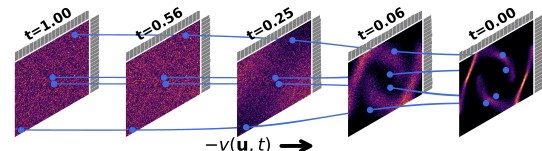

Figure 4: **Reflection copula sampling.** Initialised from uniform samples at $t = T$, following $-v^*(\boldsymbol{u}, t)$ preserves marginals and generates $\mathbf{u}_0 \sim c(\mathbf{u})$ at $t = 0$.

$$\theta^* = \arg\min_\theta \mathbb{E}_{\mathbf{u}_0 \sim c, \mathbf{v}_0 \sim \mathcal{N}(\mathbf{0}, \mathbf{I}_d)} \big[ ||v_\theta(\mathbf{u}_t, t) - \mathbf{v}_t||^2 \big] \Rightarrow v_{\theta^*}(\boldsymbol{u}, t) = \mathbb{E}[\mathbf{v}_t | \mathbf{u}_t = \boldsymbol{u}]$$

where the pair $(\mathbf{u}_t, \mathbf{v}_t)$ follows the reflection process initialised at $\mathbf{u}_0, \mathbf{v}_0$. We verify our models learn valid copulas in practice in Apdx. B.5 via statistical tests, rank diagnostics and calibration metrics. Training details are given in Alg. 4 of Apdx. B.

## 5 RELATED WORK

**Deep copulas.** In Hofert et al. (2021), the dependence is modelled on the Gaussian scale with a moment matching network. This model is improved with a generative adversarial network design by Janke et al. (2021). However, both methods are only able to output samples, precluding their use in density-reliant copula applications, and suffer from common mode collapse issues. In Kamthe et al. (2021), a normalising flow approximates the copula to obtain densities and samples. Theoretically, their model also obtains valid copulas at optimality, but unlike our work, it lacks similar results to our Props. 2, 7, and 9 on "forgetting and remembering" mechanisms to ensure the model evolves through valid copula states. Specific copula classes, such as Archimax (Ng et al., 2022) and Archimedean copulas (Ng et al., 2021), which focus on extremes, have also been estimated with deep networks. These works have a model class which exactly matches a given parametric copula family, which lends important robustness, especially for extrapolating to tail behaviours, but limits applications for the more general dependencies considered in our work. Closest to our $c_{dc}$ copula is the ratio copula of Huk et al. (2025), which we extend to the multinomial setting with diffusion-based dependence levels, motivated by the connection of Yadin et al. (2024). Our reflection copula is unique in leveraging principles of Holderrieth et al. (2024) in the context of a generative copula on $[0,1]^d$ with preservation of marginal distributions. Our designs are the first instances of copulas scaling to complex and high-dimensional dependencies, such as in image data.

**Copulas in diffusion works.** To the best of our knowledge, our work is the first to utilise diffusion and flow principles for copula modelling. In Bibbona et al. (2016), serial dependence in univariate diffusions is connected to bivariate copulas, which differs from our goal of flexible dependence models. In Liu et al. (2024), autoregressive language models are used as discrete copulas to improve sampling of text diffusion models. Finally, processes with Gaussian distributions are commonly studied in diffusion works (Kingma et al., 2021; Ou et al., 2025; Pierret & Galerne, 2025; Sahoo et al., 2025), but no model to our knowledge leverages their univariate marginal preservation.

## 6 EXPERIMENTS

In Sec. 6.1, we model challenging dependencies in synthetic data generation (Patki et al., 2016; Sun et al., 2019) and multi-agent imitation learning (Wang et al., 2021) tasks. In Sec. 6.2, we show that our models are the first instance of copulas able to capture high-dimensional structured dependencies found in images. Appendix B contains full implementation details, including computational times (B.1), an analytical simulation study (B.2), a showcase of the forward convergence rates (B.3), an ablation of our mixture loss (B.4), sample uniformity diagnostics with statistical tests and calibration metrics (B.5), as well as sample quality visualisations (B.6). Code to reproduce results is publicly available at https://github.com/Huk-David/Diffusion-and-Flow-based-Copulas.

**Benchmarks and metrics.** We compare to models whose bespoke goal is to learn dependencies. We use the Gaussian and vine copulas as baselines, and the Implicit Generative Copula (IGC) of Janke et al. (2021) and the Ratio copula of Huk et al. (2025) as state of the art deep copula models. Following (Nagler et al., 2017; Huk et al., 2025), we use the copula log-likelihood (LL) when available and use the Wasserstein-2 metric (W2) to assess samples. We compute the Frobenius norm (Frob) between samples and observations' Kendall's tau matrices (Hofert et al., 2018), measuring pair-wise dependence[3], and use the Frechet Inception Distance (FID) (Heusel et al., 2017) to assess image samples. The LL and Frob both purely measure the fit of the dependence structure, while W2 and FID also consider marginal uniformity of samples. All metrics are shown with standard deviations across 10 independent runs.

### 6.1 MULTIMODAL DEPENDENCE OF SCIENTIFIC DATASETS

Table 1: **Modelling dependence of scientific datasets.** Compared to existing copulas, the $c_{dc}$ model achieves better LL evaluations, while both the $c_{dc}$ and reflection copula obtain better samples.

| Model | Magic ($n = 19020, d = 10$) | | | Dry_Bean ($n = 13611, d = 16$) | | | Robocup ($n = 135607, d = 20$) | | |
| --- | --- | --- | --- | --- | --- | --- | --- | --- | --- |
| | LL ↑ | W2 ↓ | Frob ↓ | LL ↑ | W2 ↓ | Frob ↓ | LL ↑ | W2 ↓ | Frob ↓ |
| Gaussian | $3.92_{\pm 0.06}$ | $1.76_{\pm 0.02}$ | $0.27_{\pm 0.06}$ | $40.09_{\pm 0.29}$ | $1.57_{\pm 0.02}$ | $0.40_{\pm 0.04}$ | $0.22_{\pm 0.00}$ | $3.96_{\pm 0.01}$ | $\mathbf{0.45_{\pm 0.03}}$ |
| Vine | $6.59_{\pm 0.07}$ | $1.44_{\pm 0.01}$ | $0.30_{\pm 0.05}$ | $32.75_{\pm 0.14}$ | $1.35_{\pm 0.03}$ | $0.95_{\pm 0.07}$ | $1.80_{\pm 0.00}$ | $3.96_{\pm 0.01}$ | $0.60_{\pm 0.04}$ |
| Ratio | $6.76_{\pm 0.38}$ | $2.26_{\pm 0.79}$ | $1.24_{\pm 0.76}$ | $48.21_{\pm 0.89}$ | $2.54_{\pm 0.27}$ | $2.25_{\pm 0.55}$ | $2.30_{\pm 0.33}$ | $3.93_{\pm 0.08}$ | $0.59_{\pm 0.05}$ |
| IGC | – | $1.69_{\pm 0.04}$ | $1.24_{\pm 0.20}$ | – | $1.66_{\pm 0.01}$ | $2.31_{\pm 0.02}$ | – | $4.13_{\pm 0.02}$ | $2.85_{\pm 0.12}$ |
| $C_{dc}$ (ours) | $\mathbf{18.65_{\pm 4.85}}$ | $\mathbf{1.33_{\pm 0.03}}$ | $\mathbf{0.21_{\pm 0.05}}$ | $\mathbf{50.21_{\pm 0.82}}$ | $\mathbf{1.12_{\pm 0.03}}$ | $\mathbf{0.35_{\pm 0.08}}$ | $\mathbf{3.40_{\pm 0.37}}$ | $3.87_{\pm 0.03}$ | $0.51_{\pm 0.02}$ |
| Reflection (ours) | – | $1.34_{\pm 0.03}$ | $0.28_{\pm 0.07}$ | – | $1.35_{\pm 0.08}$ | $0.47_{\pm 0.16}$ | – | $\mathbf{3.84_{\pm 0.03}}$ | $0.49_{\pm 0.02}$ |

We choose datasets with complex dependencies between scientific variables for non-parametric copula evaluation. Used in Nagler et al. (2017); Janke et al. (2021), Magic has telescope observation data, while Dry_Bean measures dry bean shapes. Following Wang et al. (2021), we model the behaviour of a team of ten robots playing football by controlling their vertical and horizontal movements as a probabilistic multivariate time-series (which is a common use of copulas, see Apdx. B for full details). In Tab. 1, our copulas achieve the best results for LL and W2, and perform comparably to the Gaussian copula for Frob. We report pair plots in Apdx. B.6 for visualisation.

### 6.2 HIGH-DIMENSIONAL STRUCTURED DEPENDENCIES OF IMAGES

To demonstrate our methods' scalability, we require datasets with pre-estimated marginal distributions. As this scale of experiments has not been fully studied in previous works, we rely on image datasets where CDFs are trivial and the complexity resides in the dependence structures; our goal is not to compete against image models. Furthermore, our choice of monochrome images provides an analogue to spatial problems, such as in climate research, where copulas are widely adopted.

**Sampling and density estimation in high-dimensions.** We follow Huk et al. (2025) and use the digits and MNIST datasets as high-dimensional examples with complex dependencies but simple marginals. We further use a grey version of the Cifar10 dataset restricted to the two classes of cars and trucks. Notably, the dependence of copula data for MNIST is very specific for pixels in the middle of the image with noise at the edges, while Cifar requires the whole image to be dependent (see Fig. 5). In Tab. 2, the $c_{dc}$ obtains the highest LL, while for W2 our methods

---

[3]Gaussian copulas are advantaged on this metric due to their closed-form relationship to Kendall's tau.

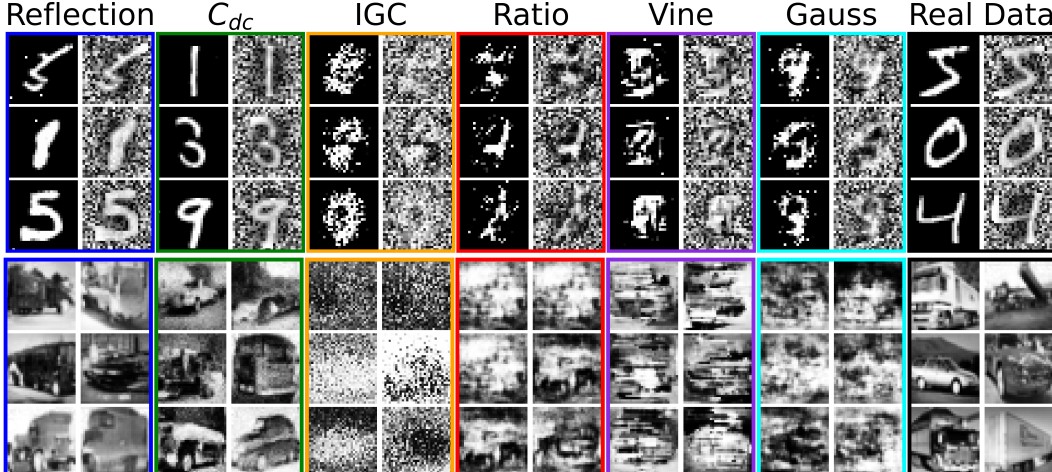

Figure 5: **Copula image samples**: Data and copula scale `MNIST` samples (top rows), copula scale `Cifar` samples (bottom rows). Only our designs accurately represent the complex dependencies.

Table 2: **Copulas of high-dimensional data:** Our methods outperform existing copulas at capturing multimodal and high-dimensional dependencies according to LL and sample-based metrics.

| | digits ($n=1797, d=64$) | | | MNIST ($n=60000, d=784$) | | | Cifar ($n=10000, d=1024$) | | |
|---|---|---|---|---|---|---|---|---|---|
| Model | LL ↑ | W2 ↓ | FID ↓ | LL ↑ | W2 ↓ | FID ↓ | LL ↑ | W2 ↓ | FID ↓ |
| Gaussian | $10.74_{\pm0.13}$ | $8.13_{\pm0.02}$ | $5.74_{\pm0.71}$ | $115.84_{\pm0.14}$ | $35.59_{\pm0.03}$ | $102.56_{\pm2.61}$ | $1258.40_{\pm5.69}$ | $30.62_{\pm0.08}$ | $140.12_{\pm2.41}$ |
| Vine | $11.20_{\pm0.86}$ | $8.20_{\pm0.01}$ | $6.06_{\pm0.73}$ | $198.10_{\pm0.40}$ | $36.30_{\pm0.05}$ | $86.48_{\pm2.82}$ | NaN | $33.84_{\pm0.15}$ | $100.04_{\pm2.52}$ |
| Ratio | $13.29_{\pm2.75}$ | $8.42_{\pm0.42}$ | $6.04_{\pm0.97}$ | $334.42_{\pm45.91}$ | $35.98_{\pm0.05}$ | $66.56_{\pm17.76}$ | $1348.18_{\pm12.31}$ | $49.91_{\pm19.22}$ | $134.41_{\pm33.99}$ |
| IGC | – | $9.52_{\pm0.15}$ | $25.41_{\pm4.35}$ | – | $36.37_{\pm0.14}$ | $128.87_{\pm5.31}$ | – | $33.09_{\pm0.30}$ | $269.68_{\pm8.03}$ |
| $C_{dc}$ (ours) | $\mathbf{13.80_{\pm1.30}}$ | $\mathbf{6.97_{\pm0.03}}$ | $15.24_{\pm0.92}$ | $\mathbf{346.70_{\pm2.52}}$ | $\mathbf{33.64_{\pm0.03}}$ | $\mathbf{7.38_{\pm0.19}}$ | $\mathbf{1470.75_{\pm24.90}}$ | $\mathbf{28.67_{\pm0.50}}$ | $\mathbf{80.51_{\pm17.32}}$ |
| Reflection (ours) | – | $\mathbf{7.86_{\pm0.07}}$ | $\mathbf{5.50_{\pm1.36}}$ | – | $35.02_{\pm0.40}$ | $9.13_{\pm0.90}$ | – | $32.40_{\pm2.08}$ | $42.14_{\pm3.23}$ |

outperform benchmarks with the exception of the $c_{dc}$ for FID on `digits` and the reflection copula for W2 on `Cifar`. The vine failed to obtain LLs on `Cifar` regardless of hyperparameters. From Fig. 5, our Reflection copula samples are smoother, while the $c_{dc}$ has slightly noisier and grainier samples, as reflected by the FID values on `digits`, and `Cifar`, see also Figs. 21, 22, 23. We believe this stems from the stochasticity of the sampling procedure as detailed in Alg. 1, which is a beneficial feature to maintain uniform marginals (see Apdx. B.5). Existing copulas do not scale well to these high dimensions and are unable to represent their dependencies.

## 7 CONCLUSION

In this work, we theoretically show that our processes only forget dependence, and that our models remember the true copula from them. Our work suggests new types of generative copulas using stochastic processes exactly targeting dependence, making copula models scale to complex dependencies in $d > 1000$ for the first time. Our copulas offer powerful options for copula likelihoods and sampling, empowering applications across the varied fields utilising such models.

**Limitations and future work.** While our studied processes define valid copulas by design, our models are only guaranteed to represent copulas at optimality. An open question remains about how to design architectures to preserve properties such as uniformity of marginals and normalisation of the density for any model parameterisation. Relatedly, the question of how to sample with diffusion and flow models to best represent the model's true distribution is an active area of research which is equally relevant to our work. Further, diffusion models based on different marginal-preserving processes could better suit copulas with extreme dependence. The ingredients for that are a CDF mapping to a latent space with a diffusion whose invariant distribution is that CDF, which is applied independently across dimensions. Similarly, different velocity distributions for the Reflection copula could lead to better inductive biases. Finally, as copulas require continuous marginal densities, an avenue for future research consists of extending our approach to discrete variables following the discrete copula notion of Geenens (2020).

## AUTHOR CONTRIBUTIONS

David Huk originated the idea, designed the project, wrote the code, ran the experiments, wrote the paper, and derived the proofs. Theodoros Damoulas supervised the project, provided valuable feedback, and helped edit the paper.

## ACKNOWLEDGMENTS

We are grateful for the helpful discussion and comments of the reviewers and Area Chair, as well as those of Peter E. Holderrieth, Miha Brešar, Aleksandar Mijatovic, Thomas Nagler, and Ji Won Park. Furthermore, David Huk would like to thank Unilink Software Ltd for supporting his attendance at the conference. David Huk is funded by the Center for Doctoral Training in Mathematical Sciences at Warwick. Theodoros Damoulas acknowledges support from a UKRI Turing AI acceleration Fellowship [EP/V02678X/1].

## REPRODUCIBILITY STATEMENT.

We take the following measures to ensure full reproducibility of our work. We give all details of proofs and theoretical results in Appendix A with statements presented in the main text. We give full details of data preprocessing and model implementation in Apdx. B. We provide Algs. 1, 2, 3, and 4 with instructions to train and sample from our proposed models. Code to reproduce experiments is publicly available at https://github.com/Huk-David/Diffusion-and-Flow-based-Copulas.

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

## A    PROOFS

We restate the propositions and theorems for completeness, and provide their proofs afterwards.

### A.1    PROPOSITION 2

**Proposition 2.**    *For copula samples $\mathbf{z}_0$ on the Gaussian scale, consider $\mathbf{u}_t^i = \Phi(\mathbf{z}_t^i)$ with $\mathbf{z}_t^i$ as defined through the process in Equation (2). Then, this process maintains uniform marginal distributions at all times:*

$$\mathbf{u}_t^i \sim U(0,1), \quad \forall t \geq 0, i \in \{1, \ldots, d\}.$$

*Moreover, denote by $c_t$ the copula of $\mathbf{z}_t$. Then, $c_t$ converges to the independence copula as $t \to \infty$ in the KL divergence with rate $\mathcal{O}(e^{-2t})$.*

The proof consists of converting results from stochastic differential equations on $\mathbf{z}$ from the Gaussian scale to $\mathbf{u}$ on the copula scale.

*Proof.* To begin, rewrite Equation (2) in the following identity:

$$\mathbf{z}_t = e^{-t} \cdot \mathbf{z}_0 + \sqrt{1 - e^{-2t}} \cdot \epsilon, \quad \epsilon \sim \mathcal{N}(\mathbf{0}, \mathbf{I}).$$

For a single dimension, note that $\mathbf{z}_0^i$ is standard Gaussian by assumption and $\epsilon^i$ is standard Gaussian by construction, and so $\mathbf{z}_t^i$ is also Gaussian. Its mean is thus zero and its variance is $e^{-2t} + 1 - e^{-2t} = 1$. Therefore, with $\Phi$ the univariate standard Gaussian CDF, $\mathbf{u}_t^i = \Phi(\mathbf{z}_t^i)$ is uniform.

For the second part, with the OU process as defines through Equation (2), under suitable assumptions on the data distribution $p$, the convergence to the stationary $\mathcal{N}(\mathbf{0}, \mathbf{I})$ is well known with rate

$\mathcal{O}(Ce^{-2t})$ for some constant $C$ which depends on the furthest mode of the initial data distribution $\tilde{p}_0$ (see *e.g.* Brešar & Mijatović (2024) for the (DATA) assumption and Proposition 1.2). To convert this convergence from $\mathbf{z}_t$ on the Gaussian scale to $\mathbf{u}_t$ on the copula scale, notice that the mapping from one to the other is the standard Gaussian CDF applied dimension-wise, which is an invertible reparametrisation. The result is obtained via the KL's invariance under a change of variable. $\qquad\square$

## A.2 PROPOSITION 3

**Proposition 3.** *For the classification task as defined in Equation (4), the true copula density $c(\mathbf{u})$ at a given value $\mathbf{u} \in [0,1]^d$ is equal to the following probability ratio:*

$$c(\mathbf{u}) = \frac{\mathbb{P}(t = T_1 | \mathbf{z} = \bar{\Phi}^{-1}(\mathbf{u}))}{\mathbb{P}(t = T_k | \mathbf{z} = \bar{\Phi}^{-1}(\mathbf{u}))},$$

*where $\bar{\Phi}^{-1}(\mathbf{u})$ is the inverse standard Gaussian CDF applied dimensions-wise to $\mathbf{u}$.*

*Proof.* We consider the classification problem of Equation (4), and in particular the class probabilities given by $\big(\mathbb{P}(t = T_1 | \mathbf{z} = \mathbf{z})), \ldots, \mathbb{P}(t = T_k | \mathbf{z} = \mathbf{z}))\big)$. We define the mixture of data likelihoods between classes by $\tilde{p}_+(\mathbf{z}) := \sum_{s=1}^{k} \pi_s \cdot \tilde{p}_{T_s}(\mathbf{z})$, where the prior probabilities $\pi_s$ are assumed identical for all classes $s \in \{1, \ldots, k\}$. By Bayes's rule, we show the following equivalence, similar to identities used in Density Ratio Estimation works (Srivastava et al., 2023):

$$\begin{aligned}
\frac{\tilde{p}_{T_1}(\mathbf{z})}{\tilde{p}_{T_k}(\mathbf{z})} &= \frac{\tilde{p}_{T_1}(\mathbf{z} = \mathbf{z} | t = T_1)}{\tilde{p}_{T_k}(\mathbf{z} = \mathbf{z} | t = T_k)} = \frac{\tilde{p}_{T_1}(\mathbf{z} = \mathbf{z}, t = T_1)/\pi_{T_1}}{\tilde{p}_{T_k}(\mathbf{z} = \mathbf{z}, t = T_k)/\pi_{T_k}} \\
&= \frac{\tilde{p}_{T_1}(t = T_1 | \mathbf{z} = \mathbf{z}) \cdot \tilde{p}_+(\mathbf{z})}{\tilde{p}_{T_k}(t = T_k | \mathbf{z} = \mathbf{z}) \cdot \tilde{p}_+(\mathbf{z})} = \frac{\tilde{p}_{T_1}(t = T_1 | \mathbf{z} = \mathbf{z})}{\tilde{p}_{T_k}(t = T_k | \mathbf{z} = \mathbf{z})},
\end{aligned}$$

where the prior odds cancel out in the first line, as they are assumed to be identical. To conclude, note that a copula is a ratio of densities as shown in Section 3 of Huk et al. (2025), and such ratios are preserved under invertible functions by Lemma 1 in Choi et al. (2021), meaning

$$c(\mathbf{u}) = \frac{\tilde{p}(\Phi^{-1}(u^1), \ldots, \Phi^{-1}(u^d))}{\prod_{i=1}^d \mathcal{N}(\Phi^{-1}(u^i); 0, 1)} = \frac{\tilde{p}_{T_1}(\bar{\Phi}^{-1}(\mathbf{u}))}{\tilde{p}_{T_k}(\bar{\Phi}^{-1}(\mathbf{u}))} = \frac{\mathbb{P}(t = T_1 | \mathbf{z} = \bar{\Phi}^{-1}(\mathbf{u}))}{\mathbb{P}(t = T_k | \mathbf{z} = \bar{\Phi}^{-1}(\mathbf{u}))},$$

where $\bar{\Phi}^{-1}(\mathbf{u})$ is the standard Gaussian CDF applied dimensions-wise to $\mathbf{u}$. $\qquad\square$

## A.3 PROPOSITION 4

**Proposition 4.** *Consider the classification task as defined in Eq. (4), and let $c_s$ be the copula of $\mathbf{u}_s = \bar{\Phi}(\mathbf{z}_s)$ with $\bar{\Phi}$ the standard Gaussian CDF applied element-wise, and $\mathbf{z}_s$ defined by the process of Eq. (2) at time $s \in \{T_1, \ldots, T_k\}$. Then, the true copula score $\nabla_{\mathbf{u}} \log c_s(\mathbf{u})$ is given by:*

$$\nabla_{\mathbf{u}} \log c_s(\mathbf{u}) = w(\mathbf{u}) \odot \Big[\nabla_{\mathbf{z}} \log(\mathbb{P}(t = T_s | \mathbf{z} = \mathbf{z})) - \nabla_{\mathbf{z}} \log(\mathbb{P}(t = T_k | \mathbf{z} = \mathbf{z}))\Big]_{\mathbf{z} = \bar{\Phi}^{-1}(\mathbf{u})}$$

*where $\odot$ is element-wise multiplication and $w(\mathbf{u}) := \big(\frac{1}{\phi(\Phi^{-1}(u^1))}, \ldots, \frac{1}{\phi(\Phi^{-1}(u^d))}\big)$.*

*Proof.* We begin by showing a link between the class probabilities and the gradient of the joint density, which has been identified in existing works (Yadin et al., 2024; Wu et al., 2024), and then utilise this link within our diffusion copula context to derive an expression for the copula score. To start, consider the classification task as defined in Equation (4) with class probabilities $\big(\mathbb{P}(t = T_1 | \mathbf{z} = \mathbf{z})), \ldots, \mathbb{P}(t = T_k | \mathbf{z} = \mathbf{z}))\big)$, and mixture of likelihood $\tilde{p}_+(\mathbf{z})$ with weights $\pi_s$ $s \in \{1, \ldots, k\}$ as in the proof of Proposition 3. Rewrite the following two conditional probabilities:

$$\tilde{p}_s(\mathbf{z} = \mathbf{z} | t = T_s) = \frac{\mathbb{P}(t = T_s | \mathbf{z} = \mathbf{z}) \cdot \tilde{p}_+(\mathbf{z})}{\pi_s} \tag{8}$$

and

$$\tilde{p}_{T_k}(\mathbf{z} = \mathbf{z} | t = T_k) = \frac{\mathbb{P}(t = T_k | \mathbf{z} = \mathbf{z}) \cdot \tilde{p}_+(\mathbf{z})}{\pi_{T_k}}. \tag{9}$$

Next, replace $\tilde{\boldsymbol{p}}_+(\boldsymbol{z})$ in Equation (8) with the appropriate quantity from Equation (9):

$$\tilde{\boldsymbol{p}}_s(\mathbf{z} = z | t = T_s) = \frac{\pi_{T_k}}{\pi_s} \cdot \frac{\mathbb{P}(t = T_s | \mathbf{z} = z)}{\mathbb{P}(t = T_k | \mathbf{z} = z)} \cdot \tilde{\boldsymbol{p}}_{T_k}(\mathbf{z} = z | t = T_k)$$

where the class probabilities cancel by assuming equal prior odds. We take the log and then the gradient with respect to $\boldsymbol{z}$ on both sides (where densities are positive and differentiable), yielding:

$$\nabla_{\boldsymbol{z}} \log(\tilde{\boldsymbol{p}}_s(\mathbf{z} = z | t = T_s)) = \tag{10}$$

$$\nabla_{\boldsymbol{z}} \log(\mathbb{P}(t = T_s | \mathbf{z} = z)) - \nabla_{\boldsymbol{z}} \log(\mathbb{P}(t = T_k | \mathbf{z} = z)) + \nabla_{\boldsymbol{z}} \log(\tilde{\boldsymbol{p}}_{T_k}(\mathbf{z} = z | t = T_k)) \tag{11}$$

Here, we note that a copula decomposition following Sklar's theorem leads to the following expression of the log scores of a copula $c$ and joint density $\boldsymbol{p}$ with marginals $p^i, i \in \{1, \ldots, d\}$:

$$\nabla_{\boldsymbol{z}} \log c(P^1(z^1), \ldots, P^d(z^d)) = \nabla_{\boldsymbol{z}} \log \boldsymbol{p}(\boldsymbol{z}) - \nabla_{\boldsymbol{z}} \sum_{i=1}^{d} \log p^i(z^i).$$

In our case, for the joint density on the Gaussian scale $\tilde{\boldsymbol{p}}_s(\mathbf{z} = z | t = T_s)$, its marginal densities are standard Gaussian by construction as a consequence of the process in Equation (2). Thus, their marginal scores are $-z_i$, resulting in the following expression for the score of the log copula $c_s$ corresponding to the inter-variable dependence at time $s$:

$$\nabla_{\boldsymbol{z}} \log c_s(\Phi(z^1), \ldots, \Phi(z^d)) = \nabla_{\boldsymbol{z}} \log \tilde{\boldsymbol{p}}_s(\mathbf{z} = z | t = T_s) + \boldsymbol{z}. \tag{12}$$

We can now replace the score of the joint log density in Equation (12) with the expression in Equation (11), obtaining:

$$\nabla_{\boldsymbol{z}} \log c_s(\Phi(z^1), \ldots, \Phi(z^d)) = \nabla_{\boldsymbol{z}} \log(\mathbb{P}(t = T_s | \mathbf{z} = z)) - \nabla_{\boldsymbol{z}} \log(\mathbb{P}(t = T_k | \mathbf{z} = z))$$

where $\nabla_{\boldsymbol{z}} \log(\tilde{\boldsymbol{p}}_{T_k}(\mathbf{z} = z | t = T_k))$ cancels out with $\boldsymbol{z}$ as $\tilde{\boldsymbol{p}}_{T_k}$ is a standard Gaussian by construction. Finally, we use a change of variable from $\boldsymbol{z}$ to $\boldsymbol{u} = \bar{\Phi}(\boldsymbol{z})$ (the standard Gaussian CDF applied dimensions-wise to $\boldsymbol{u}$), obtaining the final expression:

$$\nabla_{\boldsymbol{u}} \log c_s(\boldsymbol{u}) = w(\boldsymbol{u}) \odot \left[ \nabla_{\boldsymbol{z}} \log(\mathbb{P}(t = T_s | \mathbf{z} = z)) - \nabla_{\boldsymbol{z}} \log(\mathbb{P}(t = T_k | \mathbf{z} = z)) \right]_{\boldsymbol{z} = \bar{\Phi}^{-1}(\boldsymbol{u})}$$

where $\odot$ is element-wise multiplication and $w(\boldsymbol{u})$ is defined as:

$$w(\boldsymbol{u}) := \Big( \frac{1}{\phi\big(\Phi^{-1}(u_1)\big)}, \ldots, \frac{1}{\phi\big(\Phi^{-1}(u_d)\big)} \Big).$$

$\square$

## A.4 THEOREM 5

**Theorem 5.** *For the classification task defined in Equation (4), and for a sufficiently broad family of multinomial models $c_{dc}(\boldsymbol{z}; \theta) : \mathbb{R}^d \mapsto \{h \in [0,1]^k | \sum_{t=1}^{k} h^t = 1\}$ indexed by parameters $\theta$, assume there exists a set of parameters $\Theta^* = \{\theta : c_{dc}(\boldsymbol{z}; \theta) = \big(\mathbb{P}(t = T_1 | \mathbf{z} = z), \ldots, \mathbb{P}(t = T_k | \mathbf{z} = z)\big)\}$ for which the model is equal to the vector of class probabilities for all $\boldsymbol{z} \in \mathbb{R}^d$. Then, for any weight $\alpha > 0$, $\{\theta : \theta = \arg\min_\theta \mathcal{L}_{c_{dc}}(\theta)\} = \Theta^*$ for the loss:*

$$\mathcal{L}_{c_{dc}}(\theta) = \alpha \cdot \sum_{s=1}^{k} \mathbb{E}_{\boldsymbol{z} \sim \tilde{\boldsymbol{p}}_{T_s}} \big[ -\log c_{dc}^{(s)}(\boldsymbol{z}; \theta) \big]$$

$$+ \sum_{s=1}^{k} \mathbb{E}_{\substack{\boldsymbol{z}_{T_1} \sim \tilde{\boldsymbol{p}}_{T_1} \\ \epsilon \sim \mathcal{N}(\mathbf{0}, \mathbf{I})}} \big[ ||\hat{\epsilon}_s(c_{dc}(e^{-T_s} \cdot \boldsymbol{z}_{T_1} + \sqrt{1 - e^{-2 \cdot T_s}} \cdot \epsilon; \theta)) - \epsilon||^2 \big]$$

*where $c_{dc}^{(s)}(\boldsymbol{z}; \theta)$ denotes the $s$-th component of the vector $c_{dc}(\boldsymbol{z}; \theta)$, and with*

$$\hat{\epsilon}_s(c_{dc}(\boldsymbol{z}; \theta)) := \sqrt{1 - e^{-2 \cdot T_s}} \cdot (\nabla_{\boldsymbol{z}} \log c_{dc}^{(k)}(\boldsymbol{z}; \theta) - \nabla_{\boldsymbol{z}} \log c_{dc}^{(s)}(\boldsymbol{z}; \theta) + \boldsymbol{z}).$$

*Proof.* We first describe the set of minimisers for each component of the loss and draw our conclusion afterwards.

The first term corresponds to the cross-entropy loss with prior probabilities per class:

$$\mathcal{L}_{CE}(\theta) := \sum_{s=1}^{k} \mathbb{E}_{\mathbf{z} \sim \tilde{\mathbf{p}}_{T_s}} \big[ -\log c_{dc}^{(s)}(\mathbf{z}; \theta) \big].$$

Since cross-entropy is a strictly proper scoring rule (Dawid et al., 2016), under the assumption that the model class contains the true distribution, minimising the loss with respect to the model parameters is equivalent to recovering the true data-generating process. Thus, we have $\theta_{CE}^* := \arg\min_\theta \mathcal{L}_{CE}(\theta)$ if and only if:

$$c_{dc}(\mathbf{z}; \theta_{CE}^*) = \big( \mathbb{P}(t = T_1 | \mathbf{z} = \mathbf{z}), \dots, \mathbb{P}(t = T_k | \mathbf{z} = \mathbf{z}) \big)$$

for all $\mathbf{z}$ over the support $\mathbb{R}^d$. Denote the set of all such minimisers as $\Theta_{CE}$.

The second term is the mean squared error between the predicted noise and the true added noise to a sample:

$$\mathcal{L}_{MSE}(\theta) := \sum_{s=1}^{k} \mathbb{E}_{\substack{\mathbf{z}_{T_1} \sim \tilde{\mathbf{p}}_{T_1} \\ \epsilon \sim \mathcal{N}(\mathbf{0}, \mathbf{I})}} \big[ ||\hat{\epsilon}_s(c_{dc}(e^{-T_s} \cdot \mathbf{z}_{T_1} + \sqrt{1 - e^{-2 \cdot T_s}} \cdot \epsilon; \theta)) - \epsilon||^2 \big].$$

This expression is minimised when, for all $s \in \{T_1, \dots, T_k\}$, and all $\mathbf{z} \in \mathbb{R}^d$ (except possibly regions of 0 mass):

$$\hat{\epsilon}_s(c_{dc}(\mathbf{z}; \theta)) = \mathbb{E}[\epsilon | \mathbf{z}_s = \mathbf{z}],$$

for $\epsilon$ and $\mathbf{z}_s$ such that $\mathbf{z}_s = e^{-T_s} \cdot \mathbf{z}_{T_1} + \sqrt{1 - e^{-2 \cdot T_s}} \cdot \epsilon$. Then, by Tweedie's identity (Miyasawa et al., 1961), we get

$$-\frac{1}{\sqrt{1 - e^{-2 \cdot T_s}}} \hat{\epsilon}_s(c_{dc}(\mathbf{z}; \theta)) = \nabla_{\mathbf{z}} \log \tilde{\mathbf{p}}_{T_s}(\mathbf{z}).$$

By using the definition of $\hat{\epsilon}_s(c_{dc}(\mathbf{z}; \theta))$ and Equation (11) from the proof of Proposition 4, we obtain:

$$\nabla_{\mathbf{z}} \log c_{dc}^{(s)}(\mathbf{z}; \theta) - \nabla_{\mathbf{z}} \log c_{dc}^{(k)}(\mathbf{z}; \theta) - \mathbf{z}$$
$$= \nabla_{\mathbf{z}} \log(\mathbb{P}(t = T_s | \mathbf{z} = \mathbf{z})) - \nabla_{\mathbf{z}} \log(\mathbb{P}(t = T_k | \mathbf{z} = \mathbf{z})) + \nabla_{\mathbf{z}} \log(\tilde{\mathbf{p}}_{T_k}(\mathbf{z} = \mathbf{z} | t = T_k)).$$

Since $\tilde{\mathbf{p}}_{T_k}$ is a standard Gaussian in $d$ dimensions, the gradient of its log density is $-\mathbf{z}$, leading to

$$\nabla_{\mathbf{z}} \log \frac{c_{dc}^{(s)}(\mathbf{z}; \theta)}{c_{dc}^{(k)}(\mathbf{z}; \theta)} = \nabla_{\mathbf{z}} \log \frac{\mathbb{P}(t = T_s | \mathbf{z} = \mathbf{z})}{\mathbb{P}(t = T_k | \mathbf{z} = \mathbf{z})} \tag{13}$$

$$\Leftrightarrow \frac{c_{dc}^{(s)}(\mathbf{z}; \theta)}{c_{dc}^{(k)}(\mathbf{z}; \theta)} = \frac{\mathbb{P}(t = T_s | \mathbf{z} = \mathbf{z})}{\mathbb{P}(t = T_k | \mathbf{z} = \mathbf{z})} \cdot C_s \tag{14}$$

$$\Leftrightarrow c_{dc}^{(s)}(\mathbf{z}; \theta) = \frac{\mathbb{P}(t = T_s | \mathbf{z} = \mathbf{z})}{\mathbb{P}(t = T_k | \mathbf{z} = \mathbf{z})} \cdot C_s \cdot c_{dc}^{(k)}(\mathbf{z}; \theta), \tag{15}$$

where $C_s \in \mathbb{R}$ is a constant, possibly different for each class $s$. Note that $C_k = 1$ by necessity. Furthermore, since $c_{dc}(\mathbf{z}; \theta)$ is normalised, we have

$$1 = \sum_{s=1}^{k} c_{dc}^{(s)}(\mathbf{z}; \theta)$$

$$\Leftrightarrow 1 = c_{dc}^{(k)}(\mathbf{z}; \theta) + \sum_{s \neq k} c_{dc}^{(s)}(\mathbf{z}; \theta)$$

$$\overset{(15)}{\Leftrightarrow} 1 = c_{dc}^{(k)}(\mathbf{z}; \theta) + \sum_{s \neq k} \Big\{ \frac{\mathbb{P}(t = T_s | \mathbf{z} = \mathbf{z})}{\mathbb{P}(t = T_k | \mathbf{z} = \mathbf{z})} \cdot C_s \cdot c_{dc}^{(k)}(\mathbf{z}; \theta) \Big\}$$

$$\Leftrightarrow c_{dc}^{(k)}(\mathbf{z}; \theta) = \frac{1}{1 + \sum_{s \neq k} \big\{ \frac{\mathbb{P}(t = T_s | \mathbf{z} = \mathbf{z})}{\mathbb{P}(t = T_k | \mathbf{z} = \mathbf{z})} \cdot C_s \big\}}.$$

By substituting the last equality into Equation (15), we obtain for any $s \in \{1, \ldots, k\}$:

$$c_{dc}^{(s)}(\boldsymbol{z}; \theta) = \frac{\mathbb{P}(t = T_s | \mathbf{z} = \boldsymbol{z}) \cdot C_s}{\mathbb{P}(t = T_k | \mathbf{z} = \boldsymbol{z}) + \sum_{i \neq k} \left\{ \mathbb{P}(t = T_i | \mathbf{z} = \boldsymbol{z}) \cdot C_i \right\}}$$

$$\overset{C_k=1}{\Leftrightarrow} c_{dc}^{(s)}(\boldsymbol{z}; \theta) = \frac{\mathbb{P}(t = T_s | \mathbf{z} = \boldsymbol{z}) \cdot C_s}{\sum_{i=1}^{k} \left\{ \mathbb{P}(t = T_i | \mathbf{z} = \boldsymbol{z}) \cdot C_i \right\}}.$$

Thus the set of minimisers for $\mathcal{L}_{MSE}(\theta)$ can be expressed as

$$\left\{ \theta \; : \; c_{dc}^{(s)}(\boldsymbol{z}; \theta) = \frac{\mathbb{P}(t = T_s | \mathbf{z} = \boldsymbol{z}) \cdot C_s}{\sum_{i=1}^{k} \left\{ \mathbb{P}(t = T_i | \mathbf{z} = \boldsymbol{z}) \cdot C_i \right\}} \right\} := \Theta_{MSE}.$$

To conclude, note that

- (a) the set $\Theta_{CE}$ is a subset of $\Theta_{MSE}$, as setting $C_i = 1 \, \forall i$ in $\Theta_{MSE}$ recovers $\Theta_{CE}$,

- (b) parameters $\theta^*$ for which $c_{dc}(\boldsymbol{z}; \theta^*) = \left( \mathbb{P}(t = T_1 | \mathbf{z} = \boldsymbol{z}), \ldots, \mathbb{P}(t = T_k | \mathbf{z} = \boldsymbol{z}) \right)$ is the definition of both $\Theta^*$ and $\Theta_{CE}$.

By (a), and since $\alpha > 0$, only solution of $\Theta_{CE}$ can minimise $\mathcal{L}_{c_{dc}}(\theta)$. By (b), this is equal to $\Theta^*$. $\square$

### A.5 PROPOSITION 9, PROPOSITION 10, PROPOSITION 11

Here, we show the analogous properties from above for the case of the correlated OU process defined in Eq. 3.

**Proposition 9** (Convergence to Gaussian copula). *For copula samples $\mathbf{z}_0$ on the Gaussian scale, consider $\mathbf{u}_t^i = \Phi(\mathbf{z}_t^i)$ with $\mathbf{z}_t^i$ as defined through the process in Equation (3). Then, this process maintains uniform marginal distributions at all times:*

$$\mathbf{u}_t^i \sim U(0, 1), \quad \forall t \geq 0, i \in \{1, \ldots, d\}.$$

*Moreover, denote by $c_t$ the copula of $\mathbf{z}_t$ under Equation (3). Then, $c_t$ converges to a Gaussian copula with correlation matrix $\Sigma$ as $t \to \infty$ in the Kullback-Leibler divergence with rate $\mathcal{O}(e^{-2t})$.*

The proof is similar to that of Proposition 2, and follows the same steps.

*Proof.* To show uniformity of $\mathbf{u}_t$ at all times, decompose $\mathbf{z}$ as follows:

$$\mathbf{z}_t = e^{-t} \cdot \mathbf{z}_0 + \sqrt{1 - e^{-2t}} \cdot \epsilon, \quad \epsilon \sim \mathcal{N}(\mathbf{0}, \Sigma).$$

Since the distribution of a single dimension $z_t^i$ does not depend on the correlation matrix $\Sigma$, we can identify it as a standard Gaussian, meaning $\mathbf{u}_t$ is marginally uniform.

For the convergence rate, similar existing results establish the limiting distribution of Equation (3) to be a correlated Gaussian $\mathcal{N}(\mathbf{0}, \Sigma)$ with rate $\mathcal{O}(Ce^{-2t})$. The same reasoning as in the previous proof applies, meaning we can derive the appropriate rate in KL divergence due to reparametrisation of $u_t^i = \Phi(z_t^i)$. $\square$

**Proposition 10** ($c_{dc}$ copula density evaluation under Eq. (3)). *For the classification task as defined in Eq. (4) following the diffusion process of Eq. (3) with correlation $\Sigma$, the true copula density $c(\boldsymbol{u})$ at a given value $\boldsymbol{u} \in [0, 1]^d$ is equal to the following probability ratio:*

$$c(\boldsymbol{u}) = \frac{\mathbb{P}(t = T_1 | \mathbf{z} = \bar{\Phi}^{-1}(\boldsymbol{u}))}{\mathbb{P}(t = T_k | \mathbf{z} = \bar{\Phi}^{-1}(\boldsymbol{u}))} \cdot \frac{\mathcal{N}(\bar{\Phi}^{-1}(\boldsymbol{u}), \Sigma)}{\mathcal{N}(\bar{\Phi}^{-1}(\boldsymbol{u}), \mathbf{I}_d)},$$

*where $\bar{\Phi}^{-1}(\boldsymbol{u})$ is the inverse standard Gaussian CDF applied dimensions-wise to $\boldsymbol{u}$.*

*Proof.* The proof is identical to the steps in Proposition 3 with the exception that the terminal density is $\mathcal{N}(\bar{\Phi}^{-1}(\boldsymbol{u}), \Sigma)$. Thus the optimal classifier recovers the ratio:

$$\frac{\mathbb{P}(t = T_1|\mathbf{z} = \bar{\Phi}^{-1}(\boldsymbol{u}))}{\mathbb{P}(t = T_k|\mathbf{z} = \bar{\Phi}^{-1}(\boldsymbol{u}))} = \frac{\tilde{p}_{T_1}(\bar{\Phi}^{-1}(\boldsymbol{u}))}{\mathcal{N}(\bar{\Phi}^{-1}(\boldsymbol{u}), \Sigma)}.$$

We obtain the copula density by multiplying the ratio by a correction term

$$\frac{\tilde{p}_{T_1}(\bar{\Phi}^{-1}(\boldsymbol{u}))}{\mathcal{N}(\bar{\Phi}^{-1}(\boldsymbol{u}), \Sigma)} \cdot \frac{\mathcal{N}(\bar{\Phi}^{-1}(\boldsymbol{u}), \Sigma)}{\mathcal{N}(\bar{\Phi}^{-1}(\boldsymbol{u}), \mathbf{I}_d)} = c(\boldsymbol{u}),$$

which concludes the proof. $\qquad\square$

**Proposition 11** ($c_{dc}$ copula score-based sampling under Eq. (3)). *Consider the classification task as defined in Eq. (4) for the process of Eq. (3), and let $c_s$ be the copula of $\mathbf{u}_s = \bar{\Phi}(\mathbf{z}_s)$ with $\bar{\Phi}$ the standard Gaussian CDF applied element-wise, and $\mathbf{z}_s$ defined by the process of Eq. (3) at time $s \in \{T_1, \ldots, T_k\}$. Then, the true copula score $\nabla_{\boldsymbol{u}} \log c_s(\boldsymbol{u})$ is given by:*

$$\nabla_{\boldsymbol{u}} \log c_s(\boldsymbol{u}) = w(\boldsymbol{u}) \odot \left[ \nabla_{\boldsymbol{z}} \log(\mathbb{P}(t = T_s|\mathbf{z} = \boldsymbol{z})) - \nabla_{\boldsymbol{z}} \log(\mathbb{P}(t = T_k|\mathbf{z} = \boldsymbol{z})) + (\mathbf{I}_d - \Sigma^{-1}) \cdot \boldsymbol{z} \right]_{\boldsymbol{z} = \bar{\Phi}^{-1}(\boldsymbol{u})}$$

*where $\odot$ is element-wise multiplication and $w(\boldsymbol{u}) := \left( \frac{1}{\phi(\Phi^{-1}(u^1))}, \ldots, \frac{1}{\phi(\Phi^{-1}(u^d))} \right)$.*

The proof mirrors that of Proposition 4, with the exception that we need to account for the correlation matrix $\Sigma$ and the different ratio target, as in Proposition 10. In fact, when $\Sigma = \mathbf{I}_d$, we exactly recover Proposition 4.

*Proof.* We define the mixture of likelihood $\tilde{p}_+(\mathbf{z}) := \sum_{s=1}^k \pi_s \cdot \tilde{p}(\boldsymbol{z})$ with a priori equal class weights $\pi_s \ s \in \{1, \ldots, k\}$. We can again rewrite the following two conditional probabilities as:

$$\tilde{p}_s(\mathbf{z} = \boldsymbol{z}|t = T_s) = \frac{\mathbb{P}(t = T_s|\mathbf{z} = \boldsymbol{z}) \cdot \tilde{p}_+(\boldsymbol{z})}{\pi_s} \tag{16}$$

and

$$\tilde{p}_{T_k}(\mathbf{z} = \boldsymbol{z}|t = T_k) = \frac{\mathbb{P}(t = T_k|\mathbf{z} = \boldsymbol{z}) \cdot \tilde{p}_+(\boldsymbol{z})}{\pi_{T_k}}. \tag{17}$$

Next, replace $\tilde{p}_+(\boldsymbol{z})$ in Equation (16) with the appropriate quantity from Equation (17):

$$\tilde{p}_s(\mathbf{z} = \boldsymbol{z}|t = T_s) = \frac{\pi_{T_k}}{\pi_s} \cdot \frac{\mathbb{P}(t = T_s|\mathbf{z} = \boldsymbol{z})}{\mathbb{P}(t = T_k|\mathbf{z} = \boldsymbol{z})} \cdot \tilde{p}_{T_k}(\mathbf{z} = \boldsymbol{z}|t = T_k)$$

where the class probabilities cancel due to equal prior odds. We take the log and then the gradient with respect to $\boldsymbol{z}$ on both sides (where densities are positive and differentiable), yielding:

$$\nabla_{\boldsymbol{z}} \log(\tilde{p}_s(\mathbf{z} = \boldsymbol{z}|t = T_s)) = \tag{18}$$
$$\nabla_{\boldsymbol{z}} \log(\mathbb{P}(t = T_s|\mathbf{z} = \boldsymbol{z})) - \nabla_{\boldsymbol{z}} \log(\mathbb{P}(t = T_k|\mathbf{z} = \boldsymbol{z})) + \nabla_{\boldsymbol{z}} \log(\tilde{p}_{T_k}(\mathbf{z} = \boldsymbol{z}|t = T_k)) \tag{19}$$

Here, we note that a copula decomposition following Sklar's theorem leads to the following expression of the log scores of a copula $c$ and joint density $\boldsymbol{p}$ with marginals $p^i, i \in \{1, \ldots, d\}$:

$$\nabla_{\boldsymbol{z}} \log c(P^1(z^1), \ldots, P^d(z^d)) = \nabla_{\boldsymbol{z}} \log \boldsymbol{p}(\boldsymbol{z}) - \nabla_{\boldsymbol{z}} \sum_{i=1}^d \log p^i(z^i).$$

In our case, for the joint density on the Gaussian scale $\tilde{p}_s(\mathbf{z} = \boldsymbol{z}|t = T_s)$, its marginal densities are standard Gaussian by construction as a consequence of the process in Equation (3). Thus, their marginal scores are $-z_i$, resulting in the following expression for the score of the log copula $c_s$ corresponding to the inter-variable dependence at time $s$:

$$\nabla_{\boldsymbol{z}} \log c_s(\Phi(z^1), \ldots, \Phi(z^d)) = \nabla_{\boldsymbol{z}} \log \tilde{p}_s(\mathbf{z} = \boldsymbol{z}|t = T_s) + \boldsymbol{z}. \tag{20}$$

We can now replace the score of the joint log density in Equation (20) with the expression in Equation (19), obtaining:

$$\nabla_{\mathbf{z}} \log c_s(\Phi(z^1), \ldots, \Phi(z^d)) =$$
$$\nabla_{\mathbf{z}} \log(\mathbb{P}(t = T_s | \mathbf{z} = \mathbf{z})) - \nabla_{\mathbf{z}} \log(\mathbb{P}(t = T_k | \mathbf{z} = \mathbf{z})) + (\mathbf{I}_d - \Sigma^{-1}) \cdot \mathbf{z}$$

Where we used that $\tilde{\boldsymbol{p}}_{T_k} = \mathcal{N}(\mathbf{0}, \Sigma)$ by construction, so its gradient is $-\Sigma^{-1} \cdot \mathbf{z}$. Finally, we use a change of variable from $\mathbf{z}$ to $\boldsymbol{u} = \bar{\Phi}(\mathbf{z})$ (the standard Gaussian CDF applied dimensions-wise to $\boldsymbol{u}$), obtaining the final expression:

$$\nabla_{\boldsymbol{u}} \log c_s(\boldsymbol{u}) = w(\boldsymbol{u}) \odot \left[ \nabla_{\mathbf{z}} \log(\mathbb{P}(t = T_s | \mathbf{z} = \mathbf{z})) - \nabla_{\mathbf{z}} \log(\mathbb{P}(t = T_k | \mathbf{z} = \mathbf{z})) + (\mathbf{I}_d - \Sigma^{-1}) \cdot \mathbf{z} \right]_{\mathbf{z} = \bar{\Phi}^{-1}(\boldsymbol{u})}$$

where $\odot$ is element-wise multiplication and $w(\boldsymbol{u})$ is defined as:

$$w(\boldsymbol{u}) := \Big( \frac{1}{\phi(\Phi^{-1}(u_1))}, \ldots, \frac{1}{\phi(\Phi^{-1}(u_d))} \Big).$$

$\square$

### A.6 PROPOSITION 7

**Proposition 7.** *For any initial point* $\mathbf{u}_0 \in [0,1]^d$ *with* $\mathbf{v}_0 \sim \mathcal{N}(\mathbf{0}, \mathbf{I_d})$, *the reflected point* $\mathbf{u}_t := \mathcal{R}(\mathbf{u}_0 + t \cdot \mathbf{v}_0)$ *converges in distribution to the independent copula on the hypercube as* $t \to \infty$. *Further, if* $\mathbf{u}_0$ *follows a copula distribution, then* $\mathbf{u}_t$ *also follows some copula distribution* $c_t$ *with uniform marginals for any time* $t > 0$.

*Proof.* We first prove convergence to the independence copula and then show that $\mathbf{u}_t$ follows some copula $c_t$ at any time $t > 0$.

**Convergence to the independence copula.** As $\mathbf{v}_0$ is independent across dimensions, we initially only focus on the one-dimensional case of $\mathbf{u}_t^i = \mathcal{R}_1(\mathbf{u}_0^i + t \cdot \mathbf{v}_0^i)$, for $i \in \{1, \ldots, d\}$, with the reflection operator in one dimension on the $u^i$ component only being

$$\mathcal{R}_1(x) = \begin{cases} x - \lfloor x \rfloor, & \text{if } \lfloor x \rfloor \text{ is even}, \\ 1 - (x - \lfloor x \rfloor), & \text{if } \lfloor x \rfloor \text{ is odd}, \end{cases} \quad x \in \mathbb{R}.$$

Notice that $\mathcal{R}_1$ is a 2-periodic function. For a bounded and continuous function $f$, we have that $g(x) := f(\mathcal{R}_1(x))$ is bounded, measurable and 2-periodic. Then, with $\phi(z)$ as the one-dimensional standard Gaussian density, from definitions

$$\mathbb{E}[f(\mathbf{u}_t^i)] = \mathbb{E}[f(\mathcal{R}_1(\mathbf{u}_0^i + t \cdot \mathbf{v}_0^i))] = \mathbb{E}[g(\mathbf{u}_0^i + t \cdot \mathbf{v}_0^i)]$$
$$= \int_{-\infty}^{\infty} g(\mathbf{u}_0^i + t \cdot z) \cdot \phi(z) dz.$$

Next, we use Fejér's Theorem:

**Theorem 12** (Fejér (1910)). *Let* $g \colon \mathbb{R} \to \mathbb{R}$ *be bounded, measurable, and $k$-periodic, and let $\varphi$ be Lebesgue integrable. Then for any sequence $\sigma_n \to +\infty$ and any real constants $\alpha_n$,*

$$\lim_{n \to \infty} \int_{-\infty}^{\infty} g(\sigma_n x + \alpha_n) \cdot \varphi(x) \, dx = \frac{1}{k} \cdot \left( \int_0^k g(y) \, dy \right) \cdot \left( \int_{-\infty}^{\infty} \varphi(x) \, dx \right).$$

In our case, since $g$ is 2-periodic, and $\phi$ is integrable, we have:

$$\lim_{t \to \infty} \int_{-\infty}^{\infty} g(\mathbf{u}_0^i + tz) \cdot \phi(z) \, dz = \frac{1}{2} \cdot \int_0^2 g(y) \, dy = \frac{1}{2} \cdot \int_0^1 g(y) \, dy + \frac{1}{2} \cdot \int_1^2 g(y) \, dy$$

where we split the right-hand side based on $\lfloor y \rfloor$ being even or odd, over $[0,1]$ and $[1,2]$ respectively. For the first part, using the definition of $\mathcal{R}_1$, we identify the expectation of a uniform variable

$$\frac{1}{2} \cdot \int_0^1 g(y) \, dy = \frac{1}{2} \cdot \int_0^1 f(y) \, dy = \frac{1}{2} \cdot \mathbb{E}[f(U)] \quad \text{with } U \sim \mathcal{U}[0,1].$$

For the second part, we can achieve the same with a substitution $z = 2 - y$:

$$\frac{1}{2} \cdot \int_1^2 g(y)\, dy = \frac{1}{2} \cdot \int_1^2 f(2-y)\, dy = \frac{1}{2} \cdot \int_0^1 f(z)\, dz = \frac{1}{2} \cdot \mathbb{E}[f(U)] \quad \text{with } U \sim \mathcal{U}[0,1].$$

Thus,

$$\lim_{t \to \infty} \mathbb{E}[f(\mathsf{u}_t^i)] = \mathbb{E}[f(U)], \quad \text{with } U \sim \mathcal{U}[0,1].$$

which shows that $\mathsf{u}_t^i$ converges in distribution to the uniform density. Finally, to conclude, we note that since velocities $\mathbf{v}_0$ are independent across dimensions, so is the distribution of $\mathbf{u}_t$. The limiting distribution of $\mathbf{u}_t$ is therefore the product of the limiting distributions of $\mathsf{u}_t^i$. This is the uniform distribution on $[0,1]^d$, the independent copula.

Marginal uniformity at all times. Consider the marginal variable $\mathsf{u}_t^i$ obtained from the reflection $\mathcal{R}_1(\mathsf{u}_t^i + t \cdot \mathsf{v}_t^i, \mathsf{v}_t^i)$ for any $i \in \{1, \ldots, d\}$. Note that $\mathcal{R}_1$ is measurable as it is piecewise continuous. It is also a combination of measure-preserving operations, namely translations, and reflections across the vertical axis, meaning $\mathcal{R}_1$ is also measure-preserving. This is equivalent to having, for any bounded measurable $f$:

$$\mathbb{E}[f(\mathcal{R}_1(\mathsf{u}_0^i + t \cdot \mathsf{v}_0^i, \mathsf{v}_0^i))] = \mathbb{E}[f(\mathsf{u}_0^i)].$$

By this reasoning, assuming $\mathbf{u}_0$ follows a copula distribution, every marginal remains uniform for $t > 0$, and so $\mathbf{u}_t$ also defines a valid copula distribution at any $t > 0$. $\qquad\square$

## A.7 PROPOSITION 8

**Proposition 8** *Consider the reflection process introduced in Eq. (6) applied to copula samples $\mathbf{u}_0 \sim c(\mathbf{u})$ with velocities $\mathbf{v}_0 \sim \mathcal{N}(\mathbf{0}, \mathbf{I}_d)$. Let $v^*(\mathbf{u}, t) = \mathbb{E}[\mathbf{v}_t | \mathbf{u}_t = \mathbf{u}]$ be the expected velocity at time $t \geq 0$ and location $\mathbf{u} \in [0,1]^d$, and denote by $c_T$ the copula of samples $\mathbf{u}_T$ at time $T$. Then, starting from a value $\mathbf{u}_T \sim c_T(\mathbf{u}_T)$ and following the probability path given by the ordinary differential equation*

$$\frac{d}{dt} \mathbf{u}_t = v^*(\mathbf{u}, t)$$

*backwards from $T \to 0$, the destination point comes from the initial copula distribution $\mathbf{u}_0 \sim c(\mathbf{u})$.*

Our proof largely follows Proposition 3 in Holderrieth et al. (2024), and we include below a derivation for our reflection setup for completeness.

*Proof.* We first describe our reflection process as an ordinary differential equation (ODE) with initial distribution of samples $\mathbf{u}_0 \sim c(\mathbf{u})$, and initial velocity distribution as $\mathbf{v}_0 \sim \mathcal{N}(\mathbf{0}, \mathbf{I}_d)$. As both components are independent at time 0, our initial joint probability of being at a location-velocity pair is $(\mathbf{u}_0, \mathbf{v}_0) \sim \Pi_0(\mathbf{u}_0, \mathbf{v}_0) = c(\mathbf{u}_0) \cdot \mathcal{N}(\mathbf{v}_0; \mathbf{0}, \mathbf{I}_d)$. Therefore, the ODE governing the distribution $\Pi_t(\mathbf{u}_t, \mathbf{v}_t)$ on the interior $(0,1)^d$ through time is defined as:

$$\frac{d}{dt} \mathbf{u}_t = \mathbf{v}_t \quad \text{and} \quad \frac{d}{dt} \mathbf{v}_t = 0,$$

with a Neumann boundary condition for reflection, where for $\partial\Omega$ the boundary of the hypercube, and $\mathbf{n}(\mathbf{u})$ the outward unit normal vector at a point $\mathbf{u} \in \partial\Omega$, we have for all $\mathbf{u} \in \partial\Omega$ and $t \geq 0$:

$$\int \mathbf{v} \Pi_t(\mathbf{u}, \mathbf{v}) d\mathbf{v} \cdot \mathbf{n}(\mathbf{u}) = 0. \tag{21}$$

We will work with the distribution of being at location $\mathbf{u}$ at time $t$, given by $c_t(\mathbf{u}) := \int \Pi_t(\mathbf{u}, \mathbf{v}) d\mathbf{v}$, which has been shown to correspond to a copula for all $t \geq 0$ in Proposition 7. For our ODE, the change of the density $\Pi_t(\mathbf{u}, \mathbf{v})$ with $t$ is described by the Fokker-Planck equation:

$$\frac{\partial \Pi_t(\mathbf{u}, \mathbf{v})}{\partial t} = -\nabla_{\mathbf{u}, \mathbf{v}} \big( (\mathbf{v}, 0) \cdot \Pi_t(\mathbf{u}, \mathbf{v}) \big).$$

By integrating both sides in $v$, we obtain

$$\frac{\partial c_t(\boldsymbol{u})}{\partial t} = \int -\nabla_{\boldsymbol{u},\boldsymbol{v}}\big((\boldsymbol{v},0)\cdot\Pi_t(\boldsymbol{u},\boldsymbol{v})\big)d\boldsymbol{v} \tag{22}$$

$$\Leftrightarrow \frac{\partial c_t(\boldsymbol{u})}{\partial t} = -\nabla_{\boldsymbol{u}}\int\big(\boldsymbol{v}\Pi_t(\boldsymbol{u},\boldsymbol{v})\big)d\boldsymbol{v} - \underbrace{\int \nabla_{\boldsymbol{v}}0\cdot\Pi_t(\boldsymbol{u},\boldsymbol{v})\big)d\boldsymbol{v}}_{=0} \tag{23}$$

$$\Leftrightarrow \frac{\partial c_t(\boldsymbol{u})}{\partial t} = -\nabla_{\boldsymbol{u}}\big(c_t(\boldsymbol{u})\cdot\mathbb{E}[\mathbf{v}_t|\mathbf{u}_t=\boldsymbol{u}]\big) \tag{24}$$

When $\mathbf{u}_t \in \partial\Omega$, our boundary condition in Eq. (21) guarantees there is no loss of mass at the boundary as $\mathbb{E}[\mathbf{v}_t|\mathbf{u}_t = \boldsymbol{u}] \cdot \mathbf{n}(\boldsymbol{u}) = 0$. Therefore, Eq. (24) proves that the marginal continuity equation for our process is satisfied by $v^*(\boldsymbol{u},t) =: \mathbb{E}[\mathbf{v}_t|\mathbf{u}_t = \boldsymbol{u}]$. □

## B  EXPERIMENTAL DETAILS

**Pre-processing.**   As copulas have rarely been applied to the settings we aim to explore, we select datasets with simple marginal distributions but where the complexity resides in the dependence, the only exception being `Robocup`. For all datasets other than `Robocup`, we pre-process the data in $\mathbb{R}^d$ by applying the empirical CDF to each marginal, obtaining copula observations in $[0,1]^d$. For image datasets, we add a small Gaussian noise to dequantise it and obtain copula ranks. For 10 repeated runs, we split the copula observations into a train and test split according to Table 3, where splits for `digits` and `MNIST` were chosen by following Huk et al. (2025). We also give the terminal times $T$ at which the stochastic process has practically reached its stationary distribution, which we found to be quickly attained in experiments. We report more details on the time choices in Apdx. B.3.

Table 3: **Experimental setups:** For experiments from the main paper, we show the train and test percentages of the full dataset, as well as the terminal times used for convergence of the forward process.

|  | magic | Dry_Bean | Robocup | digits | MNIST | Cifar |
|---|---|---|---|---|---|---|
| Train/Test Split | 80/20% | 80/20% | 80/20% | 50/50% | 50/50% | 80/20% |
| $c_{dc}$ Terminal $T_k$ | 3 | 3 | 3 | 3 | 3 | 3 |
| Ref. Terminal $T$ | 1.5 | 1.5 | 1.5 | 1.5 | 2 | 2.5 |

**Pre-processing for `Robocup`.**   The `Robocup` dataset, first studied with copulas by Wang et al. (2021), consists of a $20-$dimensional time series containing the vertical and horizontal positions of a team of 10 robots. The dataset is obtained from simulations of robot football matches Michael et al. (2017), specifically from the 25 games between the two teams cyrus2017 and helios2017, where we model the cyrus2017 team. To model the movements of the robot team, we adopt the following decomposition of the multivariate time series. For $(\mathbf{x}_t^i, \mathbf{y}_t^i) \in \mathbb{R}\times\mathbb{R}$ the horizontal and vertical movement of robot $i$ at time $t$, and $(X_t, Y_t) \in \mathbb{R}^{10}\times\mathbb{R}^{10}$ the vector of aggregated horizontal and vertical movements of all 10 robots, we model the next position as:

$$\mathbf{x}_t^i = f^{(\mathbf{x}^i)}\big(X_{t-1},Y_{t-1},X_{t-2},Y_{t-2}\big) + \varepsilon^{\mathbf{x}^i}, \quad \mathbf{y}_t^i = f^{(\mathbf{y}^i)}\big(X_{t-1},Y_{t-1},X_{t-2},Y_{t-2}\big) + \varepsilon^{\mathbf{y}^i}$$

where the mean functions $f^{(\mathbf{x}^i)}, f^{(\mathbf{y}^i)}$ predict the expected next position based on the past two positions of the team, with noise coming from $\varepsilon^{\mathbf{x}^i}, \varepsilon^{\mathbf{y}^i}$, which we assume is not predictable as it is independent from the movements of the team. We model the mean functions as a fully connected network $f_\theta : \mathbb{R}^{40} \mapsto \mathbb{R}$, taking as input the 40-dimensional position of the team over the last two steps, with one hidden layer of dimension 32, using the ReLU activation function. We use the Adam optimiser (Kingma, 2014) with a 0.0001 learning rate, and optimise for 250000 steps, which takes 5 hours on a CPU. We train a separate model for each player's movement directions, resulting in 40 marginal mean models.

However, we assume that the noise vector $\varepsilon^{\mathbf{x}^1}, \varepsilon^{\mathbf{y}^1}, \ldots, \varepsilon^{\mathbf{x}^{10}}, \varepsilon^{\mathbf{y}^{10}}$ is dependent, as such, modelling it with a copula-based decomposition. We first estimate the marginal CDFs (using the empirical CDF)

$P^{x^i}, P^{y^i}$, of $\varepsilon^{x^i}, \varepsilon^{y^i}$ respectively, for $i = 1, \ldots, 10$, and use them to map the noise vector to the copula scale with $\mathbf{u} = \big(P^{x^1}(\varepsilon^{x^1}), P^{y^1}(\varepsilon^{y^1}), \ldots, (P^{x^{10}}(\varepsilon^{x^{10}}), P^{y^{10}}(\varepsilon^{y^{10}})\big)$. We then model $\mathbf{u}$ with our copulas.

This approach is a popular application of copulas to dependent time series analysis, such as in finance and climate science (Czado et al., 2022). The mean functions serve to absorb the non-stationarity of the time-series, leaving i.i.d. noise which can be modelled with a copula.

**Further application of copulas.** Another application could be synthetic data generation or data imputation. For instance, consider sensitive data such as criminal records about recidivism. The variables interact in complex ways with non-trivial dependencies, and obtaining more data is not feasible. Our copula models can then be used to model the data in a first step, and in a second step, impute missing values of incomplete records based on this dependence. First, samples are generated from the fitted copula model (condional on the observed variables), and second, these generated copula samples are transformed via marginal distributions to the data scale, filling in the missing data.

**Time discretisation.** As in our theoretical derivations, for the $c_{dc}$ we pick equal a priori class odds, meaning that during training we sample classes for times uniformly among the time discretisation. However, this discretisation does not need to be uniform in the time interval $[T_1, T_k]$. For the $c_{dc}$, on preliminary results from the training set, we found that a Kullback-Leibler discretisation worked best for the scientific datasets, while a linear optimal transport discretisation (Lipman et al., 2023) worked best for image data. Following our result on the convergence rate in Proposition 2, we define the KL discretisation as the time steps required for a constant change in the KL. That is, for $k$ the total number of timesteps, the $s^{th}$ time step is given by:

$$t_s = -\tfrac{1}{2} \ln\Big(1 - \big(1 - e^{-2T_k}\big) \tfrac{s}{k-1}\Big), \quad s = 0, 1, \ldots, k-1.$$

The linear optimal transport discretisation is simply a uniform time grid from 0 to $T_k$ with $N$ points,

$$t_s = \tfrac{s}{k-1} T_k, \quad s = 0, \ldots, k-1.$$

For the Reflection copula, we use the following power-law discretisation across all experiments:

$$t_s = T_k \cdot \Big(\frac{s}{k-1}\Big)^{0.125}.$$

We provide an ablation for the choice of scheduler. For the `Robocup` experiment, we train our $c_{dc}$ model on $90\%$ of the train set and use the last $10\%$ for validation. In Fig. 6, we report the LL for three different choices of schedulers, namely a linear optimal transport scheduler, a power-law scheduler and a KL scheduler. While the scheduler does not affect the LL of the train set, we show that the KL scheduler obtains better generalisations and more stable performance on the test set, while the other two schedulers quickly overfit.

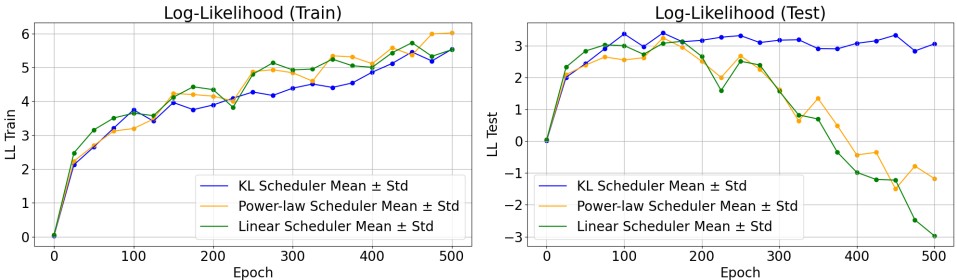

Figure 6: **Scheduler ablation:** For three different schedulers, we show the training and validation LL values through epochs. The KL scheduler is more stable and generalises better from the train to the test set.

**Practical implementation details.**    For all experiments, we first tune hyperparameters on the train set, finding that the $c_{dc}$ converges quickly, while the reflection copula benefits from more training.

For experiments on scientific datasets, we use a simple time-conditioned multi-layer perceptron architecture for the reflection copula, with 6 hidden layers of size 512 each. For the $c_{dc}$ on scientific data, we use a ResNet backbone with 6 hidden layers of size 512 each, followed by a classifier head to output 50 class probabilities with a softmax activation to normalise them. We use the Swish activation function (Ramachandran et al., 2017) for both models.

For image datasets, for both models, we use the DDPM architecture from Ho et al. (2020), again with an extra classifier head for the $c_{dc}$. We use the Adam optimiser for training (Kingma, 2014) with learning rate 0.00005 for the $c_{dc}$ and 0.0001 for the reflection copula. We train for 1K, 6K, 250, 250, 75K, 50K epochs for the $c_{dc}$, on datasets ordered as in Tab. 3. The batch size is 1024 for scientific datasets and 128 for image data with the $c_{dc}$. For the reflection copula, we train for 100K epochs on the scientific datasets, and for 500 epochs on image data, with a batch size of 512 for scientific data and 128 for images, iterating over the whole datasets at each epoch for images. We use mixture loss weights $\alpha = 0.05, 0.05, 0.15$ for the $c_{dc}$ on scientific datasets, and use 8 classes with $\alpha = 0.005$ for digits, and 256 with $\alpha = 0.005$ for the other image datasets.

For the $c_{dc}$ on digits and Dry_Bean, we use a correlated OU process, and we use an uncorrelated OU process on all other datasets. This is because their dependencies are concentrated on the diagonal when inspecting their pair plots.

For the implementation of benchmark methods, the ratio copula uses the same models as the $c_{dc}$ except for the MNIST dataset, where a better LL was obtained with a convolutional network model following Huk et al. (2025). We train the ratio copula for 3K, 25K, 3K, 1K, 2K, 5K epochs for datasets as ordered in Tab. 3. We sample the ratio copula using Hamiltonian Monte Carlo implemented in hamiltorch by Cobb & Jalaian (2021), and we tune the hyperparameters to have at least 60% acceptance rates. For the IGC, in scientific datasets, we found that the same multi-layer perceptron architecture with 6 hidden layers of size 512 each led to frequent mode collapse. We therefore use a smaller network size with only 2 hidden layers of size 512 for scientific datasets, and use the full 6 layers for image data. We train the IGC for 100K epochs. The Gaussian copula is fitted using the empirical covariance of the training data on the Gaussian scale. The vine copula is fitted using the pyvinecopulib package of Vatter & Nagler (2022) using non-parametric bivariate copulas.

**Algorithms for the $c_{dc}$.**    Here we give Alg. 1 for sampling, and Alg. 2 for optimising the $c_{dc}$ copula. Note that like most copula methods (vines, Gaussian copulas, and IGC), we sample on the Gaussian scale and transform samples back to the copula scale at the end. This slightly alters the form of the score from Prop. 4. We employ a DDPM-style sampler for scientific datasets, and use a DDIM-style sampler (see Song et al. (2020)) for image data, as it resulted in less noisy outputs. Below, we present the DDPM-style sampler for the $c_{dc}$ copula model, followed by the training algorithm.

**Algorithmic changes for $c_{dc}$ with correlated OU.**    Note that for the correlated OU process from Eq. 3, one has to compensate for the correlation matrix $\Sigma$. Concretely in Alg. 1, Line 2 becomes

$$z_{T_k} \sim \mathcal{N}(\mathbf{0}, \Sigma),$$

line 6 becomes

$$\nabla_{z_{T_t}} G(\bar{\Phi}(z_{T_t})) = \nabla_{z_{T_t}} \log \mathbb{P}(t = T_t | z_{T_t}) - \nabla_{z_{T_t}} \log \mathbb{P}(t = T_k | z_{T_t}),$$

line 7 becomes

$$z_{T_{t-1}} \leftarrow \frac{1}{\sqrt{\alpha_{T_t}}} \Big( \alpha_{T_t} \cdot z_{T_t} + (1 - \alpha_{T_t}) \cdot \Sigma \cdot \nabla_{z_{T_t}} G(\bar{\Phi}(z_{T_t})) \Big),$$

and finally line 8 becomes

$$z_{T_t} \leftarrow z_{T_t} + \mathbf{H} \cdot \epsilon \cdot \sqrt{1 - \alpha_t}$$

with $\epsilon \sim \mathcal{N}(\mathbf{0}, \mathbf{I}_d)$ for $\mathbf{H}$ the lower triangular matrix from a Cholesky decomposition of $\Sigma$.

For Alg. 2, the only changes are line 5 becoming

$$z_s = e^{-s} z_{T_1} + \sqrt{1 - e^{-2s}} \epsilon \quad \text{for } \epsilon \sim \mathcal{N}(\mathbf{0}, \Sigma),$$

and line 9 becoming

$$\hat{\epsilon}_s = \sqrt{1 - e^{-2s}} \cdot (\Sigma \cdot (\nabla_{\boldsymbol{z}_s} \log \mathbb{P}(t = T_k|\boldsymbol{z}_s) - \nabla_{\boldsymbol{z}_s} \log \mathbb{P}(t = T_t|\boldsymbol{z}_s)) + \boldsymbol{z}_s).$$

---

**Algorithm 1** Sampling algorithm for the $c_{dc}$ copula model.

1: Initialize diffusion timesteps $T_1, \ldots, T_k$
    with class probability model $c_{dc}(\boldsymbol{z}) = \big(\mathbb{P}(t = T_1|\boldsymbol{z}), \ldots, \mathbb{P}(t = T_k|\boldsymbol{z})\big)$.
2: Sample $\boldsymbol{z}_{T_k} \sim \mathcal{N}(\boldsymbol{0}, \mathbf{I}_d)$
3: **for** $t = k$ **to** $2$ **do**
4:    Pre-compute constants for OU process: $\alpha_t = \exp(2(T_{t-1} - T_t))$,
5:    Compute copula using Prop. 3: $c_t(\bar{\Phi}(\boldsymbol{z}_{T_t})) = \mathbb{P}(t = T_t|\boldsymbol{z}_{T_t})/\mathbb{P}(t = T_k|\boldsymbol{z}_{T_t})$
6:    Compute copula score using Prop. 4 on the Gaussian scale:

$$\nabla_{\boldsymbol{z}_{T_t}} \log c_{T_t}(\bar{\Phi}(\boldsymbol{z}_{T_t})) = \nabla_{\boldsymbol{z}_{T_t}} \log \mathbb{P}(t = T_t|\boldsymbol{z}_{T_t}) - \nabla_{\boldsymbol{z}_{T_t}} \log \mathbb{P}(t = T_k|\boldsymbol{z}_{T_t})$$

7:    DDPM update to Gaussian scale sample:

$$\boldsymbol{z}_{T_{t-1}} \leftarrow \frac{1}{\sqrt{\alpha_{T_t}}} \Big( \alpha_{T_t} \cdot \boldsymbol{z}_{T_t} + (1 - \alpha_{T_t}) \cdot \nabla_{\boldsymbol{z}_{T_t}} \log c_{T_t}(\bar{\Phi}(\boldsymbol{z}_{T_t})) \Big)$$

8:    Add exploration noise: $\boldsymbol{z}_{T_{t-1}} \leftarrow \boldsymbol{z}_{T_{t-1}} + \epsilon \cdot \sqrt{1 - \alpha_t}$ with $\epsilon \sim \mathcal{N}(\boldsymbol{0}, \mathbf{I}_d)$
9: **end for**
10: Map final variables back to copula space: $\boldsymbol{u}_0 = \bar{\Phi}(\boldsymbol{z}_{T_1})$
11: Return $\boldsymbol{u}_0 \sim c(\boldsymbol{u})$

---

**Algorithm 2** Training the $c_{dc}$ copula model.

1: Initialize model parameters $\theta$, number of epochs $N_{\text{epochs}}$, weight $\alpha > 0$,
    timestep $[T_1, \ldots, T_k]$
2: **for** epoch = 1 **to** $N_{\text{epochs}}$ **do**
3:    Sample Gaussian scale dependent data $\boldsymbol{z}_{T_1} \sim \tilde{\boldsymbol{p}}_{T_1}$
4:    Sample diffusion time uniformly $s \sim U[T_1, \ldots, T_k]$
5:    Compute perturbed data: $\boldsymbol{z}_s = e^{-s}\boldsymbol{z}_{T_1} + \sqrt{1 - e^{-2s}}\epsilon$    for $\epsilon \sim \mathcal{N}(\boldsymbol{0}, \mathbf{I}_d)$
6:    Compute class probabilities: $c_{dc}(\boldsymbol{z}_s; \theta) = \big(\mathbb{P}(t = T_1|\boldsymbol{z}_s), \ldots, \mathbb{P}(t = T_k|\boldsymbol{z}_s)\big)$
7:    Compute cross-entropy loss for class probabilities: $\mathcal{L}_{\text{CE}} = -\log \mathbb{P}(t = s|\boldsymbol{z}_s)$
8:    Compute score-based noise estimate:

$$\hat{\epsilon}_s = \sqrt{1 - e^{-2s}} \cdot (\nabla_{\boldsymbol{z}_s} \log \mathbb{P}(t = T_k|\boldsymbol{z}_s) - \nabla_{\boldsymbol{z}_s} \log \mathbb{P}(t = T_s|\boldsymbol{z}_s) + \boldsymbol{z}_s)$$

9:    Compute mean squared error loss for added noise: $\mathcal{L}_{\text{MSE}} = \|\hat{\epsilon}_s^{(i)} - \epsilon^{(i)}\|^2$
10:   Take gradient step with respect to $\theta$ on combined loss: $\mathcal{L}_{c_{dc}} = \alpha \cdot \mathcal{L}_{\text{CE}} + \mathcal{L}_{\text{MSE}}$
11: **end for**
12: Return trained model $c_{dc}(\boldsymbol{z}; \theta)$

---

**Algorithms for the reflection copula.** We now present Alg. 3 to sample, and Alg. 4 to train a reflection copula. Our sampling algorithm is similar to an Euler–Maruyama scheme for numerical integration. Even though we reflect samples even during sampling, in practice, the velocity predictor generally does not point outside the hypercube. Our training algorithm is simply a mean square error minimisation for velocity predictions.

---

**Algorithm 3** Sampling algorithm for the reflection copula model.

1: Initialize discretisation timesteps $T_1, \ldots, T_k$ with velocity predictor $v(\boldsymbol{u}, t)$.
2: Sample $\boldsymbol{u}_{T_k} \sim U[0,1]^d$
3: **for** $t = k$ **to** 2 **do**
4:     Predict velocity at time $t$: $\hat{v}_{T_t} \leftarrow v(\boldsymbol{u}_{T_t}, T_t)$
5:     Update copula sample: $\boldsymbol{u}_{T_{t-1}} \leftarrow \boldsymbol{u}_{T_t} + (T_{t-1} - T_t) \cdot \hat{v}_{T_t}$
6:     Apply reflection to stay within $[0,1]^d$ using Eq. 6, for $i = 1, \ldots, d$:

$$\boldsymbol{u}^i_{T_{t-1}} \leftarrow \begin{cases} \boldsymbol{u}^i_{T_{t-1}} - \lfloor \boldsymbol{u}^i_{T_{t-1}} \rfloor & \text{if } \lfloor \boldsymbol{u}^i_{T_{t-1}} \rfloor \text{ is even} \\ 1 - \boldsymbol{u}^i_{T_{t-1}} + \lfloor \boldsymbol{u}^i_{T_{t-1}} \rfloor & \text{else} \end{cases}$$

7: **end for**
8: Return $\boldsymbol{u}_0 \sim c(\boldsymbol{u})$

---

**Algorithm 4** Training algorithm for the reflection copula model.

1: Initialize model parameters $\theta$, number of epochs $N_{\text{epochs}}$, terminal time $T$
2: **for** epoch = 1 **to** $N_{\text{epochs}}$ **do**
3:     Sample copula data $\boldsymbol{u}_{T_1} \sim c(\boldsymbol{u})$
4:     Sample velocity $\boldsymbol{v} \sim \mathcal{N}(\mathbf{0}, \mathbf{I}_d)$
5:     Sample forward time $s$ uniformly as $s = T \cdot u^4$, for $u \sim U[0,1]$
6:     Compute reflected location and velocity using Eq. 5, for $i = 1, \ldots, d$:

$$(\boldsymbol{u}^i_s, \boldsymbol{v}^i_s) = \mathcal{R}(\boldsymbol{u}^i_{T_1} + s \cdot \boldsymbol{v}^i_{T_1}, \boldsymbol{v}^i_{T_1})$$

7:     Predict velocity:

$$\hat{v}_s \leftarrow v(\boldsymbol{u}_s, s)$$

8:     Take gradient step with respect to $\theta$ on mean squared velocity error:

$$\mathcal{L}_{\text{MSE}} = ||\hat{v}_s - \boldsymbol{v}_s||^2$$

9: **end for**
10: Return trained reflection copula model $v(\boldsymbol{u}, t; \theta)$

---

## B.1 COMPUTATIONAL TIMES FOR COPULA MODELS

Our copulas have different design goals; the $c_{dc}$ performs density estimation with a single function evaluation, while the Reflection copula specialises in sampling. While both designs use an iterative sampling procedure (see Algs. 1, 3), each iteration of the reflection copula is a simple function evaluation of the network, while the $c_{dc}$ is a function evaluation followed by a gradient computation of the output with respect to the input. As such, the Reflection copula is preferred for applications heavily dependent on fast sampling, while the $c_{dc}$ shines in LL-dependent applications. In Tab. 4, we provide the training times and sampling times for producing 1000 samples, all done on an NVIDIA L40S GPU except for the Vine copulas which used an AMD EPYC 9555P processor. The $c_{dc}$ uses a number of sampling steps equal to the number of classes, while the reflection copula uses 50 steps for all experiments. We do not provide times for the Gaussian copula as training is simply taking the empirical covariance of the data on the Gaussian scale, and sampling is just a sample from that Gaussian with a probability integral transform. This estimation simplicity comes at the cost of low modelling capacity.

The vine copula training time scales mostly with data size and dimension, which explains the higher time of MNIST compared to Cifar. As vines are inherently sequential and cannot take advantage

of GPU compute, our methods obtain faster sampling times compared to vines. The IGC has fast training and the fastest sampling times, which are partially due to the smaller network size we used as a consequence of the frequent model collapse with larger network sizes. The Ratio copula has a faster training time than our methods, but requires much longer to produce samples due to the reliance on Hamiltonian Monte Carlo (differences in sampling times are explained by differences in the HMC sampler on different datasets to yield a 60% acceptance rate).

Table 4: **Training and sampling times:** For all methods, we measure the time to train and sample. Compared to the $c_{dc}$, the reflection copula trains for longer but samples much faster.

| | magic | Dry_Bean | Robocup | digits | MNIST | Cifar |
|---|---|---|---|---|---|---|
| Vine training | 1.5s | 1.1s | 13s | 2s | 2h | 21min |
| IGC training | 2s | 3s | 9s | 11s | 12min | 14min |
| Ratio training | 7s | 62s | 7s | 56s | 23min | 37min |
| $c_{dc}$ training | 6min | 7min | $4\frac{1}{2}$min | 100s | 2h | 4h |
| Reflection training | 33min | 40min | 46min | 90s | 40m | 1h |
| Vine sampling | 0.09s | 0.17s | 0.23s | 0.77s | 78s | 131s |
| IGC sampling | $1.2 * 10^{-5}$s | $1.5 * 10^{-5}$s | $1.7 * 10^{-5}$s | 0.0008s | 0.0013s | 0.0015s |
| Ratio sampling | 165s | 327s | 151s | 554s | 2h | $2\frac{1}{2}$h |
| $c_{dc}$ sampling | 0.07s | 0.08s | 0.08s | 2s | 42s | 67s |
| Reflection sampling | 0.01s | 0.02s | 0.02s | 0.5s | 8s | 15s |

We additionally perform a dedicated computational time study in Fig. 7. From our computational study, we conclude that the network model is the most influential, followed by the number of timesteps used to simulate the SDE/ODE when sampling.

In more detail, for $d$ the data dimension , $w$ the network width and $h$ the number of hidden layers, $c_{dc}$ training has a complexity of $\mathcal{O}(d \cdot w + h \cdot w^2)$. This is because the model performs one forward pass (for class probabilities) and one backward pass (for scores) before taking gradients with respect to the loss during training. During inference, to sample the $c_{dc}$, we perform a forward and backward pass for each of the $k$ classes, following a SDE discretization scheme, resulting in a complexity of $\mathcal{O}(k \cdot (d \cdot w + h \cdot w^2))$. For the reflection copula, one training epoch consists of a single forward pass followed by a weight update, with complexity $\mathcal{O}(d \cdot w + h \cdot w^2)$. Generating samples requires $m$ network evaluations where $m$ is the number of steps to simulate the ODE for, which scales as $\mathcal{O}(m \cdot (d \cdot w + h \cdot w^2))$.

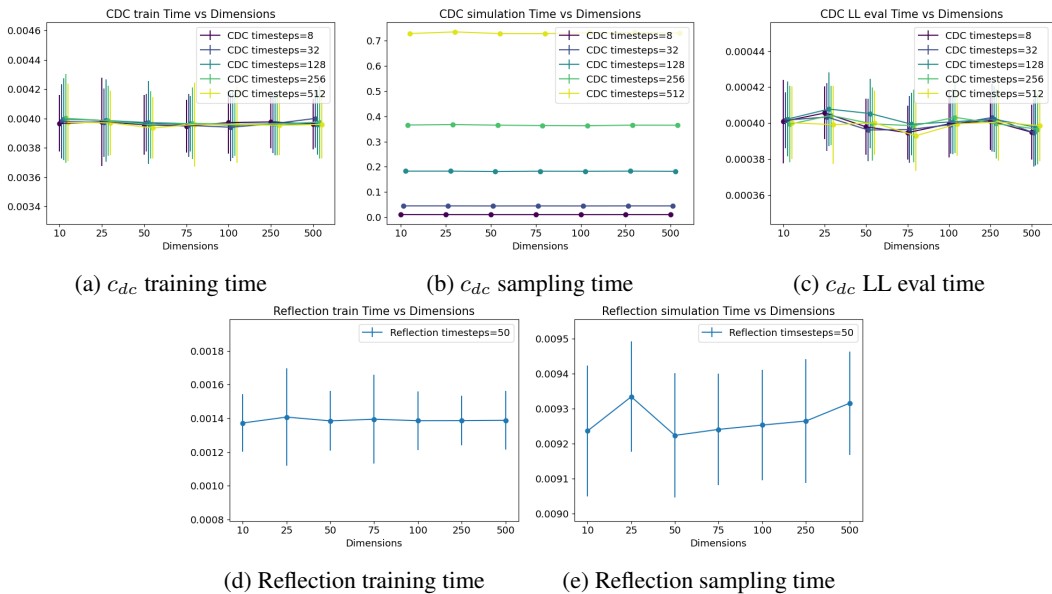

Figure 7: **Computational time study:** For varied dimensions and time steps, we report our models' training, sampling and LL eval (for the $c_{dc}$) times in seconds, reported as the mean over 25 runs with $\pm$ one standard deviation.

## B.2 SIMULATION STUDY

To show that our $c_{dc}$ copula obtains correct density estimates, we study an example with an analytical density. Concretely, we use the following data generating process as the ground truth copula:

**Copula of a mixture of Student's T distributions.** Let $k = 4$ be the number of mixture components, with probabilities $[\pi_1, \pi_2, \pi_3, \pi_4] = [0.3, 0.3, 0.2, 0.2]$, each a multivariate Student's T distribution with 10 degrees of freedom, with densities denoted by $S_k^{10}$. For a mixture component, define the mean $\mu_k$ as

$$\mu_k = 4\mathbf{x} \quad \text{with} \quad \mathbf{x} \sim \mathcal{N}(.; \mathbf{0}, \mathbf{I}_4),$$

with correlation matrix $\Sigma_k$ obtained using the Davies-Higham Algorithm (Davies & Higham, 2000) based on eigenvalues constructed by sampling 4 uniform variables in $[0, 1]$, and normalising the vector to sum to $d$. The final copula density at a value $\mathbf{u} = (F^1(\mathbf{x}^1), \dots, F^d(\mathbf{x}^d))$ with $F^i$ the univariate mixture of Student's T distribution functions, is obtained by the ratio of the joint mixture density divided by the product of mixture marginal densities:

$$c(\mathbf{u}) = \frac{\sum_{k=1}^4 \pi_k \cdot S_k^{10}(\mathbf{x}; \mu_k, \Sigma_k)}{\prod_{i=1}^d \{\sum_{k=1}^4 \pi_k \cdot S_k^{i,10}(\mathbf{x}^i; \mu_k^i, (\Sigma_k)^{ii})\}},$$

where $S_k^{i,10}$ is a univariate student's T density with 10 degrees of freedom, $\mu_k^i$ is the i$^{\text{th}}$ entry of $\mu_k$, and $(\Sigma_k)^{ii}$ is the i$^{\text{th}}$ diagonal term of $\Sigma_k$. The resulting copulas are multimodal with a shape not well captured by common parametric copulas, making them fit to assess our models' flexibility while still maintaining access to the ground truth density. They also display traits of extreme dependence due to the heavy tailed nature of student's T distributions.

**Simulation study results.** We assess our $c_{dc}$ model's LL values against those of the Ratio copula, in dimensions $d = 10, 50, 100$ with 8000 samples to train on and 2000 samples in the test set. We use the same model architectures for the $c_{dc}$ and ratio copulas as for the scientific datasets, with the $c_{dc}$ having 8 time classes with a KL-based discretisation (see discussion on time discretisations above in Apdx. B), while the ratio copula only has 2.

In Tab. 5, we report the mean absolute error (MAE) and the mean squared error (MSE) of LL values of a run aggregated over 10 independently and identically sampled datasets from a fixed mixture of

a given dimension. Standard deviations over the 10 runs are shown in subscripts. Our $c_{dc}$ obtains more accurate density estimates than the Ratio copula, and does so with less variability between runs too. This difference is accentuated in higher dimensions.

We visualise a single run from the 10-dimensional setup in Fig. 10, where we show violin plots displaying the densities of LL errors, with dots overlaid on top to represent samples for which the specific LL error was obtained. We colour-code these samples based on the ground truth copula LL, allowing us to discern patterns with respect to the original copula LL. For instance, the ratio copula overestimates the LL of high-density samples under the original copula (blue samples tend to have $c(\mathbf{u}) > ratio(\mathbf{u})$), and underestimates the LL of low-density samples (red samples tend to have $c(\mathbf{u}) < ratio(\mathbf{u})$). Our $c_{dc}$ slightly displays a similar pattern with red dots being more frequent near the bottom of the violin plot, but the pattern is less pronounced. Most of our model's errors concentrate near 0, showing it correctly represents the dependence of the ground truth copula. Bivariate visualisations are displayed in Fig. 8 as aggregate views of all but two dimensions (showing just $\mathbf{u}^1, \mathbf{u}^2$), showcasing their multimodality.

Table 5: **Simulation study:** For analytically tractable mixture copula densities in increasing dimensions, our $c_{dc}$ model obtains more accurate density estimates with lesser variance.

|  | $d = 10$ | $d = 50$ | $d = 100$ |
|---|---|---|---|
| $c_{dc}$ MAE | $3.34_{\pm 0.12}$ | $13.73_{\pm 0.56}$ | $26.56_{\pm 1.56}$ |
| Ratio MAE | $3.40_{\pm 0.22}$ | $34.18_{\pm 17.15}$ | $65.87_{\pm 11.08}$ |
| $c_{dc}$ MSE | $18.03_{\pm 1.23}$ | $291.43_{\pm 21.21}$ | $1077.30_{\pm 106.82}$ |
| Ratio MSE | $19.30_{\pm 1.96}$ | $2324.58_{\pm 2237.82}$ | $4808.75_{\pm 1239.08}$ |

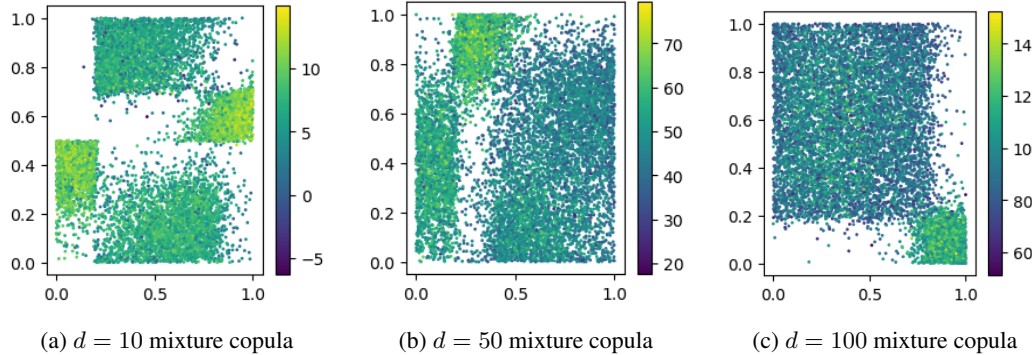

(a) $d = 10$ mixture copula     (b) $d = 50$ mixture copula     (c) $d = 100$ mixture copula

Figure 8: **Simulation study:** Plots of $\mathbf{u}^1, \mathbf{u}^2$ copula samples with colours representing LLs.

B.3 CONVERGENCE OF PROCESSES TO THEIR LIMITING DISTRIBUTION

As both methods use a terminal time at which the copula of the process has converged to independence, we show here a practical method for selecting such times. For observed copula data, we apply the forward process to it and measure the Wasserstein-2 distance of the samples at different times with respect to a uniform distribution, which is the stationary distribution here. We also measure the W2 of a uniform distribution to itself using two sets of uniformly sampled data, which are shown in orange as a baseline. The reflection process is shown in blue while the OU process is shown in purple. In Fig. 9, we see that already for $t \approx 1$, both processes obtain similar W2 values to uniform samples, indicating they are suitably close to stationarity. This exemplifies the fast convergence rates derived in Section 3 for the OU, and provides support for the reflection process as a good choice of forward process.

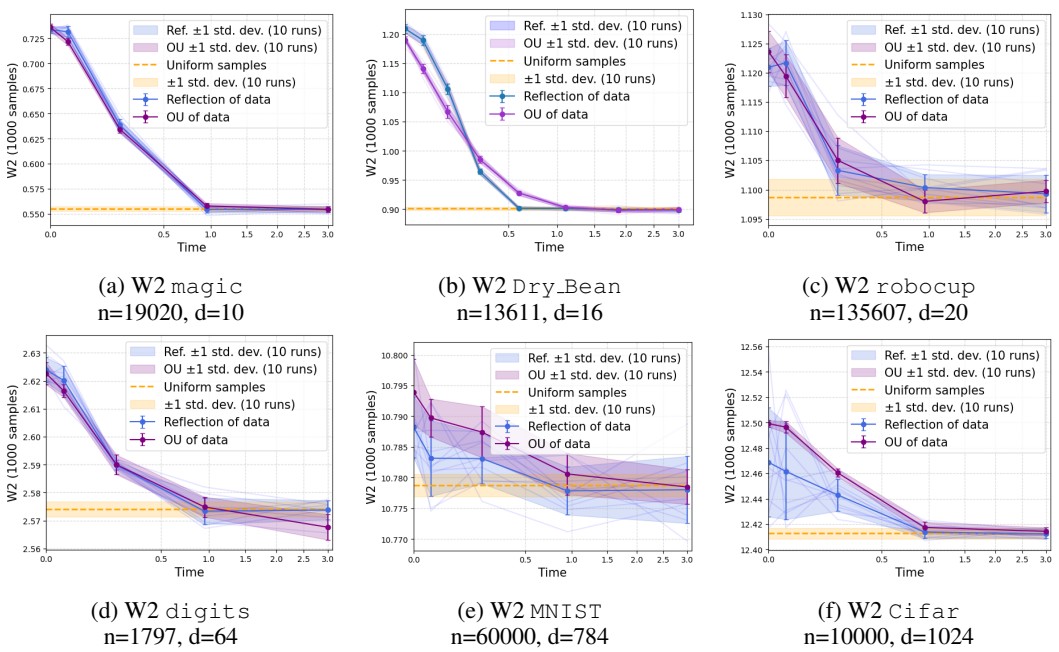

Figure 9: **Convergence to uniformity of forward copula process.** Initialised at 1000 copula observations, we run the processes forward in time and measure the Wasserstein-2 distance with respect to the Uniform distribution. We show the result with one standard deviation in blue across 10 measurements, depicting the fast convergence to uniformity.

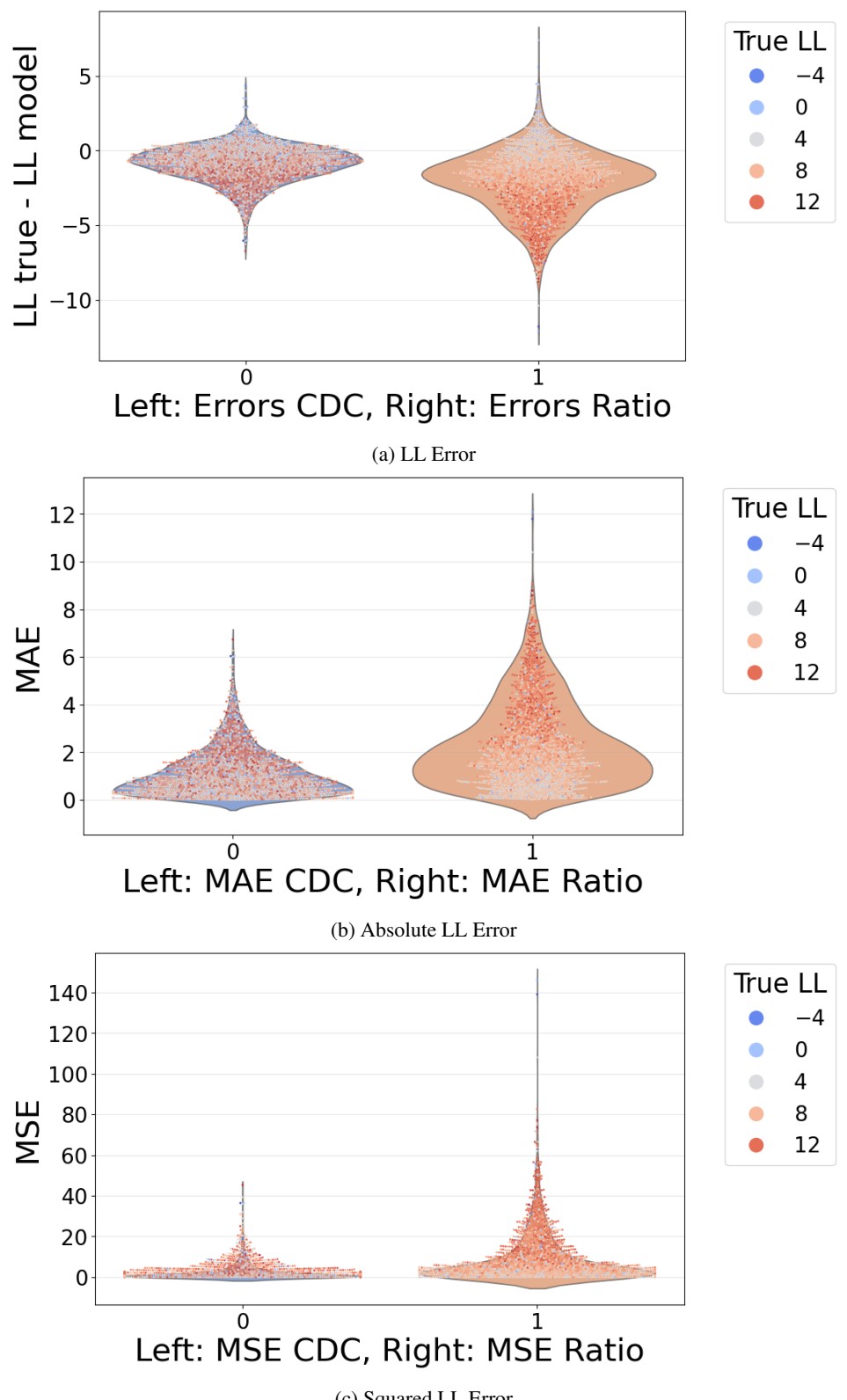

(a) LL Error

(b) Absolute LL Error

(c) Squared LL Error

Figure 10: **Simulation study:** We show the distribution of errors for our $c_{dc}$ and the ratio copula. Overall, our model achieves more accurate density estimates, with errors of smaller magnitudes.

## B.4 ABLATIONS ON THE MIXTURE LOSS

For our experiments, we use a mixture weight $\alpha$ to equalise both loss terms, following previous work Yadin et al. (2024). Here, we provide an ablation to investigate the sensitivity of the $c_{dc}$ to this choice. We perform a sweep over values $\alpha = (0.0, 0.01, 0.25, 0.5, 0.75, 1.0, 5.0, 20.0, 500.0)$ for the experiments on scientific datasets. We report the resulting LL values on a held-out validation set (10% of the train set) in Fig. 11. Our model is shown to be robust to values of $\alpha > 0.01$ with a stable performance across different values. This indicates that incorporating the CE term is important to achieve optimal performance, which echoes our theoretical analysis in Thm. 5.

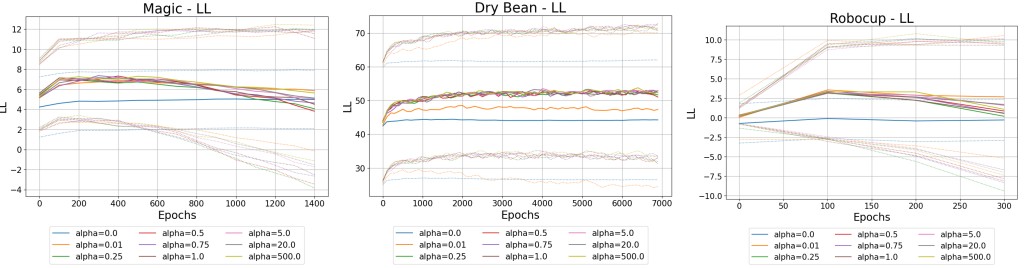

Figure 11: **Ablation on mixture loss**: We train with varied values of the mixture hyperparameter $\alpha$ and report LLs ($\pm$ std. as dotted lines) on the validation set. Our method is robust to all values $\alpha > 0.01$ with a stable performance over hyperparameters.

## B.5 UNIFORMITY OF GENERATED SAMPLES

For a copula to correctly sample a joint density, its samples must be uniformly distributed for each dimension. This property is not obvious to maintain with deep learning models. As a diagnostic tool following probabilistic calibration literature (Hamill, 2001), in Figs. 12, 13, 14, 15, 16, and 17, we show aggregate rank histograms over dimensions, split over 10 intervals of length 0.1 over the $[0, 1]$ support[4]. For each subplot, the histogram comes from 10 independent runs of a model on that dataset. As the Gaussian, Vine, and IGC copulas enforce uniform marginals on samples by construction, we only show plots for our two models and the Ratio copula. In grey, we show truly uniform samples averaged over the same quantities. Therefore, deviations from the grey histogram indicate non-uniformity of samples. Deviations to the right compared to the uniform histogram show an over-representation of samples in that percentile range, while a deviation to the left shows an under-representation of samples in that percentile range. See Hamill (2001) for more details on rank histograms. The aggregate rank for run $r$ over a bin $b_k$ in $\{b_1, \ldots, b_{10}\} = \{[0, 0.1], \ldots, [0.9, 1]\}$ is computed as:

$$\text{Rank}_{k,r} = \frac{10}{N \cdot d} \sum_{n=1}^{N} \sum_{i=1}^{d} \mathbb{I}_{b_k}(\mathbf{u}^{(i,n,r)})$$

where $\mathbf{u}^{(i,n,r)}$ denotes the $n$-th sample in dimension $i$ from run $r$, $d$ is the dimensionality, and $N = 1000$ is the sample size per run, with the indicator function $\mathbb{I}_A(x) = 1$ if $x \in A$ taking the value 0 else.

Overall, our two methods produce histograms similar to uniform samples, while the Ratio copula obtains more spread-out histograms. This means that the Ratio copula obtains biased samples each run, but the type of bias is different across runs. This issue is indicative of the difficulty of accurately sampling complex high-dimensional distributions with MCMC methods. All methods generally perform worse on `digits` and `Cifar`, possibly indicating the difficulty that those datasets represent for a copula sampling task.

---

[4]Here, ranks refers to the samples on copula space.

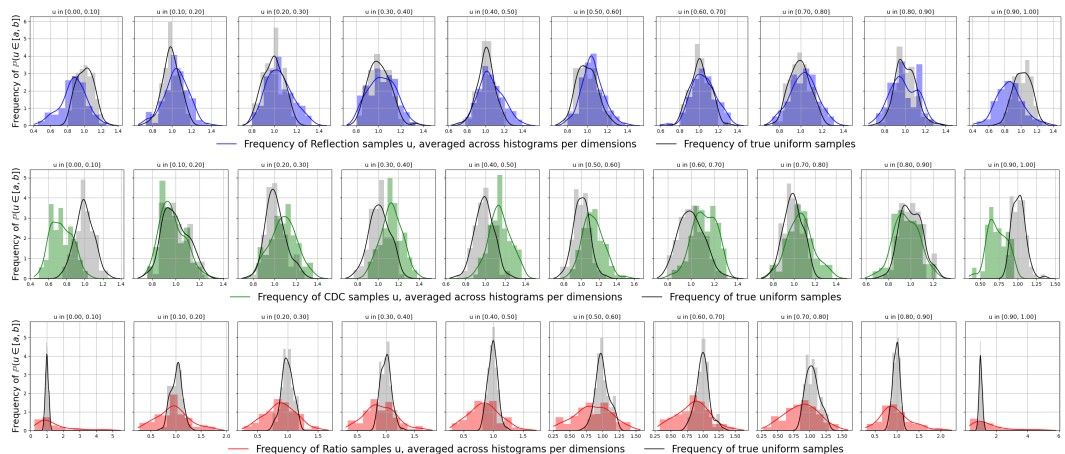

Figure 12: **Uniformity of samples for** `Magic`:

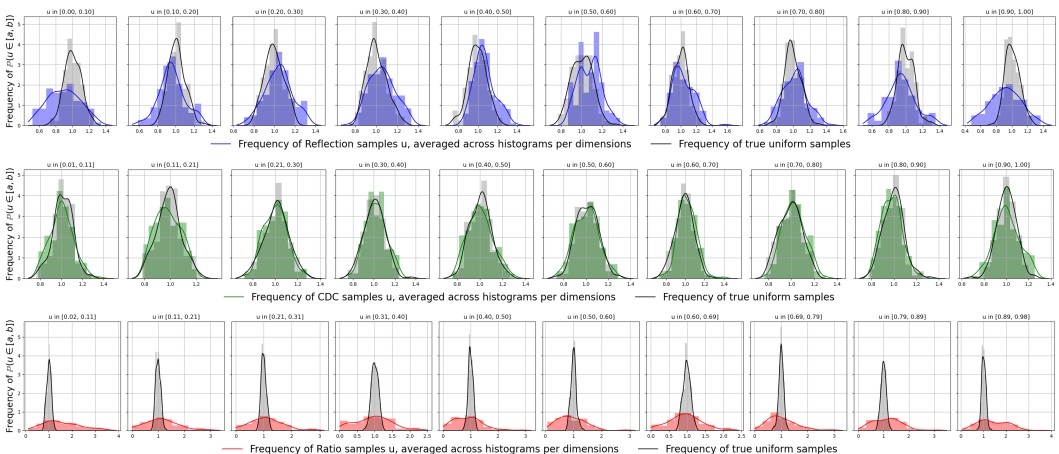

Figure 13: **Uniformity of samples for** `Dry_Bean`.

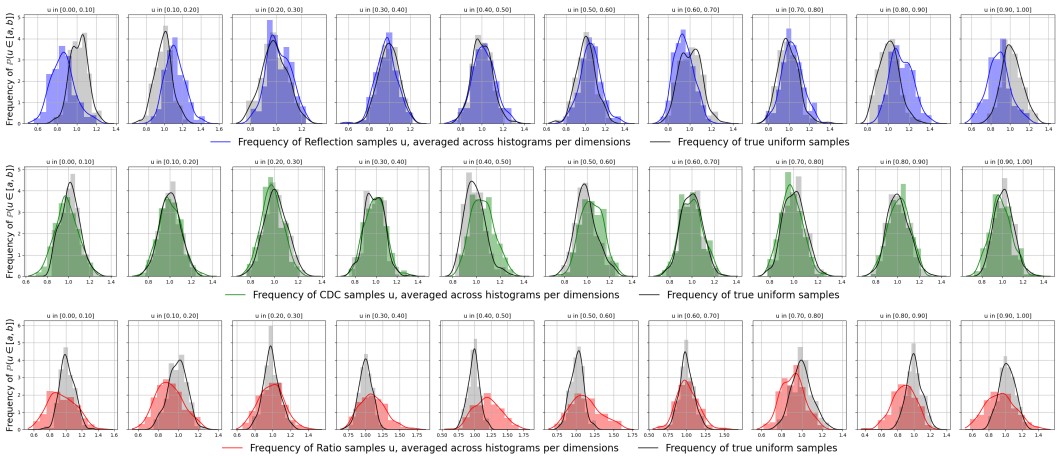

Figure 14: **Uniformity of samples for** `Robocup`.

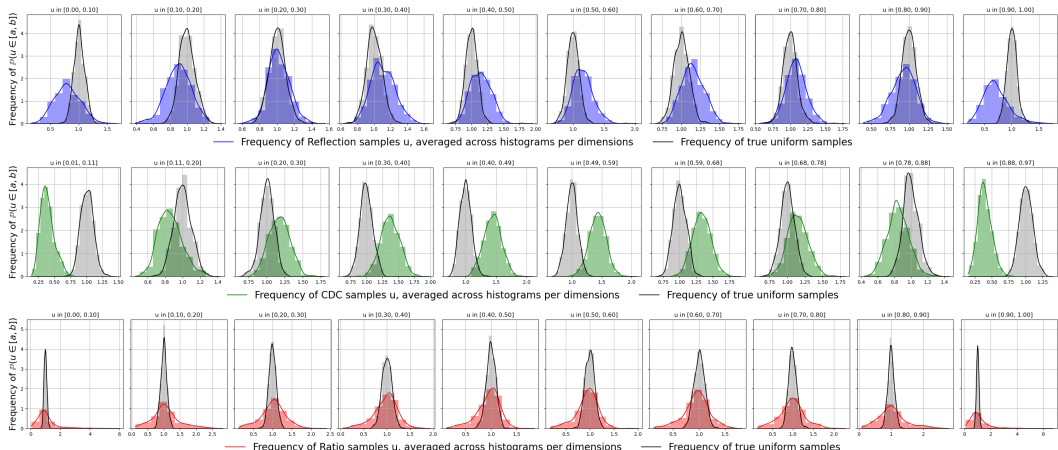

Figure 15: **Uniformity of samples for** `digits`.

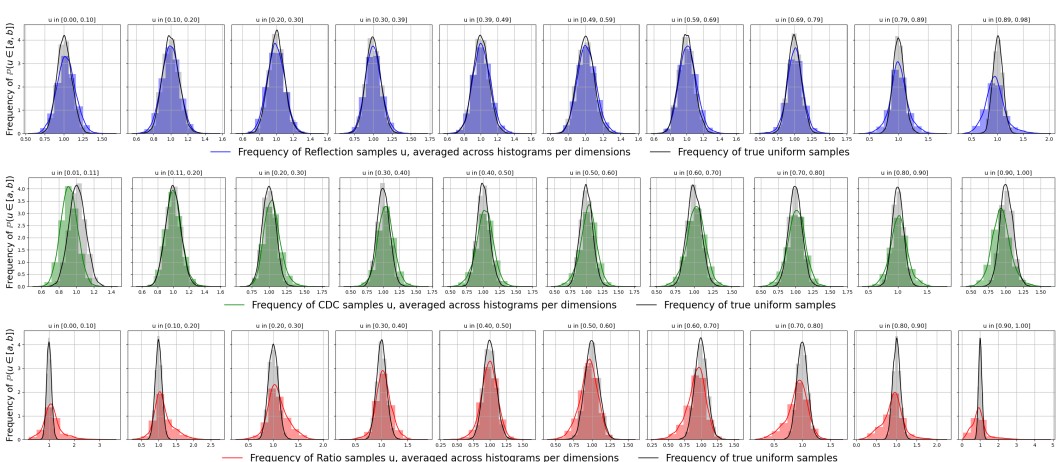

Figure 16: **Uniformity of samples for** `MNIST`.

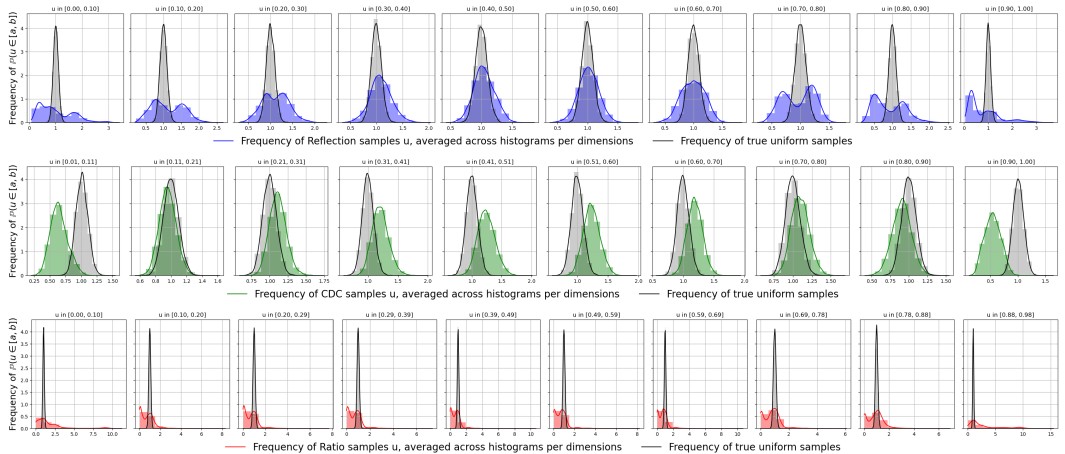

Figure 17: **Uniformity of samples for** `Cifar`.

We further investigate the uniformity of our samples with quantitative metrics in Tab. 6. For our models to produce accurate copula samples, it is required that generated samples follow the respective marginal distributions. For the $c_{dc}$, as the model operates on the Gaussian scale, we now include an additional check for the normality of the generated samples via the statistical test of D'agostino & Pearson (1973). For the reflection copula which operates on the hypercube, we verify the uniformity of samples with a two-sample Kolmogorov-Smirnov test for goodness of fit (Hodges Jr, 1958). Along with the Ratio copula (operating on the Gaussian scale), we report the rejection rate of the hypothesis that marginals are preserved, averaged over dimensions and model runs, at the $5\%$ alpha level with a Bonferroni correction due to multiple testing over runs and dimensions.

The results indicate that for most experiments, our models produce samples that do not significantly deviate from prescribed marginals and do so more effectively than the existing Ratio copula model. In particular, our $c_{dc}$ achieves a small rejection rate, which we attribute to the sampling procedure using Gaussian noise during sampling (see step 8 in Alg. 1). Note that this test only indicates departures from uniformity, but does not inform about their severity.

To strengthen these findings, we additionally report the Continuous Ranked Probability Score (CRPS) as a univariate probabilistic calibration metric to compare our generated copula samples to uniform data, dimension-wise. As a strictly proper scoring rule (Gneiting & Raftery, 2007), it is minimised if and only if both samples come from the same distribution (Dawid et al., 2016), here being the uniform on $[0, 1]$. In Tab. 6, we report the mean CRPS averaged across dimensions and runs, from which we subtract the CRPS obtained with pairs of uniform samples (computed to be 0.1666 over 100 repetitions of 1000 vs 1000 draws). Thus, the ideal value for this metric would be 0, uniquely achieved by a uniform sample.

Our results corroborate the statistical tests, showing that our models achieve samples close to uniformity, with a noticeable improvement over the Ratio copula model.

Table 6: **Marginal uniformity metrics:** For the $c_{dc}$, Reflection, and Ratio copulas, for each dimension, we report the rejection rate of statistical tests for the respective marginal distributions. We also report CRPS values to quantify deviations from uniformity. Our models generally achieve samples close to uniformity, with a noticeable improvement over the Ratio copula model.

|  | Magic | Dry_Bean | Robocup | digits | MNIST | Cifar |
|---|---|---|---|---|---|---|
| $c_{dc}$ - test rejection | 0.01 | 0.01 | 0.0 | 0.03 | 0.00 | 0.00 |
| Reflection - test rejection | 0.02 | 0.09 | 0.00 | 0.16 | 0.01 | 0.70 |
| Ratio - test rejection | 0.47 | 0.93 | 0.16 | 0.30 | 0.09 | 0.50 |
| $c_{dc}$ - CRPS | 0.0029 | 0.0021 | 0.0021 | 0.0068 | 0.0022 | 0.0041 |
| Reflection - CRPS | 0.0024 | 0.0030 | 0.0022 | 0.0036 | 0.0024 | 0.0142 |
| Ratio - CRPS | 0.0169 | 0.0128 | 0.0028 | 0.0150 | 0.0070 | 0.0680 |

## B.6 VISUALISING SAMPLES

**Visualising scientific datasets.** We show pair plots for each dataset below, obtained from 1000 samples from our model, shown in the lower left, compared to 1000 randomly drawn observations from the test set (only one of our 10 runs is shown). We note that pair plots are unable to convey higher-order dependencies; they only provide an aggregate view of the relationship between two variables. However, this lets us analyse the accuracy of our models in sampling from the correct regions. As can be seen in Fig. 20, both our methods closely match the pair plots of observed data, displaying the same patterns across dimension pairs. We also display the accuracy of different methods at targeting specific regions of the support in Fig.18.

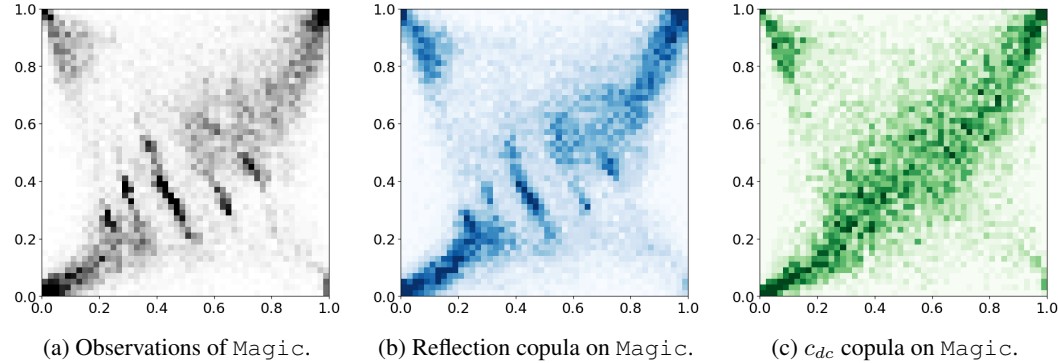

(a) Observations of `Magic`.    (b) Reflection copula on `Magic`.    (c) $c_{dc}$ copula on `Magic`.

Figure 18: **Bivariate plot of copula samples vs observations.** We show the ability of our copula models' samples to match the target distribution. The reflection copula is particularly apt at capturing details.

**Visualising high-dimensional image samples.** We further show 25 samples per method for image datasets in Figs. 21, 22, 23. From the samples shown on the copula scale in Figs. 21 and 23, note how the `MNIST` dataset consists of variation mostly in the centre of the image, while the edges are mostly noise. Thus, models need to target specific dependencies in parts of the $[0, 1]^{784}$ hypercube, while ignoring most of the remaining dimensions. In contrast, for `Cifar`, the whole image meaningfully contributes to the dependence, meaning models need to make the full 1024 dimensions fit a specific dependence pattern to generate an image. We further inspect the dependence of images by downscaling the `MNIST` images to lower resolutions until equaling that of the `digits` data in Fig. 19.

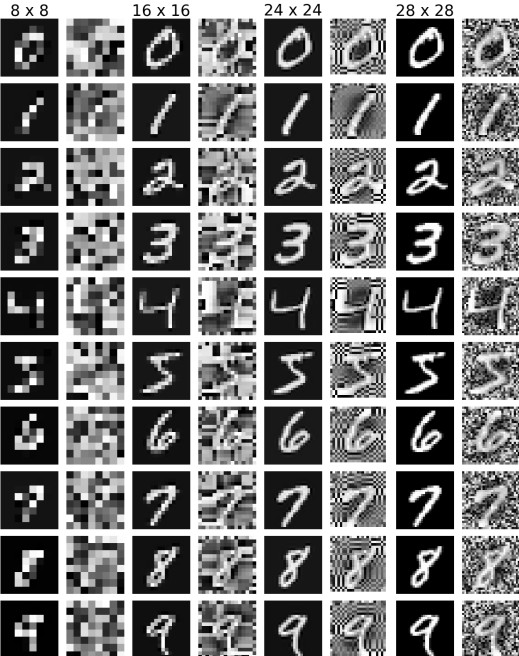

Figure 19: **Dependence of downscaled images.** We downscale the `MNIST` data to the resolution of `digits` and observe the effects this has on the dependence via the copula scale. Dependence is fundamentally different, as the digits are no longer discernible on the copula scale and appear more like random noise.

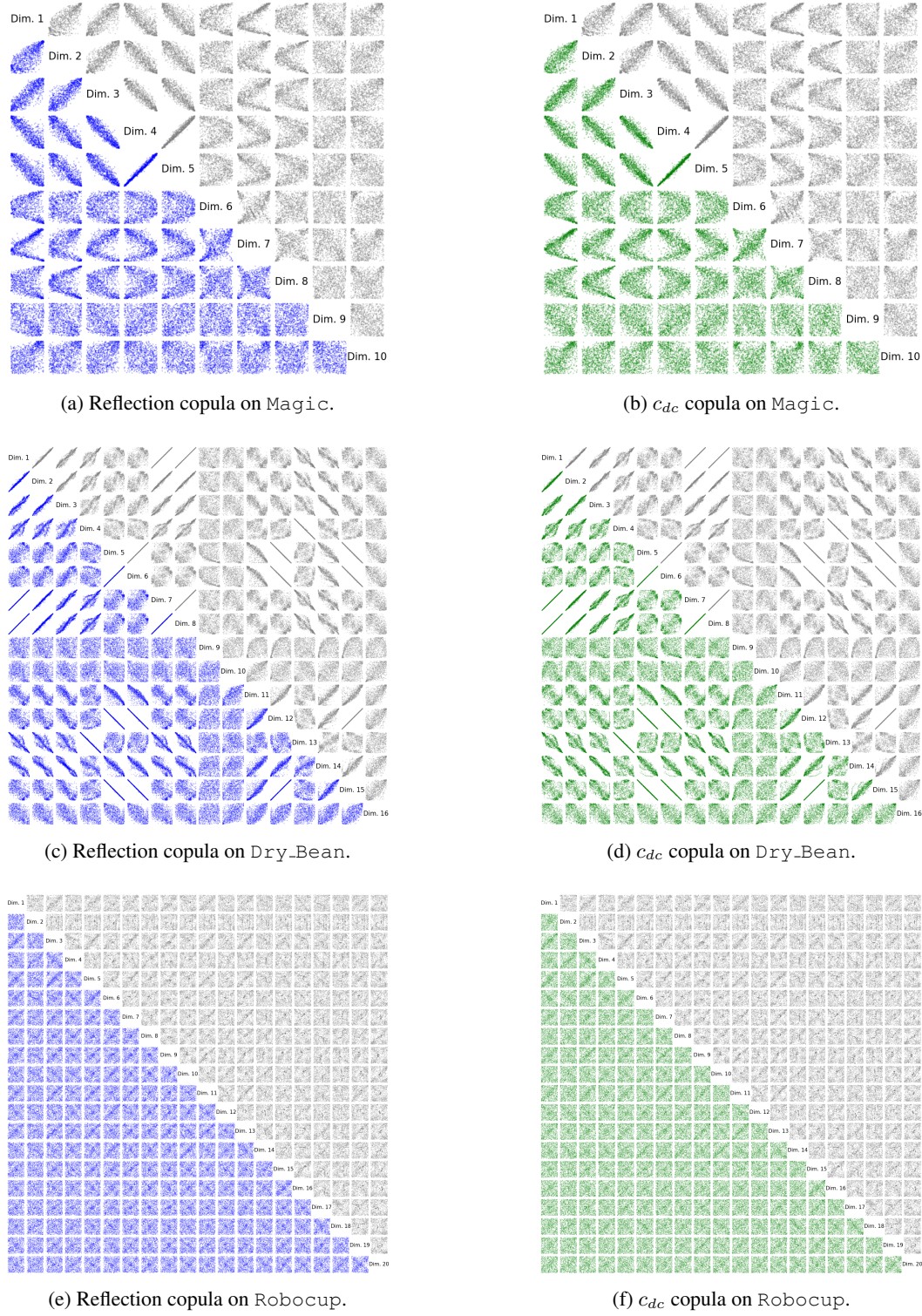

(a) Reflection copula on `Magic`.

(b) $c_{dc}$ copula on `Magic`.

(c) Reflection copula on `Dry_Bean`.

(d) $c_{dc}$ copula on `Dry_Bean`.

(e) Reflection copula on `Robocup`.

(f) $c_{dc}$ copula on `Robocup`.

Figure 20: **Pair plots of copula samples vs observations.** In the lower left of each plot, we show our copula samples **u** with the reflection copula in blue in the left column and our $c_{dc}$ copula in green in the right column. Observed data is shown in the upper right of each plot in grey. Note that instead of being mirrored, the dimension pairs are reversed for the bottom-left parts of the plots to match the orientation of the upper-right pairs. Best viewed digitally.

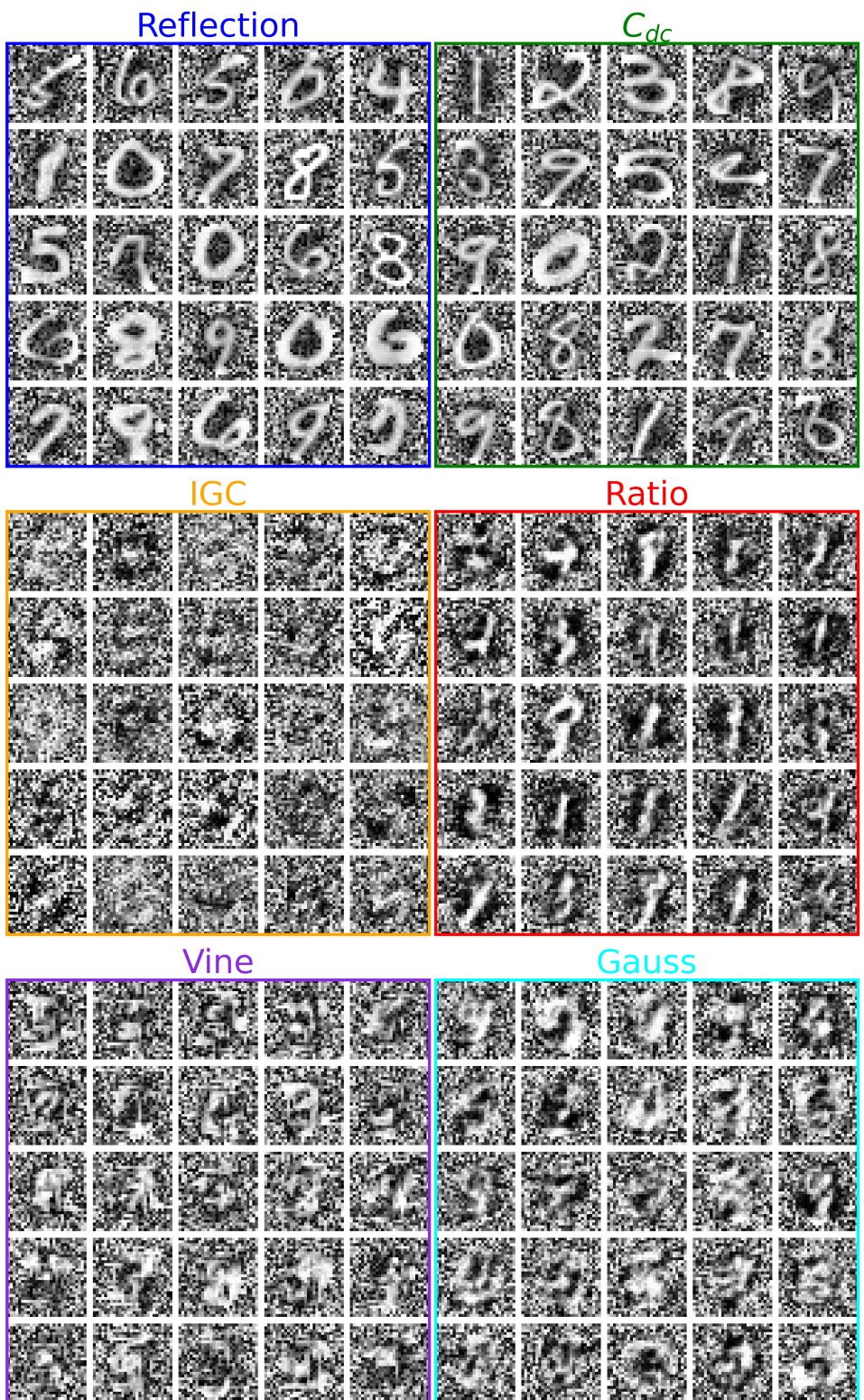

Figure 21: **Samples for** MNIST: Random samples from all methods, shown on the copula $[0, 1]^{784}$ scale with lighter values being closer to 1. Notice how most of the variation takes place in the centre, where the shape of the number appears as high **u** values. However, image edges are not very dependent and take random **u** values.

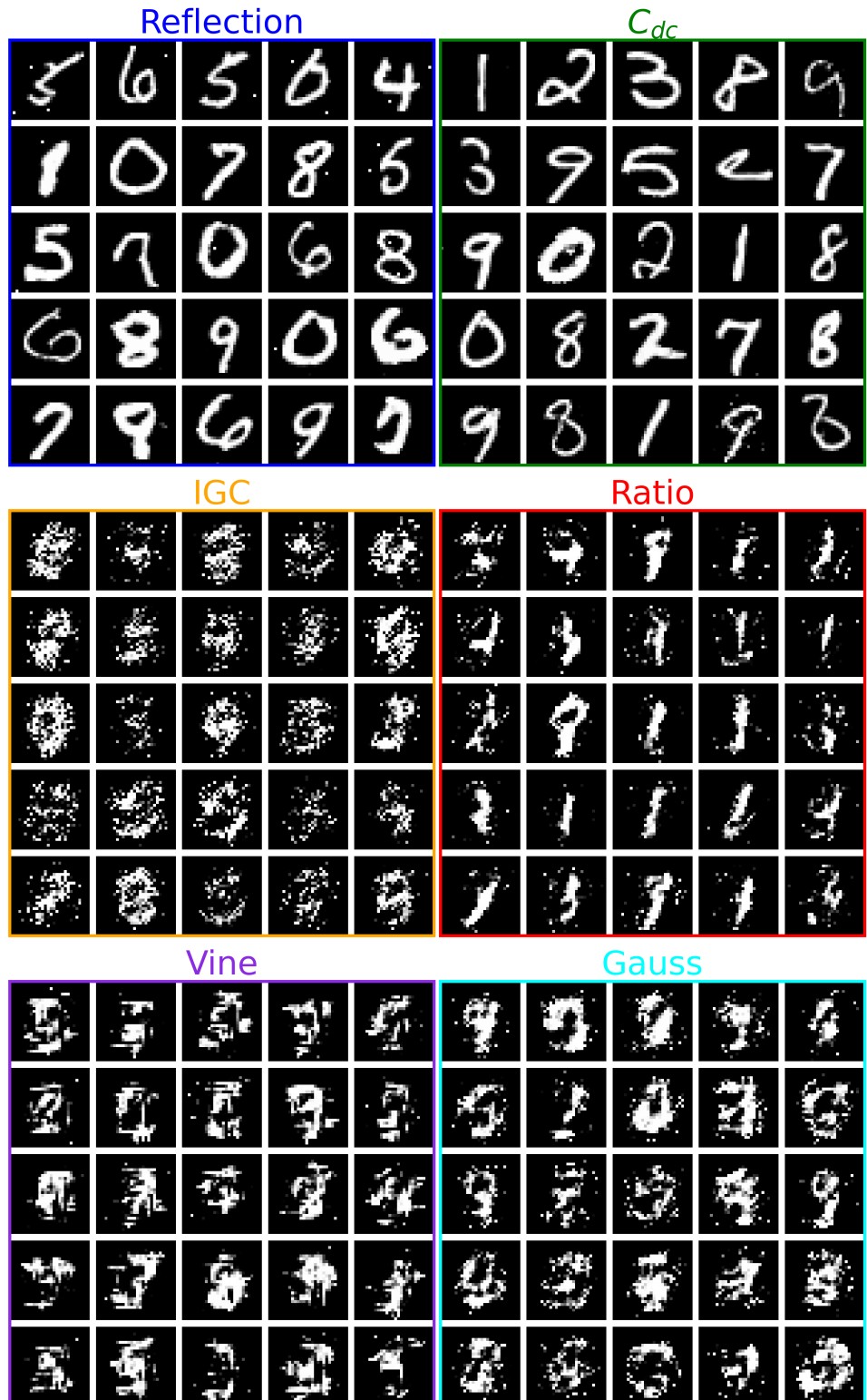

Figure 22: **Samples for** MNIST: Random samples from all methods, shown on the data $\mathbb{R}^{784}$ scale with lighter values being higher. Our reflection and $c_{dc}$ copulas correctly produce samples resembling digits, while competing copulas struggle to produce coherent samples.

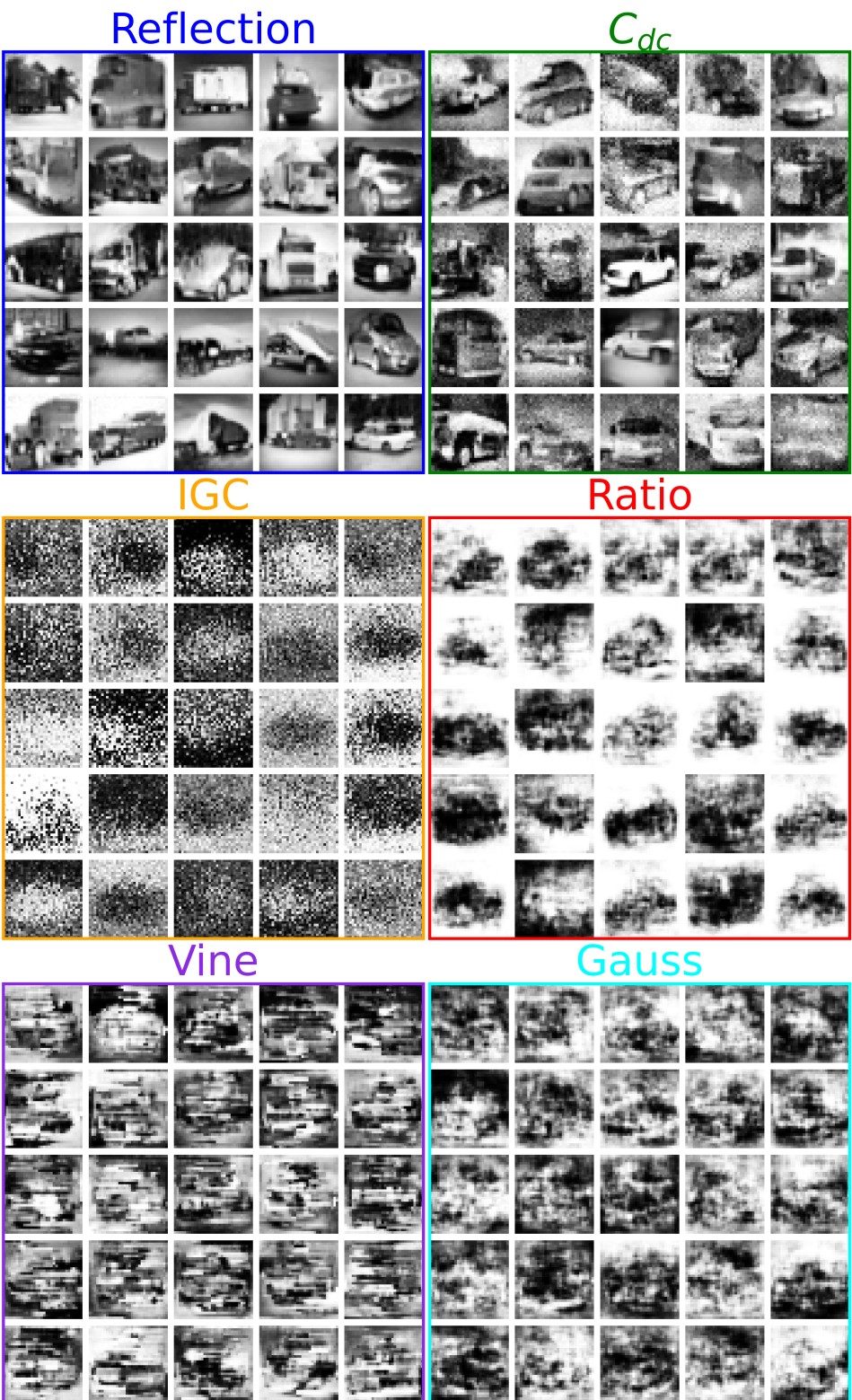

Figure 23: **Samples for** `Cifar`: Random samples from all methods, shown on the copula $[0, 1]^{1024}$ scale with lighter values being closer to 1. Notice how here, all dimensions of the image are dependent and matter in producing coherent samples. As a consequence, the images are already visible on the copula scale without needing to transform them to the data scale.

**Sanity checks on generated image samples.** For both of our copulas, we verify that our models did not simply remember the training data but truly learned the data distribution. We qualitatively inspect the $k = 10$ nearest neighbour (KNN) from the training set to our samples, measured with the Euclidean distance in $\mathbb{R}^d$, reported in Fig 24. Our samples differ from the nearest observations, showing that the model learned the data distribution beyond pure memorisation of training examples.

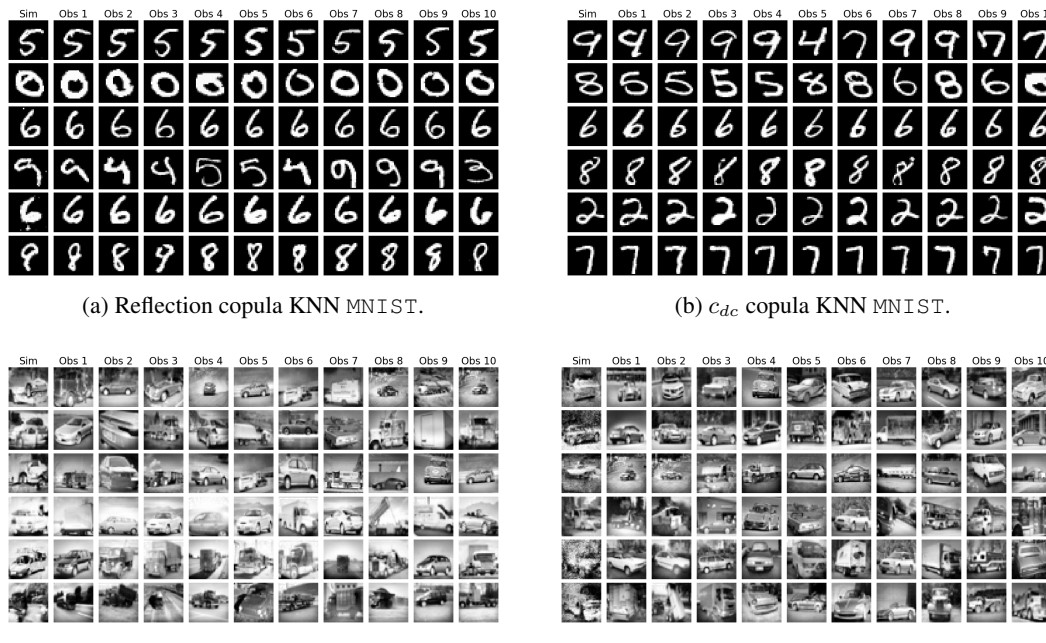

(a) Reflection copula KNN MNIST.

(b) $c_{dc}$ copula KNN MNIST.

(c) Reflection copula KNN Cifar.

(d) $c_{dc}$ copula KNN Cifar.

Figure 24: **10-nearest neighbours of generated samples:** For random samples from our models, we show the 10 nearest neighbours with respect to the Euclidean distance. It is apparent that the generated samples differ from the observations, meaning the models did not simply remember the training dataset. Best viewed digitally.

## C THE USE OF LARGE LANGUAGE MODELS (LLMS)

In this paper, we only used LLMs to prototype data visualisations, to check parts of the code for errors, and to supplement the search for related works. All outputs of LLMs were verified by humans. All other aspects of this research were done by humans, specifically, originating the research idea, deriving proofs, producing diagrams and final figures for the paper, writing code for the models, conducting experiments, reviewing related works, and writing the paper.

