# OpenReview forum: "Diffusion and Flow-based Copulas: Forgetting and Remembering Dependencies"
_ICLR.cc/2026/Conference — ICLR 2026 Poster_

### Official Review · Reviewer_5pKF · 2025-10-25

**Soundness:** 3
**Presentation:** 3
**Contribution:** 3
**Rating:** 6
**Confidence:** 3

**Summary:**

This paper introduces a novel framework for copula modeling using diffusion and flow-based methods, focusing on the processes of "forgetting" and "remembering" dependencies within data. The authors propose two new models: the classification-diffusion copula and the reflection copula. The CDC is designed as an effective density estimator by framing the problem as a classification task to estimate the ratio of copula densities at different stages of a diffusion process. The Reflection Copula is a generative model built on a flow architecture that enables efficient sampling. Theoretically, both models are shown to recover the true underlying copula at optimality.

**Strengths:**

**S1**. The core idea of explicitly modeling the "forgetting" (diffusion to independence) and "remembering" (density estimation/generation) of dependencies provides a fresh and theoretically grounded perspective on copula modeling.

**S2**. The introduction of the classification-diffusion copula (leveraging classification for density estimation) and the reflection copula (a dedicated flow for generation) are sound and well-motivated solutions.

**S3**. The most significant strength is the demonstrated ability to achieve SOTA results on complex, high-dimensional data, including image datasets. The models clearly outperform classical (e.g., Vine) and other deep copulas, which struggle with scalability and complexity. The paper also provides thorough empirical validation across diverse datasets, using multiple metrics and visualizations to support its claims.

**Weaknesses:**

**W1**. While the reflection copula samples quickly, the training times for both models, especially the CDC, are reported to be very long (e.g., up to 4 hours for CDC on CIFAR). This high computational cost could hinder practical application.

**W2**. For image generation, the CDC samples are noted to be "slightly noisier and grainier" compared to the smoother samples from the reflection copula, indicating a potential trade-off between accurate density estimation and sample fidelity.

**W3**. The paper could benefit from more detailed ablation studies to isolate the contribution of specific architectural choices or loss components to the overall performance.

**Questions:**

No other issues; please refer to the Weaknesses section for specific points.

---

> ### Author Response · Authors · 2025-11-21
> **Answers for Reviewer 5pKF - Part 1**
>
> We sincerely thank reviewer 5pKF for their time and positive review of our work. Please find answers to your comments below.
>
> ---
>
> > The training times for both models, especially the CDC, are reported to be very long (e.g., up to 4 hours for CDC on CIFAR).
>
> Our models have the same computational requirements as related diffusion and flow models, which for sampling depends on the time to obtain scores/velocities multiplied by the number of steps to simulate sampling dynamics for.
>
> Further, we note that the use cases of copulas often require quick inference times for sampling and LL evals rather than during training. This means that the important factor for practical use is not training time but rather LL and sampling times, which are strengths of our copulas.
>
> We now report computational times for all models in Apdx. C1 Tab. 4, showing that **our sampling times are competitive with vine copulas while being much faster than the Ratio copula**. We additionally present a dedicated computational time study in Apdx. C1 Fig. 6. From our computational study, we conclude that the network model is the most influential, followed by the number of timesteps used to simulate the SDE/ODE when sampling. We point to this in Section 6 of the main text.
>
> In more detail, for $d$ the data dimension , $w$ the network width and $h$ the number of hidden layers, $c_{dc}$ training has a complexity of $\mathcal{O}(d\cdot w + h\cdot w^{2})$. This is because the model performs one forward pass (for class probabilities) and one backward pass (for scores) before taking gradients with respect to the loss during training. During inference, to sample the $c_{dc}$, we perform a forward and backward pass for each of the $k$ classes, following a SDE discretization scheme, resulting in a complexity of $\mathcal{O}(k\cdot(d\cdot w + h\cdot w^{2}))$. For the reflection copula, one training epoch consists of a single forward pass followed by a weight update, with complexity $\mathcal{O}(d\cdot w + h\cdot w^{2})$. Generating samples requires $m$ network evaluations where $m$ is the number of steps to simulate the ODE for, which scales as $\mathcal{O}(m\cdot(d\cdot w + h\cdot w^{2}))$.
>
> > The CDC samples are noted to be "slightly noisier and grainier" compared to the smoother samples from the reflection copula
>
> We now provide a more comprehensive explanation of the sampling capabilities of the cdc to expand on the “noisier” comment in the main text, and now point to the following discussion in the same passage of Sec. 6.2.
>
> A key component of our $c_{dc}$ sampling approach is stochasticity in form of the exploration step 8 in Alg. 1. As argued in our general answer on uniformity, this exploration is beneficial for maintaining the uniformity of our samples, thus being a core strength of our model. The result of this stochasticity is that targeting very narrow parts of the support becomes rarer. We illustrate this in Fig.17, showing 2d histograms of samples from the $c_{dc}$ and the reflection copulas against ground truth samples for two dimensions in the magic dataset.
>
> In the case of the digits data, the dependence is **fundamentally different from the MNIST dataset, as the digits are no longer discernible on the copula scale** and appear more like random noise, see Fig. 18 of Apdx. B6. With the low sample size of <2000 compared to the dimension, only very specific regions of the support correspond to valid samples.
>
>  We believe this key difference is causing the lower FID score of the $c_{dc}$, since the FID is sensitive to such small deviations that are perceived as granularity of the image. On the other hand, the W2 prioritises adherence to the statistical properties of samples, which is reflected by the success of the $c_{dc}$ for that metric. As can be seen from FID results for other image datasets, this phenomenon largely disappears with a greater sample size.
>
> Finally, this only relates to the sampling of the $c_{dc}$ since our likelihoods are accurate across all settings, as verified by Tabs. 1, 2, and 5.
>
> > The paper could benefit from more detailed ablation studies
>
> We now give a **comprehensive evaluation of different components of our model**, including ablations on the **loss mixture** in Apdx. B4, additional **uniformity diagnostics** in Apdx. B5, an improved **computational analysis** in Apdx. B1, and an ablation on **schedulers** in Apdx. B under “Time discretisation”. We now point to these new experiments in Section 6, and include discussions of each in relevant parts of the main text, in Sections 3, 4, 6, and 7.  Please see our general answer for full details.
>
> ---
>
> We are sincerely grateful for your positive view of our work and your insightful comments. If you find that our answers have adequately addressed your comments, we would kindly ask you to consider increasing your score. We are happy to address any further questions and remain available for the rest of the discussion period.

---

### Official Review · Reviewer_dHoi · 2025-10-31

**Soundness:** 3
**Presentation:** 3
**Contribution:** 3
**Rating:** 6
**Confidence:** 3

**Summary:**

The authors propose a type of flow mapping that maps a copula to the independence copula, i.e. the authors define a mapping that forgets dependencies. The model borrows techniques inspired by diffusion models to define a mapping that forgets dependencies such that an appropriate independent copula can be reversed and transformed to new samples that map to the observed data. The authors propose two variants: one that provides likelihoods and another that only provides samples. The authors illustrate the method on a number of high dimensional datasets to showcase the extent to which the method can be applied.

**Strengths:**

The authors propose an interesting direction to enable high dimensional sapling from copulas, a difficult task that extends the applicability of copulas.

The method scales reliably to high dimensions which has been somewhat of an elusive property of copulas up to now.

**Weaknesses:**

The architectures prescribed do not preserve the necessary components to ensure a valid copula is being recovered, which can make some of these tools a bit difficult to employ in practice (e.g. if one needs specific families of marginals to be used).

The authors also miss some relevant works on deep learning and scaling copulas which should be taken into account, especially since these consider high dimensional questions using the stochastic representations [1,2,3].

[1] Inference and Sampling for Archimax Copulas, NeurIPS 2022

[2] Generative Archimedean Copulas, UAI 2022

[3] Copula Flows for Synthetic Data Generation, arxiv:2101.00598

**Questions:**

While the experiments on image data are quite interesting to showcase the capabilities of the copula, can the authors describe some additional sets of data that would more amenable to this method? Most image data would be well suited for diffusion models etc, but the interesting components of the copula are specifying possibly different parametric families for the marginals and it would be nice to describe that within the paper.

What is the computational cost of this method?

How often are the conditions of the copula empirically violated in practice?

---

> ### Author Response · Authors · 2025-11-21
> **Answers for Reviewer dHoi - Part 1**
>
> We sincerely thank reviewer dHoi for their time and positive review of our work. We answer your comments below.
>
> ---
>
>
> > The architectures prescribed do not preserve the necessary components to ensure a valid copula is being recovered
>
> This is indeed an open question for similar deep copula models. By our Propositions 3, 4 and 8, we are guaranteed to recover the exact copula of the data at optimality. To supplement our theoretical result with practical evidence, we included uniformity diagnostics for our models’ samples in the initial submission, verifying that they align with truly uniform samples. We have taken your feedback into account and have added quantitative metrics to further investigate the obtained samples in Apdx. B5 (please see the pdf and our general answer for full details).
>
> The results indicate that for most experiments, **our models produce samples that do not significantly deviate from prescribed marginals and do so better than the existing Ratio copula model**, as measured by a statistical test of distributional alignment.
>
> We further investigate the magnitude of deviations when they occur via the continuous ranked probability score. Our results reinforce conclusions from the existing rank histograms: our models obtain samples **closely adhering to uniformity, and significantly improve marginal sample quality compared to previous work**.
>
> Finally, an important direction for future work could be to use a similar construction to [Janke, Tim, Mohamed Ghanmi, and Florian Steinke. "Implicit generative copulas." Advances in Neural Information Processing Systems 34 (2021): 26028-26039.] while making it conform to our models’ data generating process. We have made all these points clear in Sec. 3.2, Sec. 4.2, and in Sec. 7.
>
> > The authors also miss some relevant works on deep learning and scaling copulas
>
> We thank you for pointing us to these highly relevant and interesting works, which we now discuss in the paper. We now **acknowledge these works in Section 2** (next to our revised discussion of deep Archimedean copulas) along with our previous Apdx. B which we moved to the main text.  In particular, we discuss the improved scaling of the Archimax and Archimedean works compared to earlier work on the topic, and their robustness to extremes.
>
> We now also **discuss these works in the related works in Sec. 5** of the updated pdf. We are eager to acknowledge relevant works and are happy to discuss any further suggestions we may have missed.
>
> > Can the authors describe some additional sets of data that would more amenable to this method?
>
> Yes! In Apdx. B, we discuss in more detail the Robocup experiment as a prime application of copulas for multi-agent imitation learning and broader multivariate timeseries forecasting.
>
> Such uses of copulas are a way of dealing with the exceedingly large dimensionalities of multivariate timeseries by decomposing them into marginal forecasts which are stitched together through a copula, decomposing the original problem into manageable subtasks. Copulas are especially useful in such applications as they ensure the marginal distributions are well calibrated, while also prescribing a given dependence structure.
>
> Another application could be synthetic data generation or data imputation. For instance, consider sensitive data such as criminal records about recidivism. The variables interact in complex ways with non-trivial dependencies, and obtaining more data is not feasible. Our copula models can then be used to model the data in a first step, and in a second step, impute missing values of incomplete records based on this dependence. First, samples are generated from the fitted copula model (conditional on the observed variables), and second, these generated copula samples are transformed via marginal distributions to the data scale, filling in the missing data.
>
> We thank you for raising this point and have therefore **added details to Section 2 to better describe the task of copula modelling**.

---

> > ### Author Response · Authors · 2025-11-21
> > **Answers for Reviewer dHoi - Part 2**
> >
> > > What is the computational cost of this method?
> >
> > We report computational times for all models in Apdx. C1 Tab. 4, showing that **our sampling times are competitive with vine copulas while being much faster than the Ratio copula**.
> >
> > We now additionally present a dedicated computational time study in Apdx. C1 Fig. 6. From our computational study, we conclude that the network model is the most influential, followed by the number of timesteps used to simulate the SDE/ODE when sampling. We point to this in Section 6 of the main text.
> >
> > In more detail, for $d$ the data dimension , $w$ the network width and $h$ the number of hidden layers, $c_{dc}$ training has a complexity of $\mathcal{O}(d\cdot w + h\cdot w^{2})$. This is because the model performs one forward pass (for class probabilities) and one backward pass (for scores) before taking gradients with respect to the loss during training. During inference, to sample the $c_{dc}$, we perform a forward and backward pass for each of the $k$ classes, following a SDE discretization scheme, resulting in a complexity of $\mathcal{O}(k\cdot(d\cdot w + h\cdot w^{2}))$. For the reflection copula, one training epoch consists of a single forward pass followed by a weight update, with complexity $\mathcal{O}(d\cdot w + h\cdot w^{2})$. Generating samples requires $m$ network evaluations where $m$ is the number of steps to simulate the ODE for, which scales as $\mathcal{O}(m\cdot(d\cdot w + h\cdot w^{2}))$.
> >
> > > How often are the conditions of the copula empirically violated in practice?
> >
> > We thank you for this important question. We investigate this in Apdx. B5 of the updated pdf by analysing the samples from our models, both qualitatively through rank histograms and quantitatively through statistical tests and strictly proper scoring rules. We provide details on this in our answer to your first comment as well as in our general answer on uniformity.
> >
> > The results indicate that across experiments, **our models produce samples that closely align with prescribed marginals, and in particular, do so much better than the existing Ratio copula model**.
> >
> > ---
> >
> > We thank you again for your time and effort in providing a quality review of our work, and for your positive stance on our paper. Your feedback has helped us improve the paper, both in presentation and in technical quality. If you find our answers have adequately answered your comments, we would be grateful if you might consider increasing your score. We are happy to address any further questions and remain available for the rest of the discussion period.

---

### Official Review · Reviewer_if98 · 2025-11-01

**Soundness:** 4
**Presentation:** 3
**Contribution:** 3
**Rating:** 6
**Confidence:** 4

**Summary:**

This paper introduces a novel framework for modeling multivariate dependencies using copulas, leveraging principles from diffusion and flow-based models. The core idea is to design forward processes that progressively "forget" inter-variable dependencies while provably preserving the marginal distributions. The authors propose two such processes: one based on an Ornstein-Uhlenbeck process on a Gaussian-transformed space, and another based on a reflection process on the copula hypercube.

Based on these forward processes, the paper develops two distinct copula models. The first, the "classification-diffusion copula" (cdc), is designed for accurate density estimation. It learns the copula density by framing the problem as classifying the "time" or level of dependence in the forward process. The second, the "reflection copula," is a generative model optimized for efficient sampling by learning to reverse the reflection process. A key achievement of this work is demonstrating that these models can effectively capture complex, multi-modal dependencies and scale to very high dimensions (d > 1000), a significant advancement over existing copula methods. Empirical results on scientific datasets and images show good performance.

**Strengths:**

1. The core idea of modeling dependence by learning to reverse a dependence-forgetting process is both elegant and powerful. It provides a new perspective on generative modeling that explicitly disentangles marginal behavior from joint dependence.
2. The paper is built on solid theoretical ground. The authors provide proofs for the validity of their forward processes and demonstrate that their proposed models can recover the true copula at optimality. This rigor adds significant credibility to the methods.
3. The models achieve impressive results, outperforming existing copula methods on complex scientific datasets.
4. The most significant strength is the demonstrated ability to scale copula models to thousands of dimensions, as shown with the image datasets. This was previously considered infeasible for flexible copula models and represents a major leap forward for the field.
3. The paper is clear, well-structured, and easy to read, which is commendable given the technical depth of the topic.

**Weaknesses:**

1. The loss function for the cdc model (Theorem 5) involves a hyperparameter α that balances the cross-entropy and score-matching terms. The paper states this is chosen to balance the magnitude of the terms, which is a common but heuristic approach. The paper would be stronger with a more detailed analysis of the sensitivity to this parameter or a more principled selection method.
2. The paper correctly notes that the models are only guaranteed to be valid copulas (i.e., have uniform marginals) at optimality. While the appendix includes rank histograms as a diagnostic, a more quantitative analysis (e.g., using statistical tests for uniformity) of how close the generated samples are to being marginally uniform in practice would be beneficial.
3. The FID score for the cdc model on the digits dataset is notably worse than other methods, including the simpler reflection copula. While the authors suggest the samples are "noisier," a deeper explanation for this specific performance drop would be helpful. It raises questions about the robustness of the cdc approach on certain data types.

**Questions:**

The reflection copula initializes velocities from an isotropic Gaussian distribution. Have you considered alternative or learnable initial velocity distributions? Could this provide a better inductive bias for certain types of dependencies and improve performance or training efficiency?

---

> ### Author Response · Authors · 2025-11-21
> **Answers for Reviewer if98**
>
> We sincerely thank reviewer if98 for the positive review of our work. Please find answers to your comments below.
>
> ---
>
> > The paper would be stronger with a more detailed analysis of the sensitivity to this parameter or a more principled selection method.
>
> To add rigour to our experiments, we now include an ablation of the $\alpha$ hyperparameter in Apdx. B4. **Our model is shown to be robust to values of $\alpha>0.01$ with a stable performance across different values**. This indicates that incorporating the CE term is important to achieve optimal performance, which echoes our theoretical analysis in Thm. 5,  where $\alpha>0$ is a necessary condition for optimality. Please see the general answer for more details. We now point to this ablation when introducing the loss in Sec. 3.2 as well as at the start of Sec. 6 of the main text.
>
> > A more quantitative analysis of how close the generated samples are to being marginally uniform
>
> Indeed, for our models to produce accurate copula samples, it is required that generated samples follow the respective marginal distributions. We now include a study with statistical tests for marginal distribution mismatch, as well as strictly proper scoring rule values to quantify the discrepancy in Apdx. B5. Our results show that **our models produce samples in agreement with their marginals and do so much better than existing work**.
>
> We also show that when marginal deviations happen, they are of small magnitudes as measured by the continuous ranked probability score, once again surpassing existing work at being marginally calibrated. Please see the general answer for full details.
>
> We thank you for raising this point, which has greatly solidified the quality of our results. We have made those points clear in Sec. 3.2, Sec. 4.2, and in Sec. 7.
>
> > While the authors suggest the samples are "noisier," a deeper explanation for this specific performance drop would be helpful.
>
> We now provide a more comprehensive explanation of the sampling capabilities of the cdc to expand on the “noisier” comment in the main text, and now point to the following discussion in the same passage of Sec. 6.2.
>
> A key component of our $c_{dc}$'s sampling approach is stochasticity in form of the exploration step 8 of Alg. 1. As argued in our general answer on uniformity, this exploration is beneficial for maintaining the uniformity of our samples, thus being a core strength of our model. The result of this stochasticity is that targeting very narrow parts of the support becomes more rare. We illustrate this in Fig.17, showing 2d histograms of samples from the cdc and the reflection copulas against ground truth samples for two dimensions in the magic dataset.
>
> In the case of the digits data, the **dependence is fundamentally different from the MNIST dataset, as the digits are no longer discernible on the copula scale** and appear more like random noise, see Fig. 18 of Apdx. B6. With the low sample size of <2000 compared to the dimension, only very specific regions of the support correspond to valid samples.
>
>  We believe this key difference is causing the lower FID score of the $c_{dc}$, since the FID is sensitive to such small deviations that are perceived as granularity of the image. On the other hand, the W2 prioritises adherence to the statistical properties of samples, which is reflected by the success of the $c_{dc}$ for that metric. As can be seen from FID results for other image datasets, this phenomenon largely disappears with a greater sample size. Finally, this only relates to the sampling of the cdc since our likelihoods are accurate across all settings, as verified by Tabs.1, 2, and 5.
>
> > Have you considered alternative or learnable initial velocity distributions? Could this provide a better inductive bias for certain types of dependencies and improve performance or training efficiency?
>
> This is a perceptive question. Indeed, our proof of Prop. 7 **can be extended to a broader range of distributions** due to the generality of Fejer’s Theorem. One could imagine a correlated Gaussian could help maintain samples closer to the diagonal for longer, or even possible mixtures of densities with different correlation could guide the samples to specific regions.
>
> This would require further analysis and a careful theoretical investigation of the signal-to-noise ratio induced by different velocity choices to assess the advantages. We regard this as a meaningful direction for future work, and thank you for highlighting it. We have included it in Sec. 7.
>
> ---
>
> We thank you again for your thoughtful review and support of our work. If our answers have adequately addressed your comments, we would be grateful if the reviewer might consider increasing their score. We are happy to address any further questions and remain available for the rest of the discussion period.

---

### Official Review · Reviewer_JPHs · 2025-11-01

**Soundness:** 3
**Presentation:** 3
**Contribution:** 3
**Rating:** 6
**Confidence:** 3

**Summary:**

The paper proposes a principled framework that unifies copula theory with continuous-time generative modeling. It defines marginal-preserving forward processes (diffusions or flows) that progressively remove dependence while keeping all marginals uniform, so every intermediate distribution remains a valid copula. The core mechanism uses an Ornstein–Uhlenbeck diffusion in Gaussian space followed by the Gaussian CDF, which preserves uniform marginals and converges to the independence copula. Reverse dynamics are learned either as a score-based diffusion (likelihood) or a deterministic flow (fast sampling). Experiments report strong performance on high-dimensional data versus vine and neural-flow baselines. Conceptually, the work reframes dependence modeling as a diffusion/flow on the copula manifold, yielding a mathematically grounded and scalable approach to complex dependencies.

**Strengths:**

Originality: Defining stochastic/deterministic dynamics directly on the copula manifold (uniform marginals at all times) is novel and technically

Quality: The marginal-preserving construction (OU + CDF map) is well-motivated and, as stated, supported by a formal proposition; the dual reverse instantiations (score vs. flow) are complementary and well engineered. Empirically, results indicate improved likelihoods and scalability where vines or generic neural copulas struggle.

Clarity: Exposition and notation are clean.

Significance: Establishes a geometric generative foundation for high-dimensional copulas and clarifies a long-standing issue of marginal inconsistency in prior deep copula models.

**Weaknesses:**

In practice, sampling or training in copula space requires numerical integration on $[0,1]^d$ or its Gaussian transform.  The paper does not describe how boundaries $u_j \in \{0,1\}$ are handled under discretization, nor how marginal uniformity is maintained when the OU process is approximated with finite steps.  Since the theoretical result assumes continuous time and ideal mapping, discretization could introduce marginal leakage.

The paper benchmarks primarily against classical vine and neural-flow baselines but omits newer methos such as [1].
Although the paper claims linear scaling in dimension, no runtime, or memory/parameter analysis are presented compared to other methods and as the dimension increases (in addition to Table 4).

The weighting function governs how quickly dependence is forgotten. This choice may strongly influences the likelihood–sample-quality trade-off.  The paper fixes a single schedule without ablation, leaving unclear how robust the results are to this design.


Refs:
[1] Kamthe, Sanket, Samuel Assefa, and Marc Deisenroth. "Copula flows for synthetic data generation." arXiv preprint arXiv:2101.00598 (2021).

**Questions:**

Your Proposition 2 shows marginal preservation for the OU + CDF forward process. Does this extend to other forward processes (e.g., non-OU or state-dependent drifts/β-schedules)? If not, is OU intended as the recommended path?

Your results report overall log-likelihoods and FID-type metrics, but these conflate marginal and dependence contributions. Since the defining advantage of a copula-based model lies in capturing joint dependence given fixed marginals, how do you disentangle and assess the quality of dependence fit separately from the quality of marginal fit?

---

> ### Author Response · Authors · 2025-11-21
> **Answers for Reviewer JPHs - Part 1**
>
> We sincerely thank reviewer JPHs for their positive feedback and effort in reviewing our work. Please find answers to your comments below.
>
> > how boundaries  are handled under discretization
>
> At time $0$, copula observations are always in $(0,1)$ in practice due to being continuous variables, and are therefore always finite values on the Gaussian scale. When simulating the forward process for the $c_{dc}$, OU dynamics have the advantage of having closed form Gaussian transition densities, for any fixed time $t$, including our discrete timesteps, which ensure $z_{t}$ is finite (and so $u_t \in (0,1)$). During sampling, the $c_{dc}$ operates in the same way as a standard DDPM sampler, discretising the reverse SDE via a numerical scheme. Again, here $z_t$ stays finite, and so $u_t \in (0,1)$.
>
> For the reflection forward process, particles are in $(0,1)$ with prob 1 at a fixed time due to velocities being continuous variables, meaning there are no boundary issues either. The reflection operation is analytical, so it exactly computes the position of a particle at a fixed time, having no problems with discrete time steps. For the reflection copula, the reflection operator ensures samples stay within $(0,1)$ at all times, and the continuous nature of the variables ensures none are exactly equal to $\{0,1\}$.
>
> Therefore, whether simulating the forward processes during training or using the estimated reverse process during sampling, **there are no boundary problems**. We thank the reviewer for raising this point, and have added a comment on this in Sec. 3.1.
>
> > discretization could introduce marginal leakage
>
> Potentially not respecting the model’s distribution is a feature common to most diffusion model samplers in practice due to the need to numerically simulate a continuous system. To verify that this does not prevent our model from obtaining uniform samples, we have added additional uniformity diagnostics in Apdx. B5 (please see also our general answer on the topic).
>
> Our results reinforce conclusions from the existing rank histograms: our models obtain samples **closely adhering to uniformity, and significantly improve marginal sample quality compared to previous work**. Finally, we note that LL evaluations (the $c_dc$’s primary goal) do not depend on discretisation schemes and so are unaffected. We thank you for bringing this to our attention, and have made those points clear in Sec. 3.2, Sec. 4.2, and in the conclusion.
>
> > omits newer methos such as [1]
>
> We thank the reviewer for bringing this work to our attention. We have now **included it along with other references** from reviewer dHoi, in Section 2 along with our previous Apdx. B on existing copula models. We also discuss these works in Sec. 5 of the updated pdf. As [1] does not have available code and is unpublished, we do not include it as a benchmark. Furthermore, unlike our work, it lacks similar results to our Props. 2, 7, and 9 on "forgetting and remembering" mechanisms to ensure the model evolves through valid copula states.
>
> > the paper claims linear scaling in dimension & no runtime, or memory/parameter analysis are presented compared to other methods and as the dimension increases (in addition to Table 4)
>
> We now report computational times for all models in Apdx. C1 Tab. 4, showing that **our sampling times are competitive with vine copulas while being much faster than the Ratio copula**.
>
> Additionally, we also present a **dedicated computational time study** in Apdx. C1 Fig. 6. From our computational study, we conclude that the network model is the most influential, followed by the number of timesteps used to simulate the SDE/ODE when sampling. We point to this in Section 6 of the main text.
>
> In more detail, for $d$ the data dimension , $w$ the network width and $h$ the number of hidden layers, $c_{dc}$ training has a complexity of $\mathcal{O}(d\cdot w + h\cdot w^{2})$. This is because the model performs one forward pass (for class probabilities) and one backward pass (for scores) before taking gradients with respect to the loss during training. During inference, to sample the $c_{dc}$, we perform a forward and backward pass for each of the $k$ classes, following a SDE discretization scheme, resulting in a complexity of $\mathcal{O}(k\cdot(d\cdot w + h\cdot w^{2}))$. For the reflection copula, one training epoch consists of a single forward pass followed by a weight update, with complexity $\mathcal{O}(d\cdot w + h\cdot w^{2})$. Generating samples requires $m$ network evaluations where $m$ is the number of steps to simulate the ODE for, which scales as $\mathcal{O}(m\cdot(d\cdot w + h\cdot w^{2}))$.

---

> > ### Author Response · Authors · 2025-11-21
> > **Answers for Reviewer JPHs - Part 2**
> >
> > > The weighting function governs how quickly dependence is forgotten.
> >
> > The weight $\alpha$ is governing the tradeoff between CE and MSE components of the loss, **it does not affect the forgetting of dependencies**. The forgetting of dependencies is controlled by the choice of time steps $T_1…T_k$, which we study in Apdx. B3.
> >
> > We now also provide an ablation study for the tradeoffs for the two loss terms in Apdx. B4 (please also see the general answer for more details).  We now point to this ablation when introducing the loss in Sec. 3.2 as well as at the start of Sec. 6 of the main text.
> >
> > > The paper fixes a single schedule without ablation, leaving unclear how robust the results are to this design.
> >
> > To justify our choice of scheduler, we now provide an ablation on this in Apdx. B under “Time discretization”. For the Robocup experiment, we train our $c_{dc}$ model on $90$% of the train set and use the last $10$% for validation. We report the LL for three different choices of schedulers, namely a linear optimal transport scheduler, a power-law scheduler and a KL scheduler.
> >
> > From our results, while the scheduler does not affect the LL of the train set, we show that **the KL scheduler obtains better generalizations and more stable performance** on the test set while the other two schedulers quickly overfit. We now mention this in the main text in Sec. 3.2.
> >
> > > Does this extend to other forward processes
> >
> > Certainly, other paths are an interesting direction to explore in future work. As we have noted in the paragraph on future works in the conclusion, **one can indeed construct a different path**. The following steps are needed: a) map copula scale observations to a latent scale (Gaussian for the OU) via the marginal inverse CDF $\phi^{-1}$ of an adequate distribution. b) Use a stochastic process with $\phi$ as the invariant or stationary distribution, such that initialising the process from it guarantees the marginals to be maintained with time. c) Run the process independently across dimensions, which under suitable assumptions will converge to the factorisation of marginals, forgetting the dependence and resulting in an independent copula. We view the exploration of other processes, e.g. extreme value distributions for extreme dependencies, as a fruitful direction for future work. We have expanded our discussion of this in Sec. 7 in the paper.
> >
> > > If not, is OU intended as the recommended path?
> >
> > The OU is the recommended path choice since it is a widely adopted choice in practice with methods designed with it in mind, and its properties are well studied from the theoretical side too (please see our discussion before Prop. 2). For our purposes, the Gaussianity of **the OU provides a well-motivated and interpretable transformation from the copula scale**. We have made this clearer in Sec. 3.1.
> >
> > > Your results report overall log-likelihoods and FID-type metrics, but these conflate marginal and dependence contributions. How do you disentangle and assess the quality of dependence fit separately from the quality of marginal fit?
> >
> > We report log-likelihoods of the copula throughout the paper, which **only measure the fit of the dependence**. We also report the Frob norm of Kendal’s tau matrices, which is purely a dependence metric. We agree that the FID does not isolate dependence, but as all samples are measured on a common scale with the same marginals, the main disparities stem from the dependence structure, with possible contributions due to departures from uniformity of the marginals, both of which are relevant for copula sampling.
> >
> > We further assess purely the quality of marginal samples in Apdx. B5, via quantitative metrics and statistical tests, showing that our models achieve samples which respect uniformity, with a noticeable improvement over the Ratio copula model. (please see the general answer for further details on this). We expanded the paragraph on Benchmarks and metrics in Section 6 to make this clear.
> >
> > ---
> >
> > We thank you again for reviewing our work. If our answers have adequately addressed your comments, would be grateful if the reviewer might consider increasing their score. We are happy to address any further questions and remain available for the rest of the discussion period.

---

> > > ### Comment · Reviewer_JPHs · 2025-11-26
> > >
> > > Thank you for the detailed and thoughtful rebuttal. I appreciate the effort you have taken to address the concerns raised. The new runtime analysis is appreciated, and the ablations on the scheduling choices are helpful additions. The clarification regarding dependence versus marginal fit metrics is also useful. Finally, thank you for incorporating additional references and for discussing Copula Flows; I understand that the absence of available code makes benchmarking against this method challenging.

---

> > > > ### Author Response · Authors · 2025-11-26
> > > > **Thank you!**
> > > >
> > > > Thank you very much for your engagement in the discussion phase and for the positive follow‑up comments. If there are no remaining suggestions or unaddressed comments, we would be grateful if this could be reflected in the final evaluation.

---

### Author Response · Authors · 2025-11-21
**Common Answer - Part 1**

Dear reviewers,

We are sincerely thankful for your time and effort in reviewing our paper. We are encouraged by your comments denoting our work as:

-	**A Significant contribution**:
“clearly outperform classical (e.g., Vine) and other deep copulas, which struggle with scalability and complexity” (5pKF),
“This was previously considered infeasible […] and represents a major leap forward for the field.” (if98),
“clarifies a long-standing issue of marginal inconsistency” (JPHs),
“scales reliably […] which has been somewhat of an elusive property of copulas.” (dHoi),
-	**Clear**
“Exposition and notation are clean” (JPHs),
“sound and well-motivated”(5pKF),
“clear, well-structured, and easy to read, which is commendable given the technical depth” (if98),

-	**Rigorous**
“fresh and theoretically grounded perspective on copula modeling” (5pKF),
“novel and technical” (JPHs),
“solid theoretical ground […]. This rigor adds significant credibility” (if98).

We address each individual point under your comments and additionally summarize the main improvements thanks to your reviews below this message. We give a comprehensive evaluation of different components of our model, including **ablations on the loss mixture** in Apdx. B4, an **improved computational analysis** in Apdx. B1, an **ablation on schedulers** in Apdx. B under “Time discretization”, and **additional uniformity diagnostics** in Apdx. B5. Relevant parts of our replies have also been added to the main text in the updated pdf.

---

## Ablations on the loss mixture:

In Appendix B4 of the revised pdf, we provide an ablation to investigate the sensitivity of the $c_{dc}$ to the choice of $\alpha$. We point to this ablation when introducing the loss in Section 3 as well as at the start of Section 6 of the main text.

We perform a sweep over values $\alpha=(0.0,0.01,0.25,0.5,0.75,1.0,5.0,20.0,500.0)$ for the experiments on all scientific datasets. We report the resulting LL values on a held-out validation set ($10\%$ of the train set) in Fig. 9. **Our model is shown to be robust to all values of $\alpha>0.01$, resulting in a stable performance**.

This clarifies that incorporating the CE term is important to achieve optimal copulas, which echoes our theoretical analysis in Thm. 5 where $\alpha>0$ is a necessary condition for optimality.

---

## Improved computational analysis:
We report computational times for all models across all experiments in Apdx. C1 Tab. 4, **showing that our sampling times are competitive with vine copulas while being much faster than the Ratio copula**.

We now additionally present a dedicated computational time study in Apdx. C1 Fig. 6. From our computational study, we conclude that the network model is the most influential, followed by the number of timesteps used to simulate the SDE/ODE when sampling. We point to this in Section 6 of the main text.

In more detail, for $d$ the data dimension , $w$ the network width and $h$ the number of hidden layers, $c_{dc}$ training has a complexity of $\mathcal{O}(d\cdot w + h\cdot w^{2})$. This is because the model performs one forward pass (for class probabilities) and one backward pass (for scores) before taking gradients with respect to the loss during training. During inference, to sample the $c_{dc}$, we perform a forward and backward pass for each of the $k$ classes, following a SDE discretization scheme, resulting in a complexity of $\mathcal{O}(k\cdot(d\cdot w + h\cdot w^{2}))$. For the reflection copula, one training epoch consists of a single forward pass followed by a weight update, with complexity $\mathcal{O}(d\cdot w + h\cdot w^{2})$. Generating samples requires $m$ network evaluations where $m$ is the number of steps to simulate the ODE for, which scales as $\mathcal{O}(m\cdot(d\cdot w + h\cdot w^{2}))$.

---


## Ablation on schedulers:
To justify our choice of scheduler, we now provide an ablation on this in Apdx. B under “Time discretization”. We mention it in the main text in Sec. 3.2.

For the Robocup experiment, we train our $c_{dc}$ model on $90$% of the train set and use the last $10$% for validation/testing. We report the LL for three different choices of schedulers, namely a linear optimal transport scheduler, a power-law scheduler and a KL scheduler. While the scheduler choice does not affect the LL of the train set, we show that **the KL scheduler is more stable and generalises better from the train to the test set**. This empirically demonstrates our result in Prop. 2 as the KL scheduler is informed by the convergence rate of our forward process.

---

> ### Author Response · Authors · 2025-11-21
> **Common Answer - Part 2**
>
> # Additional uniformity diagnostics:
>
> For our models to produce accurate copula samples, it is required that generated samples follow the respective marginal distributions. For the $c_{dc}$, as the model operates on the Gaussian scale, we now include an additional check for the normality of the generated samples via the statistical test of [1]. For the reflection copula which operates on the hypercube, we verify the uniformity of samples with a two-sample Kolmogorov-Smirnov test for goodness of fit [2].  Along with the Ratio copula (operating on the Gaussian scale), we report the rejection rate of the hypothesis that marginals are preserved, averaged over dimensions and model runs, at the $5$% alpha level with a Bonferroni correction due to multiple testing.
>
> The results, in Apdx. C5 Tab. 6, indicate that across experiments, **our models produce samples that do not significantly deviate from prescribed marginals and do so better than the existing Ratio copula model**. In particular, our $c_{dc}$ achieves a small rejection rate, which we attribute to the sampling procedure using Gaussian noise during sampling (see step 8 in Alg. 1). Note that this result only indicates departures from marginals, but does not inform about their severity.
>
> To strengthen these findings, in Tab. 6, we additionally report the Continuous Ranked Probability Score (CRPS) as a univariate probabilistic calibration metric to compare our generated copula samples to uniform data, dimension-wise. As a strictly proper scoring rule [3], it is minimised if and only if both samples come from the same distribution [4], here being the uniform on $[0,1]$. In Tab. 6, we report the mean CRPS averaged across dimensions and runs, from which we subtract the CRPS obtained with pairs of uniform samples (computed to be 0.1666 over 100 repetitions of 1000 vs 1000 draws). Thus, the ideal value for this metric would be 0, uniquely achieved by a uniform sample.
>
> Our results corroborate the statistical tests, showing that **our models achieve samples very close to uniformity, with a noticeable improvement over the Ratio copula model**.We have added sentences to the main text referencing the adherence of marginals and this study at the end of Section 3 and of Section 4, as well as at the start of Section 6.
>
> | Metric                  | Col1   | Col2   | Col3   | Col4   | Col5   | Col6   |
> |-------------------------|--------|--------|--------|--------|--------|--------|
> | $c_{dc}$ - test rejection | 0.01   | 0.01   | 0.00   | 0.03   | 0.00   | 0.00   |
> | Reflection - test rejection | 0.02   | 0.09   | 0.00   | 0.16   | 0.01   | 0.70   |
> | Ratio - test rejection    | 0.47   | 0.93   | 0.16   | 0.30   | 0.09   | 0.50   |
> | $c_{dc}$ - CRPS          | 0.0029 | 0.0021 | 0.0021 | 0.0068 | 0.0022 | 0.0041 |
> | Reflection - CRPS        | 0.0024 | 0.0030 | 0.0022 | 0.0036 | 0.0024 | 0.0142 |
> | Ratio - CRPS             | 0.0169 | 0.0128 | 0.0028 | 0.0150 | 0.0070 | 0.0680 |
>
> [1] D’Agostino, R. and Pearson, E. S. (1973), “Tests for departure from normality”, Biometrika, 60, 613-622
> [2] Hodges, J.L. Jr., “The Significance Probability of the Smirnov Two-Sample Test,” Arkiv fiur Matematik, 3, No. 43 (1958), 469-486.
> [3] Gneiting, Tilmann, and Adrian E. Raftery. "Strictly proper scoring rules, prediction, and estimation." Journal of the American statistical Association 102.477 (2007): 359-378.
> [4] Dawid, A. Philip, Monica Musio, and Laura Ventura. "Minimum scoring rule inference." Scandinavian Journal of Statistics 43.1 (2016): 123-138.

---

### Meta-Review · Area_Chair_fXKV · 2026-01-05

**Summary:**

Te main concerns are :

- Unclear role of discretization (which might cause marginal leakage)
- Experimental validation: missing strong baseline, missing ablation studies, unclear parameter sensitivity, long raining times and further scalability questions,  etc.
- Open questions about the validity of the copulas
- Missing related work & discussion

**Reviewer Concerns:**

The more fundamental concerns about the validity of the copulas and the role of discretization could be addressed in a convincing way in the rebuttal. Further, additional experiments seem to answer several questions about computational costs and quality of the results.

**Reviewer Scores:**

I guess that the first reviewer slightly increased the score.

---

### Decision · Program_Chairs · 2026-01-26

Accept (Poster)